# PENCIL: Long Thoughts with Short Memory

**Chenxiao Yang** [1]  **Nathan Srebro** [1]  **David McAllester** [1]  **Zhiyuan Li** [1]

## Abstract

While state-of-the-art LLMs have demonstrated great promise of using long Chains-of-Thought (CoT) to boost reasoning, scaling it up to more challenging problems is fundamentally limited by suboptimal memory usage — intermediate computations accumulate indefinitely in context even no longer needed for future thoughts. We introduce PENCIL, which incorporates a novel reduction mechanism into the autoregressive generation process that recursively clean up intermediate thoughts based on patterns learned from training. By alternately generating and erasing, PENCIL can think deeper to solve harder problems using shorter context and less computes. Empirically, for example, we demonstrate PENCIL with a small 25M-parameter transformer and 2048 context length solves Einstein's puzzle — a task that challenges much larger models like GPT-4. Theoretically, we prove PENCIL can perform universal efficient computation by simulating any Turing machines with optimal time and space complexity, and thus can solve arbitrary computable tasks that are otherwise intractable for vanilla CoT.

## 1. Introduction

Recently, there has been a surge of interest in reasoning with *Chain-of-Thought* (CoT) (Wei et al., 2022) and generating longer thoughts at test-time to tackle larger-scale and more complicated problems (OpenAI, 2024; Guo et al., 2025; Snell et al., 2024; Muennighoff et al., 2025). CoT is an iterative generation process: each intermediate reasoning step is appended to the current context and treated as the input in subsequent reasoning. The context grows until reaching a final answer. While such an iterative model is theoretically powerful – capable, in principle, of tackling many intricate problems given unlimited length (Merrill &

Sabharwal, 2023; Feng et al., 2024; Li et al., 2024b) – it suffers from the inherent *write-only* limitation: partial computation remains in the context even when no longer needed for future thought generation. This design becomes particularly problematic for inherently hard reasoning tasks, where no efficient algorithm exists and thus reasoning inevitably spans many steps, forcing the context length to grow indefinitely. This not only demands excessive memory resources that become impractical for computationally hard tasks, but could also degrades the model's ability to effectively retrieve information in the context, even when the maximum length is not exceeded (Liu et al., 2024).

Memory management is a major issue in modern computer systems. Turing machines, for example, can overwrite tape cells and reclaim space for new computations, while high-level programming languages rely on stack frames, function calls, and garbage collection to discard unneeded data. While some previous works have attempted to augment LLMs with external memory (e.g. (Gao et al., 2023; Wang et al., 2024)), they often lack a direct mechanism for reclamation of no longer needed memory as stack deallocation or garbage collection. This paper proposes **PENCIL**, [1] which introduces cleaning mechanisms to CoT for space-efficient and long-chain reasoning.

In a nutshell, PENCIL combines a next-token generator (e.g., a decoder-only transformer) and a *reduction rule*, and applies the reduction rule whenever possible throughout the standard iterative next-token generation process to reduce context length. In this paper, we focus on a simple yet universal reduction rule motivated by the function call stack in modern computers.

$$\mathbf{C}\,[\texttt{CALL}]\,\mathbf{T}\,[\texttt{SEP}]\,\mathbf{A}\,[\texttt{RETURN}]\;\Rightarrow\;\mathbf{C}\,\mathbf{A} \qquad (1)$$

where [CALL], [SEP], and [RETURN] are special tokens that separate the context ($\mathbf{C}$), thoughts ($\mathbf{T}$), and answer ($\mathbf{A}$) in the sequence. Once a computation completes (marked by [RETURN]), all intermediate reasoning steps (those between [CALL] and [SEP]) will be removed, merging the answer back into the context. Importantly, this process can be applied recursively, allowing for hierarchical reasoning structures similar to nested function calls in programming. PENCIL alternates between standard CoT-style generation

---

[1]Toyota Technological Institute at Chicago. Correspondence to: Chenxiao Yang <chenxiao@ttic.edu>.

*Proceedings of the 42nd International Conference on Machine Learning*, Vancouver, Canada. PMLR 267, 2025. Copyright 2025 by the author(s).

[1]**P**ENCIL **EN**ables **C**ontext-efficient **I**nference and **L**earning

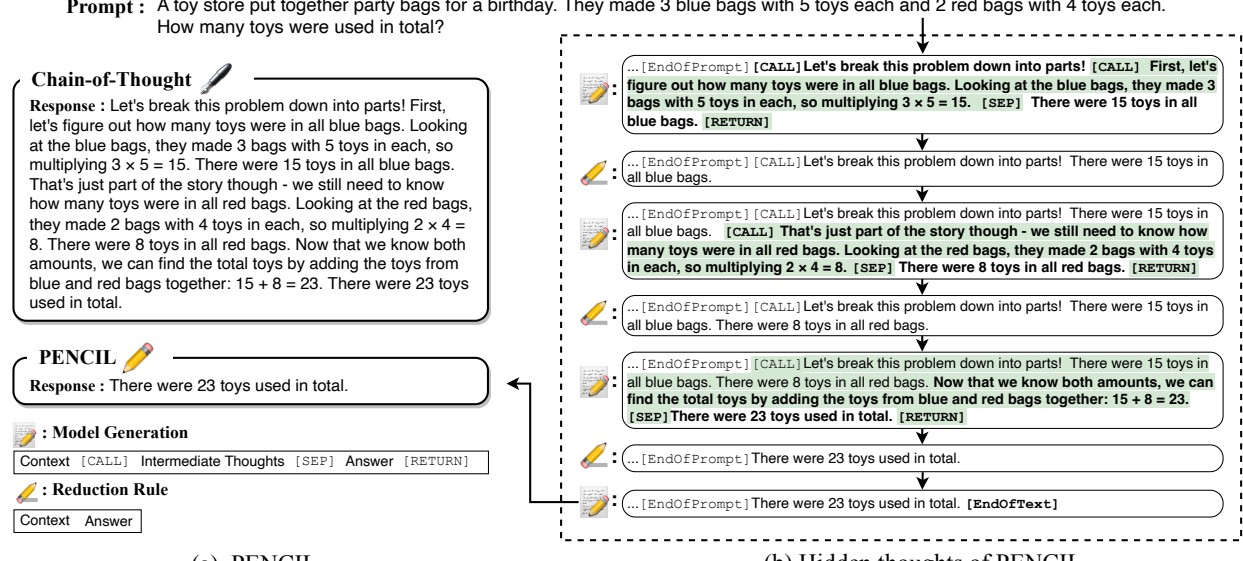

Figure 1: A toy example illustrating how PENCIL would potentially solve an arithmetic problem. Bold text indicates content generated in the current iteration, content highlighted in blue indicates intermediate thoughts to be erased by the reduction rule. See a concrete example of the complete thinking process for solving QBF in Fig. 2 and Einstein's puzzle in Fig. 8.

and this reduction step, automatically discarding unneeded thoughts based on patterns learned from training. Figure 1 gives a hypothetical example of how PENCIL might be applied to natural language thoughts.

We train and evaluate PENCIL on SAT, QBF, and Einstein's puzzle — tasks that inherently require exponential computation time. PENCIL effectively reduces the maximal CoT length (i.e. the space requirement) from exponential to polynomial. Consequently, under fixed architecture and context window, PENCIL allows solving larger-sized problems whereas CoT fails due to exploding context length. Furthermore, by continually discarding irrelevant tokens, PENCIL can significantly save training computes and converge faster even when memory or expressiveness is not a bottleneck. Notably, on the 5×5 Einstein puzzle – a challenging natural-language logic puzzle that even large models like GPT-4 struggle with – PENCIL achieves a 97% success rate by using a small transformer with 25M-parameter and 2048-token context.

Theoretically, we show that PENCIL with a fixed finite-size decoder-only transformer can perform universal space-efficient computation, by simulating Turing machine running in $T$ steps and $S$ space with $\mathcal{O}(T)$ generated tokens and maximal sequence length $\mathcal{O}(S)$. This indicates its power for solving any computational tasks with optimal time and space efficiency. This is a significant improvement over standard CoT, which require context length to grow proportionally with $\mathcal{O}(T)$, making them fundamentally unable to solve problems requiring extensive computation within fixed memory constraints.

See discussions about related work in Appendix A. Codes are available at https://github.com/chr26195/PENCIL.

## 2. PENCIL: Iterative Generation & Reduction

Chain-of-Thought (CoT) (Wei et al., 2022) allows language models to generate intermediate reasoning steps before producing a final answer. Formally, given a finite alphabet $\Sigma$, let $\pi : \Sigma^* \to \Sigma$ be a *next-token predictor*, which maps an input sequence $(x_1, x_2, \cdots, x_n) \in \Sigma^n$ to the next token $x_{n+1} \in \Sigma$. Correspondingly, we can define a sequence-to-sequence mapping $f : \Sigma^* \to \Sigma^*$ as

$$f_\pi(x_1, \ldots, x_n) \triangleq (x_1, \ldots, x_n, \pi(x_1, \ldots, x_n)) \quad (2)$$

which concatenates the next token to the current context. For brevity, we will write $f$ instead of $f_\pi$ when the context is clear. CoT with $k$ steps is denoted as $f^k : \Sigma^* \to \Sigma^*$, where $f^k \triangleq f \circ f^{k-1}$ and $f^1 \triangleq f$. Given any input sequence $x = (x_1, x_2, \ldots, x_n) \in \Sigma^n$, each application of $f$ extends the sequence by one token, such that $f^k(x) \in \Sigma^{n+k}$. Throughout this paper, we use shorthand $x_{:j}$ to denote $(x_1, \ldots, x_j)$, and $x_{i:j}$ the subsequence from $x_i$ to $x_j$, for any string $x \in \Sigma^*$ longer than $j$.

The iterative generation process of CoT is inherently limited by its write-once nature; that is, once written, intermediate computations permanently occupy the context, regardless of their relevance in the subsequent reasoning steps. Consequently, the context length would eventually grow overwhelmingly large for complex reasoning problems. To address this, we introduce PENCIL, which is CoT equipped

with a reduction rule that enables selective elimination of reasoning traces, allowing the model to generate longer thoughts to solve larger problems with less memory.

## 2.1. The Reduction Rule and PENCIL

A *reduction rule* (a.k.a. rewriting rule) (Baader & Nipkow, 1998) is a formal mechanism originated from logic for transforming one expression to another via predefined patterns and ultimately reaching a final normal form, i.e. the answer. It serves as a fundamental model of computation in classic functional programming languages such as $\lambda$-calculus (O'Donnell, 1985), and proof assistants for automated theorem proving and reasoning (Wos et al., 1992). Mathematically, the reduction rule can be thought of as a sequence-to-sequence function $\phi : \Sigma^* \to \Sigma^*$, which in this paper is from a longer sequence $(x_1, \ldots, x_a) \in \Sigma^a$ to a shorter one $(x_{i_1}, \ldots, x_{i_b}) \in \Sigma^b$ where $b \leq a$.

**The Reduction Rule**  Let $\hat{\Sigma} = \Sigma \cup \{$ [CALL], [SEP], [RETURN]$\}$ be the extended alphabet including three special tokens that indicate certain structures of the reasoning trace. Given the new alphabet, we can instantiate the rule $\phi$ as (1), where

$$
\begin{aligned}
\mathbf{C} &\in (\Sigma \cup \{[\text{CALL}], [\text{SEP}], [\text{RETURN}]\})^* \\
\mathbf{T} &\in (\Sigma \cup \{[\text{SEP}], [\text{RETURN}]\})^* \\
\mathbf{A} &\in (\Sigma \cup \{[\text{CALL}]\})^*
\end{aligned}
\tag{3}
$$

are subsequences separated by the special tokens. The allowance of difference special tokens in $\mathbf{C}$, $\mathbf{T}$, $\mathbf{A}$ ensures that: 1) the [RETURN] token is the last [RETURN] token in the sequence; 2) the [SEP] token in (1) is the one immediately before the [RETURN] token ; 3) and the [CALL] token is immediately before the [SEP] token.

Intuitively, $\mathbf{C}$ can be understood as context that can include information that is either directly relevant to solving the current problem or irrelevant but useful for solving future problems; $\mathbf{T}$ represents the intermediate thoughts for deriving the answer and $\mathbf{A}$ represents the answer. If the input sequence satisfy the pattern $\mathbf{C}$ [CALL] $\mathbf{T}$ [SEP] $\mathbf{A}$ [RETURN], the rule will activate. Consequently, the entire intermediate thoughts and the special token triplet will be removed, with the answer being merged back into the context. Otherwise if the pattern is not satisfied, the rule will leave the input sequence unchanged.

It is important to note that the inclusion of [CALL] in $\mathbf{C}$ enables nested reasoning structures critical for achieving optimal space efficiency, while allowing [CALL] in $\mathbf{A}$ enables tail recursion optimization for better efficiency as will be discussed in Sec. 3.

**PENCIL** consists of a learnable next-token predictor $f$ as defined in (2) which is responsible for generating the intermediate reasoning steps (including special tokens [CALL],

[SEP], [RETURN]) as in the standard CoT, and the reduction rule $\phi$ as defined in (1) that serves to reduce the context and clean the memory. Formally, we define one step and $k$-steps of PENCIL as $\text{PENCIL}^1_{\phi,f} = \phi \circ f$ and $\text{PENCIL}^k_{\phi,f} = (\phi \circ f)^k$. Namely, each step of PENCIL first generates the next token as in standard CoT and then applies the reduction rule $\phi$, deleting the intermediate computations if the new sequence matches the pattern.

## 2.2. Alternated Generation and Reduction Process

The alternated generation and reduction process of PENCIL can also be interpreted by grouping the $f$ functions that are interleaved by ineffective reduction steps (where $\phi$ does not match the pattern):

$$
\text{PENCIL}^k_{\phi,f} = f^{k_{r+1}} \circ \phi \circ f^{k_r} \circ \phi \circ \cdots \circ \phi \circ f^{k_1} \tag{4}
$$

where $k = \sum_{i=1}^{r+1} k_i$, and $k_i$ denotes the number of tokens generated between the $(i-1)$-th and $i$-th effective reduction. Here $r$ is the total number of effective reductions, assuming the model terminates with a [EOS] token indicating stop generation. This process alternates between two phases

$$
\begin{aligned}
\textbf{Generation:} \quad & x^{(i)} \triangleq f^{k_i} \circ \underbrace{\phi \cdots \phi \circ f^{k_1}(x)}_{x^{(i-0.5)}}, \\
\textbf{Reduction:} \quad & x^{(i+0.5)} \triangleq \phi \circ \underbrace{f^{k_i} \cdots \phi \circ f^{k_1}(x)}_{x^{(i)}}
\end{aligned}
\tag{5}
$$

where $x^{(i)}$ represents a generated sequence ending with [RETURN] except for $x^{(r+1)}$ which ends with the [EOS] token, and $x^{(i+0.5)}$ represents the reduced sequence after each effective reduction, with $x^{(0.5)} \triangleq x$ defined as the input prompt. The complete reasoning trace can be expressed as:

$$
x \xrightarrow{f^{k_1}} x^{(1)} \xrightarrow{\phi} x^{(1.5)} \cdots x^{(r+0.5)} \xrightarrow{f^{k_{r+1}}} x^{(r+1)} \tag{6}
$$

That is, at each iteration $i$, PENCIL first generates from $x^{(i-0.5)}$, which could be understood as the prompt for the current iteration, to $x^{(i)}$, a prompt-response pair that ends with the [RETURN] token; then PENCIL applies the reduction rule to transform the prompt-response pair $x^{(i)}$ into a new prompt $x^{(i+0.5)}$ for the next iteration $i + 1$.

**Space Efficiency**  To compare the space efficiency of CoT and PENCIL, we define *scaffolded CoT* as the trace that would be produced by PENCIL but without actually removing the thoughts. (We refer to it as "scaffolded" because it includes the special tokens that mark the hierarchical reasoning structure.) Formally, for any input sequence $x$, scaffolded CoT is defined as

$$
(x \ , \ x^{(1)} \backslash x^{(0.5)} \ , \ \cdots \ , \ x^{(r+1)} \backslash x^{(r+0.5)}) \tag{7}
$$

where $x^{(i)} \backslash x^{(i-0.5)}$ represents the tokens generated at iteration $i$. The max sequence length in PENCIL is

**Prompt :** ∃ 2 ∀ 1 : #1 ( 2 ∨ ¬ 2 ∨ 1 ) #2 ( 1 ∨ 2 ) #3 ( 2 ) #4 ( ¬ 2 ∨ ¬ 1 ) #5 ( 1 ∨ ¬ 1 ) #6 ( ¬ 1 ∨ ¬ 2 )

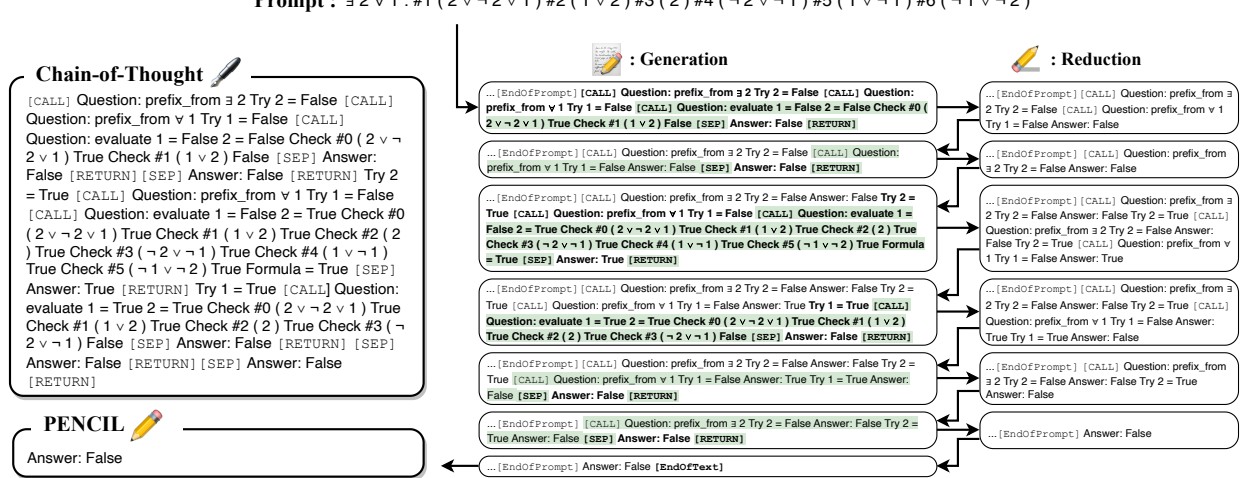

Figure 2: The complete thinking process of PENCIL on a small-sized QBF instance. The "..." at the beginning of a thought hides the prompt. Bold text represents newly generated thoughts, while blue highlights indicate thoughts to be removed.

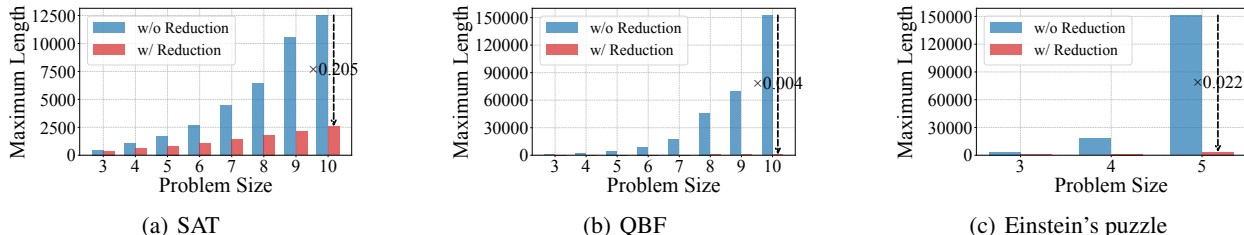

(a) SAT      (b) QBF      (c) Einstein's puzzle

Figure 3: Maximal sequence length with and without the reduction rule.

$\max_{i \in [r+1]} \{|x^{(i)}|\}$, whereas CoT has a length of $n + k$. As we will demonstrate in Sec. 3, their difference becomes particularly significant (i.e. $\max_{i \in [r+1]} \{|x^{(i)}|\} \ll n + k$) for complex reasoning tasks, where the context length of CoT can grow exponentially while the context length length of PENCIL is kept polynomial.

**Computational Benefits** Moreover, even though the total number of predicted tokens or reasoning steps is the same with or without reduction, PENCIL can significantly save computes by maintaining a substantially shorter context for each generated token. We discuss in Appendix B the computational benefit of PENCIL in terms of the FLOPs for generating a sequence and empirically quantify it in Sec. 4.

## 3. Thinking with PENCIL

We next demonstrate how the reduction rule can be applied to several concrete computationally intensive problems.

### 3.1. SAT and QBF

**SAT** is a canonical NP-complete problem. We consider the 3-SAT variant, where each instance is a Boolean formula in conjunctive normal form with clauses of length three, e.g. $(x_1 \vee \neg x_2 \vee x_3) \wedge (\neg x_1 \vee x_2 \vee \neg x_3)$. The ratio between number of clauses and variables is set as $4.3$, larger than the

threshold $4.267$ where instances are empirically hardest to solve and satisfiability probability transitions sharply from 1 to 0 (Selman et al., 1996). **QBF** is a PSPACE-complete problem that generalizes SAT by adding universal (∀) and existential (∃) quantifiers. Each instance is a quantified Boolean formula in Prenex normal form, e.g., $\exists x_1 \forall x_2 \exists x_3 : (x_1 \vee \neg x_2 \vee x_3) \wedge (\neg x_1 \vee x_2 \vee \neg x_3)$. We set the probability of a variable being existentially quantified as $0.5$.

We consider using the DPLL algorithm to solve the SAT problem, and solving the QBF problem by recursively handling quantifiers and trying variable values. The PENCIL reasoning traces are generated as we run the algorithm. Both algorithms recursively explore variable assignments by splitting on an unassigned variable $x_i$ and trying branches $x_i = \text{True}$ and $x_i = \text{False}$. The reduction rule wraps each branch with [CALL], [SEP] and [RETURN], which creates a hierarchical binary tree structure. See Fig. 2 for a concrete example.

Without the reduction rule, the context must retain the complete recursive trace — all partial assignments and intermediate formulas — leading to worst-case exponential space complexity $\mathcal{O}(2^n)$. For PENCIL, once a branch returns, its intermediate reasoning steps are discarded, therefore search paths will be discarded, preserving only the final answer.

This reduces the maximal length to $\mathcal{O}(n)$, bounded by the search tree depth. As shown in Fig. 3, at $n = 10$, the maximal sequence length drops from $13,804$ to $2,507$ for SAT and from $151,661$ to $649$ for QBF.

## 3.2. Einstein's Puzzle

**Einstein's Puzzle**    We further consider Einstein's puzzle (Prosser, 1993), a classic constraint satisfaction problem where the model must learn to reason in natural language. Each problem instance consists of a list of houses with different attributes (e.g., color, nationality, pet), and given a set of constraints or clues as the prompt (e.g. the green house is immediately to the right of the one who keeps birds), the goal is to determine the attributes of each house through logical deduction. The original puzzle has size 5 × 5 (5 houses and 5 attribute categories, totaling 25 variables), which presents a significant challenge for language models to solve – even GPT-4 fails to solve it with few-shot CoT (Dziri et al., 2024).

**Special Use Case: Summarization**    A notable special case of the reduction rule used for solving Einstein's puzzle is when the answer itself leads to another question: when **A** = [CALL] **T'**, (1) becomes

$$\begin{aligned} & \mathbf{C} \ [\text{CALL}] \ \mathbf{T} \ [\text{SEP}] \ [\text{CALL}] \ \mathbf{T'} \ [\text{RETURN}] \\ \Rightarrow \ & \mathbf{C} \ [\text{CALL}] \ \mathbf{T'}. \end{aligned} \quad (8)$$

This mimics the tail recursion in functional programming where a function's returned value is another function call. A practical application of this rule is to simplify an originally complex question by iteratively reducing it, through some intermediate reasoning steps, to a more tractable form. In Sec. 5 we will use this to prove PENCIL's space efficiency.

See Fig. 8 for an illustration of how reduction rules can be applied to solve the Einstein puzzle, which consists of the following steps in one round of iteration: **(a)** Propagating constraints to eliminate impossible attributes combinations; **(b)** Use the tail recursion rule to merge results from constraints propagation and update the house states; **(c)** Iteratively explore different solution branches and discard intermediate reasoning steps from each branch, only preserving the final answer. As shown in Fig. 3, for 5×5 puzzle, the maximal sequence reduces dramatically from $151,192$ to $3,335$ (without tail recursion this number is $7,705$).

## 4. Experiments

**Training**    The training of PENCIL is nearly identical to that of CoT with a key difference being how the data is processed. Specifically, the training pipeline of PENCIL consists of the following steps:

*For data preparation*, we implement the algorithms for solving the problems mentioned in Sec. 3, generates the corresponding scaffolded CoT (7) with special tokens [CALL],

[SEP], [RETURN] as we run the algorithm, and then transform the long scaffolded CoT sequence into a set of smaller sequences $\{x^{(1)}, x^{(2)}, \ldots, x^{(r+1)}\}$ that ends with either [RETURN] or EOS.

*During training*, the loss function is crucial for the success of training PENCIL. In particular, we need not compute loss on every single token in each shorter sequence $x^{(i)}$, but only those that are generated starting from last iteration's reduction step (i.e. $x^{(i)} \backslash x^{(i-0.5)}$). We maintain an index for each $x^{(i)}$ for storing the information of the index where the model generation starts. We can either feed all shorter sequences into one batch (which is our default choice in experiments), which makes it possible to reuse the KV cache of other sequences to reduce training computes, or randomly sample from these sequences from all problem instance, which would lead to similar performance.

**Implementation**    Unless otherwise stated, for model architecture, we choose a 6-layer transformer with 10.63M parameters for SAT and QBF problems, and an 8-layer transformer with 25.19M parameters for the more complex Einstein's puzzle. All experiments use a context window of 2048 tokens and rotary positional encoding (Su et al., 2024); we truncate the sequence to the maximal context window to fit into the model for all methods if it exceeds the model's capacity. We use the same batch size and learning rate for all methods across experiments.

**Experimental Setting**    We adopt the online learning setting where models train until convergence with unconstrained data access, mirroring the common scenarios in language model training where data can be effectively infinite (Hoffmann et al., 2022). To ensure fair comparison, we include special tokens in the CoT, which might benefit its training by introducing additional structural information.

**Evaluation Protocol**    We evaluate on a held-out validation set of 100 problem instances using two metrics: accuracy (percentage of correct predictions) and trace rate (percentage of reasoning steps matching the ground truth). For all problems, the labels for different classes are balanced.

### 4.1. Results on SAT and QBF

**Performance**    As shown in Table 1, both CoT and PENCIL significantly outperform the baseline (i.e. without using CoT) and achieve almost perfect performance ($\geq 95\%$ accuracy) on small problems ($n \leq 6$ for SAT and 5 for QBF). While CoT's performance degrades sharply when problem size increases - dropping to 50% accuracy on SAT and 61% on QBF when $n = 10$, PENCIL maintains near-perfect accuracy across all problem sizes. Furthermore, PENCIL's consistently high trace rate (above 90% for most problem sizes) indicates that it precisely follows the intended algorithm's reasoning steps.

| $n =$ | 3 | 4 | 5 | 6 | 7 | 8 | 9 | 10 |
|---|---|---|---|---|---|---|---|---|
| Baseline Acc. | 66 | 57 | 46 | 51 | 46 | 51 | 49 | 51 |
| CoT Acc. | 100 | 100 | 100 | 99 | 84 | 63 | 54 | 50 |
| CoT TR. | 99.6 | 99.0 | 98.0 | 96.2 | 74.0 | 69.9 | 63.8 | 51.4 |
| PENCIL Acc. | 100 | 100 | 100 | 99 | 99 | 100 | 100 | 100 |
| PENCIL TR. | 100 | 99.0 | 97.1 | 95.9 | 91.8 | 93.3 | 92.9 | 83.0 |

| $n =$ | 3 | 4 | 5 | 6 | 7 | 8 | 9 | 10 |
|---|---|---|---|---|---|---|---|---|
| Baseline Acc. | 90 | 82 | 85 | 68 | 60 | 69 | 71 | 66 |
| CoT Acc. | 100 | 100 | 97 | 94 | 74 | 72 | 69 | 73 |
| CoT TR. | 100 | 100 | 98.3 | 93.9 | 65.1 | 49.4 | 40.7 | 32.8 |
| PENCIL Acc. | 100 | 100 | 100 | 100 | 100 | 100 | 100 | 100 |
| PENCIL TR. | 100 | 100 | 100 | 100 | 100 | 100 | 100 | 100 |

Table 1: Performance on SAT (left) and QBF (right). Acc denotes the Accuracy (%) and TR denotes the trace rate (%).

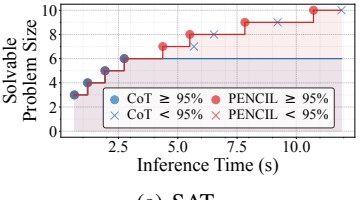

(a) SAT

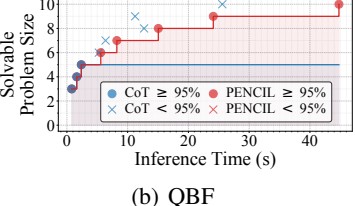

(b) QBF

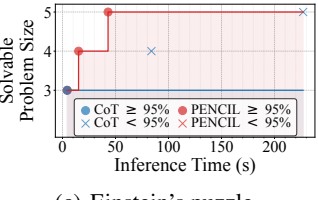

(c) Einstein's puzzle

Figure 4: Comparison of maximally solvable problem size (with $\geq 95\%$ accuracy) given different inference time budgets.

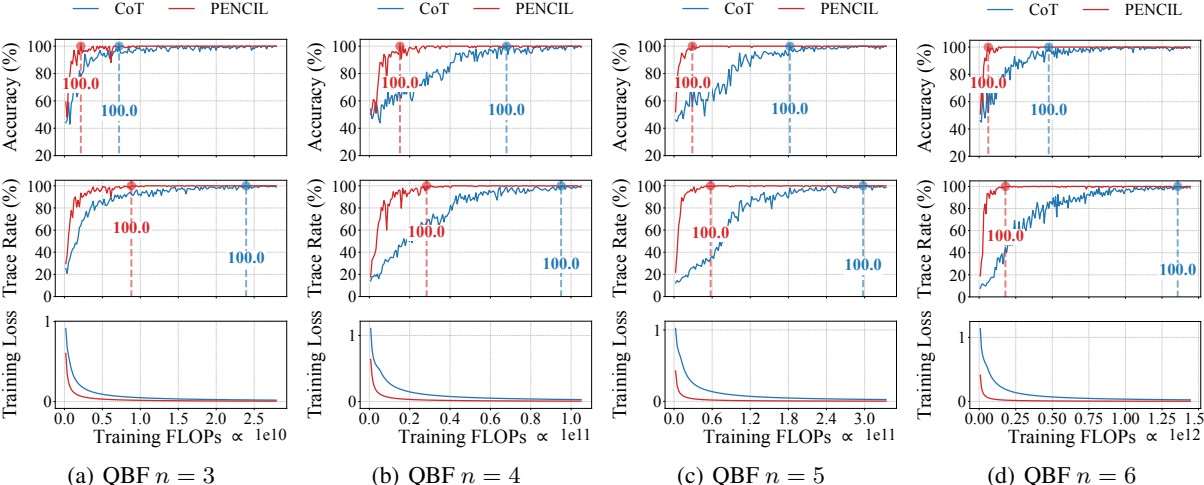

(a) QBF $n = 3$  (b) QBF $n = 4$  (c) QBF $n = 5$  (d) QBF $n = 6$

Figure 5: Comparison of convergence speed for training on the QBF problem (with $n$ ranges from 3 to 6). Circles and vertical lines indicate the first time each method reaches optimal performance. The x-axis is the FLOPs budget for self-attention calculated based on (11). See Fig. 9 in Appendix where the x-axis is the number of iterations.

**Test-Time Scalability** Figure 4 compares the test-time scalability of CoT and PENCIL given different inference time budget. For both SAT and QBF problems, PENCIL can effectively solve larger problems with increased time budget, handling up to $n = 10$ with inference time around 10s and 40s respectively while CoT struggles to scale up even when given more time. This is because the reduction rule enables PENCIL to keep the reasoning length growing polynomially rather than exponentially with problem size, significantly reducing the requirement of space during generation.

**Convergence** Figure 5 compares the convergence speed of CoT and PENCIL on the QBF problem given fixed training FLOPs budget calculated based on (11). To isolate the impact of memory constraints, which limit the expressiveness of models, we allow unlimited context window length in this experiment, enabling both methods to potentially achieve perfect performance. Since since for larger prob-

lems CoT's space consumption becomes prohibitively large and will cause out-of-memory, we only report results for $n = 3$ to 6. The results show that PENCIL can effectively save computation, and thus can consistently achieve better performance under the same compute budget and converge faster, with the gap becoming more significant as problem size increases.

## 4.2. Results on Einstein's Puzzle

Besides of the original challenging 5×5 Einstein's puzzle, we also consider two simplified variants: 3×3, 4×4. For each size of the puzzle, we generate $10,000$ training instances by randomly assigning attributes to houses and deriving valid constraints that ensure a unique solution. The accuracy is evaluated based on whether the model can successfully answer the question "who owns the Fish" on 100 unseen validation samples.

| Puzzle Size | | CoT | PENCIL |
|---|---|---|---|
| $5 \times 5$ | Accuracy (%) | 25 | 97 |
| | Trace Rate (%) | 2.97 | 78.27 |
| $4 \times 4$ | Accuracy (%) | 34 | 100 |
| | Trace Rate (%) | 8.33 | 86.52 |
| $3 \times 3$ | Accuracy (%) | 99 | 99 |
| | Trace Rate (%) | 99.37 | 99.66 |

Table 2: Comparison of performance w/o and with the reduction rule on Einstein's puzzle.

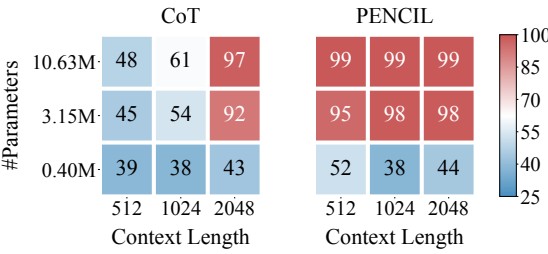

Figure 6: Effects of model size and context length on accuracy for $3\times3$ Einstein's puzzle.

**Main Results** Table 2 reports the performance with and without using the reduction rule to solve different sizes of Einstein's puzzles. Remarkably, PENCIL solves the original $5\times5$ puzzle at 97% accuracy using only 25.19M parameters (significantly smaller than GPT-2) and 2048 context length (the same as GPT-2), with average inference time per sample 42.98s. In comparison, CoT fails catastrophically on puzzles beyond $3\times3$, with accuracy dropping to 25% (i.e. close to random guessing) on $5\times5$ puzzles, despite using the same architecture and training.

**Effects of Model Size** As shown in Figure 6, PENCIL achieves consistently high accuracy with sufficient model capacity (with $\geq 3.15M$ parameters, i.e. a 4-layer transformer) even with limited context length, while CoT requires both larger models and longer context to achieve comparable performance. However, when the model size is too small, both methods fail to solve the puzzle effectively, suggesting a minimum model capacity threshold.

## 5. Universal Space-Efficient Computation

A natural theoretical question arises as to **how powerful is PENCIL on general tasks?** In this section, we answer it by theoretically showing that **PENCIL can perform universal space-efficient computation for solving any task**. More specifically, we prove that PENCIL using transformers as the base model can simulate Turing machines with optimal efficiency in both time and space. Our main result can be summarized informally as follows (see detailed statements in Theorem H.1, Appendix H):

**Theorem 5.1** (Main, Informal). For any Turing Machine, there exists a fixed finite-size transformer such that for any input, on which the computation of Turing Machine uses $T$ steps and $S$ space, PENCIL with this transformer computes the same output with $\mathcal{O}(T)$ generated tokens and using maximal context length of $\mathcal{O}(S)$.

This result is a significant improvement over CoT (Pérez et al., 2021; Merrill & Sabharwal, 2023), which showed that even though CoT can perform universal computation, it does so inefficiently; that is, it requires the context length to grow at the same rate as the time $\mathcal{O}(T)$ required to solve those problems. This is a fundamental limitation since most meaningful computations require much less memory than

time (i.e. $S \ll T$). To the best of our knowledge, PENCIL is the first approach that provably enables universal space-efficient computation for transformers. A direct implication of Theorem 5.1 is:

**Corollary 5.2.** With polynomial maximal context length (to input length), PENCIL with transformers can solve all problems in PSPACE (solvable by a Turing machine using polynomial space) while standard CoT with any poly-time next-token generator can only solve P (solvable by a Turing machine using polynomial time).[2]

It is well-known that $P \subset NP \subset PSPACE$ and widely-conjectured that $P \subsetneq PSPACE$ (a weaker assumption than the famous $P \neq NP$ hypothesis). Under this assumption, any PSPACE-complete problem (e.g., QBF (Stockmeyer & Meyer, 1973) cannot be solved by CoT using polynomial length. In contrast, PENCIL can solve these problems with polynomial maximal context length, which is a significant improvement in the computational power. Similarly, under a slightly stronger yet widely-accepted assumption called Exponential Time Hypothesis (ETH, Impagliazzo & Paturi (2001)), even SAT requires exponential length and thus cannot be solved by CoT efficiently.

**Proof Overview** The remaining of this section provides an overview and the key ideas for the proof of Theorem 5.1 (the complete proof is deferred to Appendix H). In high level, the proof contains the following three steps:

**Section 5.1:** We define a new abstract computational model called *Autoregressive Machine*, which formalizes the computation of Turing machines as a process of generating token sequences (as illustrated in Figure 7(a)), and introduces the *State Function* that transforms sequences into shorter ones (i.e. the state) representing Turing machine's configuration.

**Section 5.2:** We show that by iteratively *simulating* the next-token generation of the autoregressive machine and *summarizing* the generated tokens into its state periodically, PENCIL can reduce the maximal context length to the optimal level $\mathcal{O}(S)$ while maintaining the running time at $\mathcal{O}(T)$

---

[2]Poly-time next-token generator includes transformers, state-space models (Gu et al., 2021). Exceptions include usage of infinite-precision version of transcendental functions like exp.

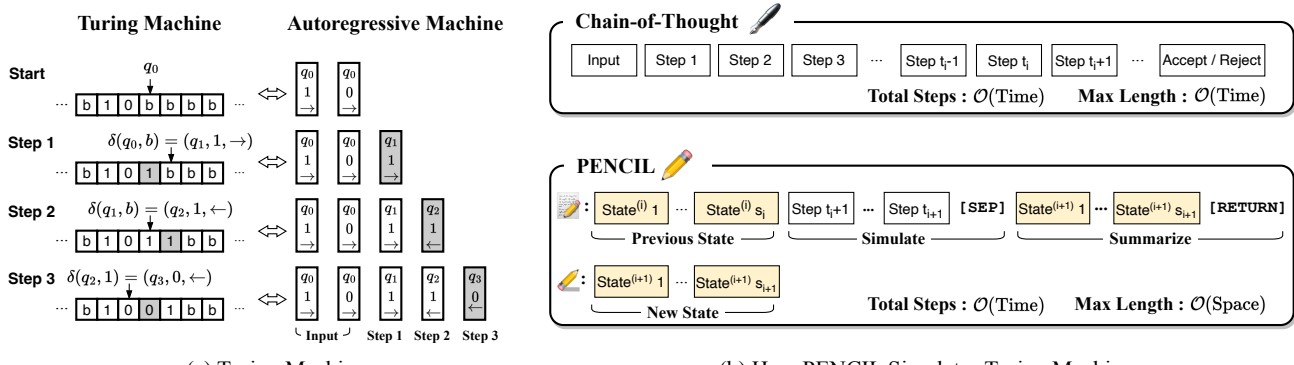

(a) Turing Machine                    (b) How PENCIL Simulates Turing Machine

Figure 7: **(a)** Autoregressive machine encodes each step of Turing machine's computation as a triplet containing the state, tape symbol, and movement direction. **(b)** PENCIL simulates Turing machine iteratively using two phases: simulating computation steps from the previous state (i.e. State$^{(i)}$), and summarizing into the new state (i.e. State$^{(i+1)}$).

(as illustrated in Figure 7(b)).

**Section 5.3:** Finally, we form a new programming language called Full-Access Sequence Processing (FASP) and use it to establish that, under specific choices of the model architecture, finite-sized transformers are expressive enough to perform this iterative generation and summarization process, thus completing the proof.

### 5.1. Autoregressive Machine and Complexity

We begin by defining *autoregressive machine* as a general purpose computation model. It subsumes Turing machine as an example and can potentially include other models.

**Definition 5.3** (Autoregressive Machine). An *autoregressive machine* is a tuple $\mathcal{M} = (\Sigma, \pi, \Sigma_{\text{accept}}, \Sigma_{\text{reject}})$, where $\Sigma$ is a finite alphabet, $\pi : \Sigma^* \to \Sigma$ is a next-token generator, and $\Sigma_{\text{accept}}, \Sigma_{\text{reject}} \subseteq \Sigma$ are accepting and rejecting tokens. For any input $x \in \Sigma^*$, $\mathcal{M}$ iteratively generates one token per step and appends it to the current sequence, with $f_\pi^k(x)$ denoting the sequence after $k$ iterations where $f_\pi(x) = (x, \pi(x))$. The machine halts when it generates a token in $\Sigma_{\text{accept}}$ or $\Sigma_{\text{reject}}$.

To achieve space efficiency in computation, we need a mechanism to compress the growing computational trace into a minimal representation that preserves only the information necessary for future steps. This is formalized by:

**Definition 5.4** (State Function). A function $s : \Sigma^* \to \Sigma^*$ is a *state* function of a autoregressive machine $\mathcal{M} = (\Sigma, \pi, \Sigma_{\text{accept}}, \Sigma_{\text{reject}})$ if **(1)** $\pi \circ s = \pi$; **(2)** for all $x, x', y \in \Sigma^*$, $s(x) = s(x') \implies s((x, y)) = s((x', y))$; **(3)** $s^2 = s$.

Note the above definition automatically implies that the future trace of the autoregressive machine $\mathcal{M}$, i.e. $\pi^k(x)$ for $k = 1, 2, \ldots$, can be uniquely determined by the state function $s$ of $\mathcal{M}$. Formally, $s \circ f_\pi^k \circ s = s \circ f_\pi^k$ and $\pi^{k+1} = \pi^{k+1} \circ s$ for any $k \geq 0$ (see Lemma I.1 in Appendix). In other words, $s$ defines a equivalent class over all possible

computational traces of $\mathcal{M}$, where the mapping $x \mapsto s(x)$ erases irrelevant information while preserving the essential information for future computation.

Correspondingly, *time complexity* $T(\mathcal{M}, x)$ can be defined as the number of steps the autoregressive machine $\mathcal{M}$ takes to halt on input $x$. *Space complexity* $S(\mathcal{M}, s, x)$ is defined as the maximal length of the states $(s \circ f_\pi)^k(x)$ for all steps $k$. This quantifies the minimal memory required to continue the computation at any point.

**Example: Turing Machine** Indeed, Turing machine can be represented as a autoregressive machine by letting each transition step produce a single token (encoding the new state, symbol, and head movement), formalized as follows (see proof in Appendix D):

**Lemma 5.5** (Turing Machine as $\mathcal{M}$). Any Turing machine TM can be represented as a autoregressive machine $\mathcal{M}_{\text{TM}}$ associated with a state function $s_{\text{TM}}$ that preserves its time and space complexity.

Specifically, the time complexity of $\mathcal{M}_{\text{TM}}$ equals the Turing machine's total step count, and the space complexity of $\mathcal{M}_{\text{TM}}$ matches the Turing machine's actual memory usage.

### 5.2. Space and Time-Efficient Simulation using PENCIL

For proving Theorem 5.1, we consider a variant of PENCIL with a simplified reduction rule $\phi'$, whichis already powerful enough for space-efficient universal simulation

$$\phi' : \quad \mathbf{T} \, [\texttt{SEP}] \, \mathbf{T'} \, [\texttt{RETURN}] \quad \Rightarrow \quad \mathbf{T'} \qquad (9)$$

This rule uses one less special token than our initial reduction rule (1) and can be expressed by it through tail recursion (8), i.e. by substituting $\mathbf{T} \leftarrow [\texttt{CALL}] \mathbf{T}$ and $\mathbf{T'} \leftarrow [\texttt{CALL}] \mathbf{T'}$ in (9). For our proof, we simply set $\mathbf{T'} = s(\mathbf{T})$, since the state contains the minimal information for future computation per definition. Therefore, the question remains as to when to trigger (9) and summarize:

**Space-Efficient but Time-Inefficient Solution** Naively, if PENCIL trigger the summarization procedure too frequently, e.g. after every new token generation, the maximal context length would be bounded by $\mathcal{O}(S)$. However, this approach would blow up the time complexity by a factor proportional to the space complexity, i.e. $\mathcal{O}(S \cdot T)$, making it highly time inefficient.

**Space and Time Efficient Solution** To achieve both optimal time and space efficiency (up to some multiplicative constant), PENCIL can keep generating new tokens to simulate running autoregressive machine, and trigger the summarization only when the length of **T** exceeds a certain threshold. In particular, we define the time (i.e. the number of tokens generated so far) to apply $i$-th summarization/reduction rule $t_i$ as the smallest integer larger than $t_{i-1}$ such that length of the state **T'** is *smaller than half* of the length of $\mathbf{T} = f_\pi^{t_i - t_{i-1}} \circ s \circ f_\pi^{t_{i-1}}(x)$, where $s \circ f_\pi^{t_{i-1}}(x)$ is the state reduced from the last iteration and $t_i - t_{i-1}$ is the number of simulated steps of autoregressive machine in the current iteration. Correspondingly, we can define the trace of PENCIL as $x^{(i)} =$

$$f_\pi^{t_i - t_{i-1}} \circ s \circ f_\pi^{t_{i-1}}(x) \,,\, \texttt{[SEP]}\,,\, s \circ f_\pi^{t_i}(x)\,,\, \texttt{[RETURN]} \quad (10)$$

where $s \circ f_\pi^{t_i}(x)$ is equivalent to $s \circ f_\pi^{t_i - t_{i-1}} \circ s \circ f_\pi^{t_{i-1}}(x)$ per Definition 5.4. In short, PENCIL compresses the current sequence into its state representation whenever its length exceeds twice the state length, enforcing space stays within $\mathcal{O}(S)$ without performing reductions so frequently that the overall time cost exceeds $\mathcal{O}(T)$. Formally:

**Proposition 5.6.** For any autoregressive machine $\mathcal{M} = (\Sigma, \pi, \Sigma_{\text{accept}}, \Sigma_{\text{reject}})$ with state function $s$, if a next-token predictor $f_{\pi_\theta}$ accurately generates the next token in (10) from the prefix for every $i$ on any input $x \in \Sigma^*$, then $\text{PENCIL}_{f_{\pi_\theta}, \phi'}$ can simulate $\mathcal{M}$ by using $\mathcal{O}(T(\mathcal{M}, x))$ steps and a maximal sequence length of $\mathcal{O}(S(\mathcal{M}, s, x))$.

Note that this result applies not just to Turing machines but to any computational model representable as an autoregressive machine with a suitable state function.

### 5.3. `FASP` for Proving Transformer Expressiveness

Now we complete our proof by demonstrating that transformers are indeed expressive enough to produce the trace described in (10) under specific architectural choices including Gated ReLU activation (Dauphin et al., 2017), positional embedding $n \mapsto n$, and average-hard casual attention (Merrill et al., 2022); details are specified in Appendix E.

**Full-Access Sequence Processing (`FASP`)** Since directly constructing a transformer is challenging, following Weiss et al. (2021); Yang & Chiang (2024), we developed a novel programming language called `FASP`, where each code in `FASP` represents a sequence-to-embedding mapping. The language defines a set of primitives or functions (termed

*Closed Operators*) for writing the program and allows defining customized operators. Depending on positional encoding and activation functions allowed to use in transformers, `FASP` has different variants supporting an increasingly rich family of primitives. A formal introduction of `FASP` and variants thereof is deferred to Appendix F and G.

`FASP` is useful for the proof because it precisely characterizes the class of functions that can be implemented by finite-size transformers with average-hard casual attention, denoted by $\mathcal{H}_{\text{TF}}$ (see formal definition in Definition E.9):

**Lemma 5.7** (Theorem F.2, Informal). `FASP` $= \mathcal{H}_{\text{TF}}$.

Thus our proof reduces to a `FASP` program that executes the space and time efficient solution mentioned in Sec 5.2.

`FASP` is more powerful than RASP (Weiss et al., 2021) because RASP cannot simulate certain hard attentions, as its selection mechanism uses boolean condition function based only query and local key. As a result, RASP cannot even retrieve the value vector of the key that is the closest to the query, which is essential in our construction (`FASP` code).

**Program** In a high level, the program implements the following three operations simultaneously (which is exactly the premise of Proposition 5.6):

**1. Summarization Trigger**: Detecting when to transition from the simulation phase to summarization phase by dynamically comparing the length of the current sequence with the its state length throughout the generation process.

**2. Simulation**: During the simulation phase, generating the next token of the autoregressive machine that simulates one step of the Turing machine.

**3. Summarization**: During the summarization phase, computing the compressed state representation of the current token sequence.

The specific program is given in Appendix H, which completes the proof for Theorem 5.1. This technique can also be used be prove other expressiveness results of transformer with CoT, e.g. Merrill & Sabharwal (2023).

## 6. Conclusion

This paper identifies a fundamental limitation of CoT where intermediate computations accumulate indefinitely in the context, and introduce PENCIL to address this. PENCIL adopts a simple reduction rule to "clean up" unneeded reasoning steps as soon as they are finalized. This mechanism effectively transforms long traces into compact representations, enabling efficient training and allowing the model to handle substantially larger problems under the same memory constraints. Extensive experiments are done to demonstrate the effectiveness of PENCIL to handle inherently challenging tasks with less computes and smaller memory.

## Impact Statement

The goal of this paper to advance our understanding of the reasoning capabilities in language models. There is no immediate negative societal impact as far as we can foresee unless language models are used for unethical purposes.

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

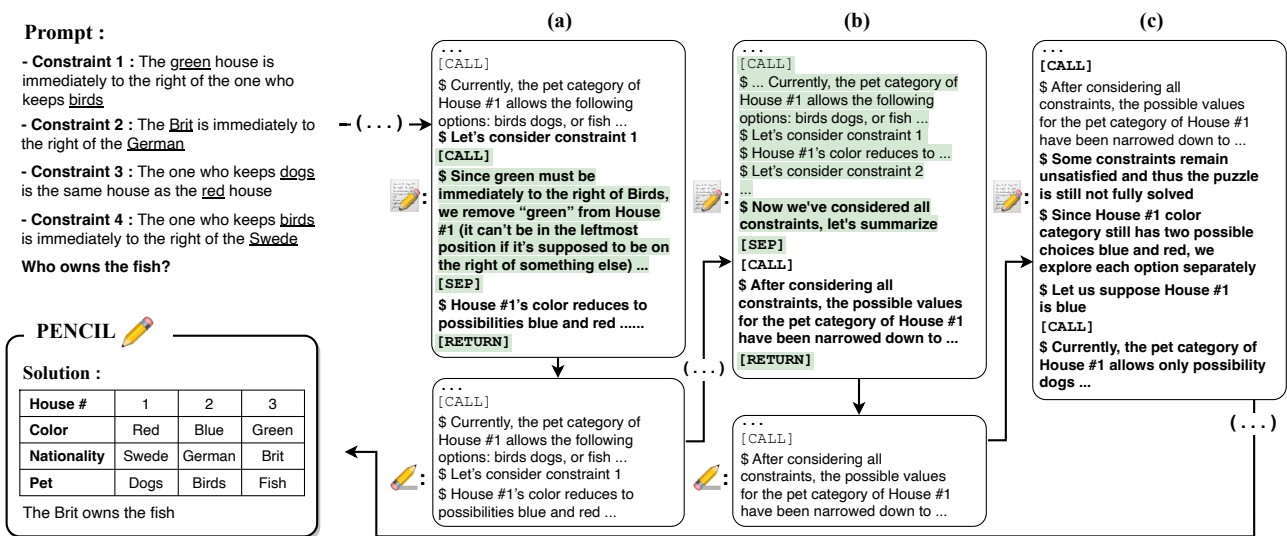

Figure 8: A simplified illustration of the algorithm for generating the thinking process for Einstein's puzzle (3×3). The puzzle requires determining attributes of each house (Color: Blue/Green/Red, Nationality: Brit/German/Swede, Pet: Birds/Dogs/Fish) given a set of constraints, with each house having unique attributes. The "..." in the arrow denotes omitted thoughts for conciseness; the "..." in the box denotes omitted thought. See the complete example in Appendix M.

## A. Related Work

**Structured Reasoning** A key distinction of scaffolded reasoning approaches stems from how space is managed during generation. At one extreme, Chain-of-Thought (Wei et al., 2022; Nye et al., 2021; Kojima et al., 2022) demonstrates that explicit intermediate steps can dramatically improve performance on complex problems, but at the expense of unbounded context growth. This limitation has motivated approaches leveraging reasoning structures such as trees and graphs (Yao et al., 2024; Long, 2023; Besta et al., 2024; Sel et al., 2023; Chen et al., 2022), adopting task decomposition strategies (Zhou et al., 2022; Drozdov et al., 2022; Khot et al., 2022) or some other prompting frameworks (Zelikman et al., 2022; Madaan et al., 2024; Suzgun & Kalai, 2024). While these methods enable more complex reasoning patterns, they require carefully crafted prompts and multiple rounds of interactions, whereas our approach achieves structured reasoning through end-to-end training.

**Test-Time Scaling** Extensive work has focused on addressing the computational bottlenecks of transformer architectures, particularly during long-context inference. One line of research explores architectural innovations through sparse and local attention patterns (Beltagy et al., 2020; Kitaev et al., 2020; Zaheer et al., 2020; Choromanski et al., 2020), while another focuses on memory optimization via KV-cache reduction (Zhang et al., 2023; Fu et al., 2024; Li et al., 2024a; Nawrot et al., 2024) and strategic context pruning (Kim et al., 2022; Jiang et al., 2023). However, these approaches still rely on next-token prediction that fundamentally treats the context window as append-only storage, leading to inherently inefficient space utilization.

**LLMs as Programming Language** Recent work has also explored intersections between programming languages and LLMs. For example, Weiss et al. (2021) proposes a language called RASP, programs in which can be encoded into and learned by transformers (Lindner et al., 2024; Friedman et al., 2024; Zhou et al., 2023). Liu et al. (2023) empirically shows that language models can be pre-trained to predict the execution traces of Python code. The reduction rule introduced in this work draws inspiration from term rewriting systems (Baader & Nipkow, 1998), a foundational means of computation in functional programming. This enables language models to explicitly emulate recursion that is otherwise hard to learn (Zhang et al., 2024), and manage space efficiently by erasing irrelevant contents in memory and focusing attention on those that are useful.

**Computational Power / Limitation of CoT** While transformers can theoretically simulate Turing machines (Pérez et al., 2021; Merrill & Sabharwal, 2023; Strobl et al., 2024; Nowak et al., 2024) with CoT, their practical computational power is fundamentally constrained by context window limitations. Particularly, we show that even with CoT, transformers with

inherent space constraints would fail to handle problems requiring extensive intermediate computation. This parallels classical space-bounded computation theory, where memory management is crucial for algorithmic capabilities (Arora & Barak, 2009; Garrison, 2024). Our approach addresses this limitation by enabling more efficient use of the context.

## B. Computational Benefits of PENCIL

To quantify the computational gap between PENCIL and CoT, consider using a standard causal-masking transformer and an ideal case where one uses KV cache for storing key and value matrices for subsequent computation, the corresponding FLOPs for self-attention (which is typically the bottleneck for very long sequences, see Kaplan et al. (2020) for a more precise method for estimating the FLOPs) required for a problem instance $x \in \Sigma^n$ is proportional to:

$$\sum_{i=1}^{r+1} \left(|x^{(i-0.5)}| + |x^{(i)}| + 1\right) \cdot \underbrace{\left(|x^{(i)}| - |x^{(i-0.5)}|\right)}_{\text{number of generated tokens}}$$
$$+ \sum_{i=1}^{r} \left(|x^{(i)} \cap x^{(i+0.5)}| + |x^{(i+0.5)}| + 1\right) \cdot \underbrace{|x^{(i+0.5)} \backslash x^{(i)}|}_{\text{length of the answer } \mathbf{A}} \tag{11}$$

where $x^{(i)} \cap x^{(i+0.5)}$ represents the shared context $\mathbf{C}$ before the [CALL] token, and $x^{(i+0.5)} \backslash x^{(i)}$ denotes the answer $\mathbf{A}$ between [SEP] and [RETURN] tokens. The first term accounts for model generation steps, while the second term captures the computation cost of reduction steps where KV cache must be recomputed for $\mathbf{A}$ after merging it back into the context (since the prefix has been changed).

## C. Additional Experimental Results

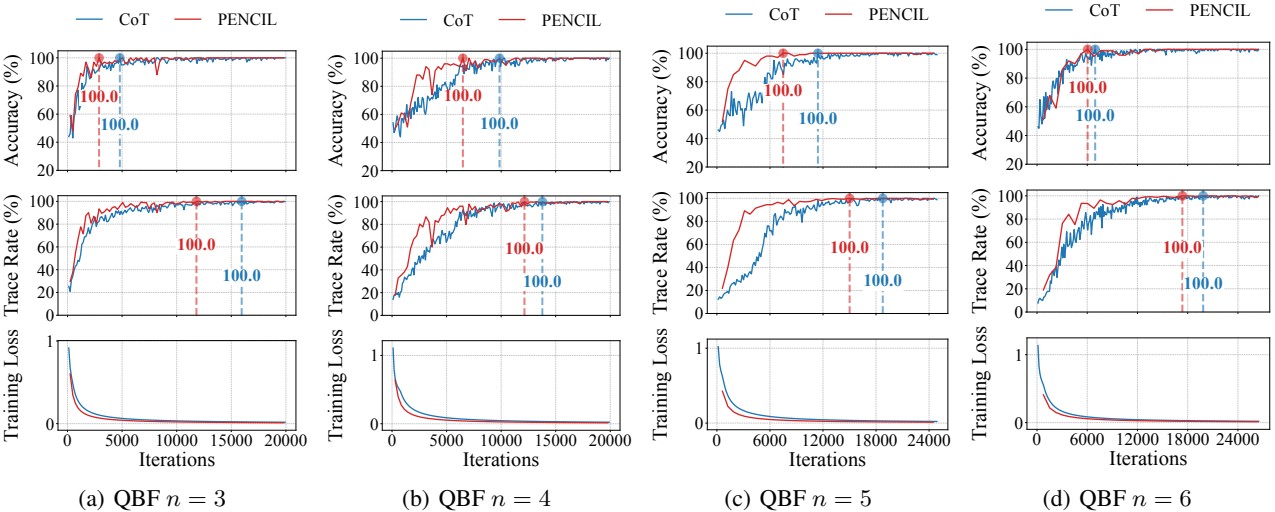

(a) QBF $n = 3$      (b) QBF $n = 4$      (c) QBF $n = 5$      (d) QBF $n = 6$

Figure 9: Comparison of convergence speed for training on the QBF problem (with $n$ ranges from 3 to 6). Circles and vertical lines indicate the first time each method reaches optimal performance. The x-axis is number of iterations.

## D. Turing Machine as Autoregressive Machine

We will restate the definition of a single-tape Turing machine, then show how each of its steps can be turned into tokens generated by an autoregressive machine $\mathcal{M}_{\text{TM}}$, associated with a state function that captures only the machine's current configuration.

### D.1. Definition of Turing Machine

A single-tape Turing machine is defined by:

**Definition D.1** (Turing Machine). A single-tape Turing machine works on a infinitely long "Tape" on both of its ends with

cells indexed by integers $\mathbb{Z}$. It is specified by a 7-tuple

$$\mathsf{TM} \;=\; (\mathcal{A}, b, Q, q_0, \delta, Q_{\text{accept}}, Q_{\text{reject}}), \tag{12}$$

where:

- $\mathcal{A}$ is a finite tape alphabet.
- $b \in \mathcal{A}$ is the designated blank symbol.
- $Q$ is a finite set of control states.
- $q_0 \in Q$ is the initial control state.
- $\delta : Q \times \mathcal{A} \;\to\; Q \times (\mathcal{A} \setminus \{b\}) \times \{-1, 0, 1\}$ is the transition function.
- $Q_{\text{accept}} \subseteq Q$ is the set of accepting states.
- $Q_{\text{reject}} \subseteq Q$ is the set of rejecting states, disjoint from $Q_{\text{accept}}$.

**Computation of Turing Machines.** At the beginning of the computation, the initial tape content $\sigma'_0 \in \mathcal{A}^{\mathbb{Z}}$ is set by the input $\sigma \in (\mathcal{A} \setminus \{b\})^*$ for the cells indexed from 0 through $|\sigma| - 1$ and the other cells contain $b$. The head of the machine is at the position $|\sigma|$ and its control state is initialized to $q_0 \in Q$. For convenience we use the $p_t$ to denote the head position at step $t$. In each time step $0 \leq t$, the machine computes $(q', a', d') = \delta(q_t, a_t)$, where $q_t$ is the control state of the Turing machine at step $t$ and $a_t = \sigma'_t[p_t]$ is the symbol on the infinite-long tape before step $t$ update $\sigma'_t$ at the Turing machine's head position $p_t$. Then the Turing machine moves its position to $p_{t+1} = p_t + d'$, change the symbol on the current tape to $a'$, and updates its new control state to $q_{t+1} = q'$. The Turing machine halts only when reaching an accept/reject state in $Q_{\text{accept}} \cup Q_{\text{reject}}$, otherwise it runs forever. We denote the output of Turing machine on input $\sigma$ by $\mathsf{TM}(\sigma)$, and we set $\mathsf{TM}(\sigma) = 1$ is the final state is in $Q_{\text{accept}}$ and $\mathsf{TM}(\sigma) = 0$ is the final control state is in $Q_{\text{reject}}$.

The computation of Turing machine is intrinsically an iterated process — applying the same transition rule $\delta$ until the halting condition is met. Such iterated models can naturally be described as an autoregressive machine (Appendix D.2). We will give the formal definition (Definition D.6) of Turing Machine as an autoregressive machine in Appendix D.2. Towards that, we will first introduce a few more useful notations.

**Definition D.2** (Configuration). The *configuration* of a Turing machine is defined as the tuple of $(q, \sigma', p) \in Q \times \mathcal{A}^{\mathbb{Z}} \times \mathbb{Z} \triangleq C$, where $q$ is its current control state, $\sigma$ is the current symbols on the tape, starting from the leftmost non-blank one to the rightmost non-blank one, and $p$ is its current head position relative to the leftmost non-blank symbol. The configuration can be thought as a snapshot or the "global" state of Turing machine, which completely determines its future computation steps.

We also extend the update rule $\delta$ to the configuration space as follows: for any configuration $c = (q, \sigma', p) \in C$, we define

$$\delta(q, \sigma', p) \triangleq \delta(q, \sigma'[p]). \tag{13}$$

**Definition D.3** (Space of Update and Update Rule). We define the space of the update as the range of transition function $\delta$, denoted by

$$\Sigma = Q \times (\mathcal{A} \setminus \{b\}) \times \{-1, 0, 1\}. \tag{14}$$

Given a configuration $c = (q, \sigma', p) \in C$ and update $x = (q', a, d) \in \Sigma$, we define the updated configuration of $c$ with $x$ as

$$\mathsf{Update}(x, c) = (\tilde{q}, \sigma', \tilde{p}) \tag{15}$$

where $\tilde{p} = p + d$, and $\tilde{\sigma}'[i] = \sigma'[i]$ for all $i \in \mathbb{Z}, i \neq p$ and $\tilde{\sigma}'[p] = a$. We denote the update function as $\mathsf{Update}(c, x)$. We also extend the notion of update function to any sequence of updates $x_{1:n} = (x_1, \ldots, x_n) \in \Sigma^n$ and and configuration $c$, where we define $\mathsf{Update}(c, x_{1:n}) = \mathsf{Update}(\mathsf{Update}(c, x_{1:n-1}), x_n)$ recursively.

Given the update rule $\delta$, the transition rule of the configuration of the Turing machine is defined as

$$g_\delta : Q \times \mathcal{A}^{\mathbb{Z}} \times \mathbb{Z} \to Q \times \mathcal{A}^{\mathbb{Z}} \times \mathbb{Z}$$
$$g_\delta(q, \sigma', p) \triangleq \mathsf{Update}(\delta(q, \sigma', p), (q, \sigma', p)).$$

Denoting configuration as at step $t$ as $c_t = (q_t, \sigma'_t, p_t) \in Q \times \mathcal{A}^{\mathbb{Z}} \times \mathbb{Z}$ with $c_0 = (q_0, \sigma'_0, |\sigma|)$, the configuration of Turing Machine at each step $t$ can be formally defined as $(q_{t+1}, \sigma'_{t+1}, p_{t+1}) \triangleq g_\delta(q_t, \sigma'_t, p_t) = g_\delta^{t+1}(c_0)$.

**Definition D.4** (Translationally Equivalent Configurations). Two Turing machine configurations $c_1 = (q_1, \sigma_1', p_1)$ and $c_2 = (q_2, \sigma_2', p_2)$ are said to be *translationally equivalent* (denoted by $c_1 \sim c_2$)if:

1. They have the same control state: $q_1 = q_2$

2. There exists an integer $k$ such that:
   - Their tape contents are equivalent up to translation: $\sigma_1'[i] = \sigma_2'[i - k]$ for all $i \in \mathbb{Z}$
   - Their head positions are equivalent up to the same translation: $p_1 = p_2 + k$

Translationally equivalent configurations will produce the same future computation behavior, differing only in the absolute positions of symbols on the tape, which is formally described by the following Lemma D.5.

We omit the proof of the following lemma, which is straightforward from the definition of Turing machine configuration and update rule.

**Lemma D.5** (Translational Equivalence of Turing Machine Configurations). For any Turing machine TM and any configurations $c_1, c_2 \in C$, if $c_1 \sim c_2$, then $\delta(c_1) = \delta(c_2)$ and that for any update $x \in \Sigma$, $\mathsf{Update}(c_1, x) \sim \mathsf{Update}(c_2, x)$. As a result, $g_\delta^k(c_1) \sim g_\delta^k(c_2)$ for any $k \in \mathbb{N}$.

### D.2. Construction of Autoregressive Machine

We now build a autoregressive machine $\mathcal{M}_{\mathsf{TM}}$ from TM by letting each Turing step correspond to the generation of a single token (new state, symbol written, head movement).

**Definition D.6** (Autoregressive Representation of a Turing Machine). Let $\mathsf{TM} = (\mathcal{A}, b, Q, q_0, \delta, Q_{\text{accept}}, Q_{\text{reject}})$ be a single-tape Turing machine. We define a autoregressive machine

$$\mathcal{M}_{\mathsf{TM}} = (\Sigma, \pi, \Sigma_{\text{accept}}, \Sigma_{\text{reject}}) \tag{16}$$

as follows:

• **Alphabet / Tokens** $\Sigma = Q \times \mathcal{A} \times \{-1, 1, 0\}$: Each token $(q, a, d) \in \Sigma$ represents a configuration that means "the machine transitions to state $q$, writes symbol $a$ on the current cell, and moves the head in direction $d$," where N indicates "no move" if desired. Furthermore, we let $\Sigma_{\text{accept}} = Q_{\text{accept}} \times (\mathcal{A} \setminus \{b\}) \times \{-1, 1, 0\}$ and $\Sigma_{\text{reject}} = Q_{\text{reject}} \times (\mathcal{A} \setminus \{b\}) \times \{-1, 1, 0\}$.

• **Next-Token Generator** $\pi : \Sigma^* \to \Sigma$: Let $c_0 = (q_0, b^{\mathbb{Z}}, 0)$ be the initial configuration of the Turing machine, we define the next-token generator $\pi$ by $\pi(\cdot) \triangleq \delta(\mathsf{Update}(c_0, \cdot))$. That is, given an input token sequence $x = (x_1, \ldots, x_n) \in \Sigma^*$, the next token is the next Turing Machine update after the configuration $c_n$ obtained by applying the updates $x_1, \ldots, x_n$ to the initial configuration $c_0$.

**Definition D.7** (Maximum and Minimum Non-Blank Positions). For any tape configuration $\sigma' \in \mathcal{A}^{\mathbb{Z}}$ with finitely many non-blank symbols and position $p$, we define:

- $\mathtt{max\_pos}(\sigma') = \max\{j \in \mathbb{Z} \mid \sigma'[j] \neq b\}$, which is the position of the rightmost non-blank symbol on the tape or head position, whichever is larger.
- $\mathtt{min\_pos}(\sigma') = \min\{j \in \mathbb{Z} \mid \sigma'[j] \neq b\}$, which is the position of the leftmost non-blank symbol on the tape or head position, whichever is smaller.

**Definition D.8** (Embedding Function from Turing Machine to Autoregressive Machine). Given a Turing machine TM and its corresponding autoregressive machine $\mathcal{M}_{\mathsf{TM}}$, we define an embedding function

$$\mathtt{embed} : C \to \Sigma^*$$

that maps a Turing machine configuration $c = (q, \sigma', p) \in C$ to a sequence of tokens in $\Sigma^*$ that represents the configuration in the autoregressive machine, where $\sigma'$ only has finitely many non-blank symbol $b$. Specifically: [3]

$$\mathtt{embed}(q, \sigma', p) = (x_1, x_2, \ldots, x_n)$$

---

[3]Note we only need to consider the case where $\mathtt{min\_pos}(\sigma') - 1 \leq p \leq \mathtt{max\_pos}(\sigma') + 1$ since Turing Machine has to write non-blank tokens on every tape cell it visits.

where $n = \texttt{max\_pos}(\sigma') - \texttt{min\_pos}(\sigma') + [\texttt{max\_pos}(\sigma') - p - 1]_+ + 1$, and each $x_i = (q_i, a_i, d_i)$ is defined as:

$$q_i = q, \quad a_i = \sigma'\left[\sum_{j=1}^{i-1} d_j + \texttt{min\_pos}(\sigma')\right], \tag{17}$$

and

$$d_i = \texttt{compute\_move}(i, p, \texttt{max\_pos}(\sigma'), \texttt{min\_pos}(\sigma')) \tag{18}$$

$$\triangleq \begin{cases} +1 & \text{if } 1 \le i \le \texttt{max\_pos}(\sigma') - \texttt{min\_pos}(\sigma') \\ +1 & \text{if } i = \texttt{max\_pos}(\sigma') - \texttt{min\_pos}(\sigma') + 1 \wedge p = \texttt{max\_pos}(\sigma') + 1 \\ 0 & \text{if } i = \texttt{max\_pos}(\sigma') - \texttt{min\_pos}(\sigma') + 1 \wedge p = \texttt{max\_pos}(\sigma') \\ -1 & \text{if } n \ge i \ge \texttt{max\_pos}(\sigma') - \texttt{min\_pos}(\sigma') + 1 \wedge p \le \texttt{max\_pos}(\sigma') - 1. \end{cases}$$

This is a standard construction used to show transformer can simulate Turing machine (Pérez et al., 2021; Merrill et al., 2022) which allows the tape contents to be reconstructed from the computation history.

From the definition of the embedding function, we can see that the embedding of a configuration $c$ of Turing Machine into a series of tokens in $\Sigma$ of Autoregressive Machine that encode the control state, the symbols on the tape, and the head position. The embedding function is translationally invariant by defintiion and we omit the proof here.

**Lemma D.9** (Embedding is Translationally Invariant). For any Turing machine TM and any configurations $c_1, c_2 \in C$, if $c_1 \sim c_2$, then $\texttt{embed}(c_1) = \texttt{embed}(c_2)$.

**Theorem D.10.** The autoregressive machine $\mathcal{M}_{\mathsf{TM}}$ defined in Definition D.6 faithfully simulates the Turing machine TM in the sense that, for any input $x \in \mathcal{A}^*$, the output of $\mathcal{M}_{\mathsf{TM}}$ on $x$ (accept or reject) is the same as the output of TM on $x$.

More specifically, the equivalence is established by the following property. Recall $c_0 = (q_0, b^{\mathbb{Z}}, 0)$, it holds that for any configuration $c = (q, \sigma', p) \in C$ and non-negative integer $k$,

$$\texttt{Update}(c_0, f_\pi^k(\texttt{embed}(c))) \sim g_\delta^k(c). \tag{19}$$

*Proof of Theorem D.10.* We will prove equation (19) by induction on $k$. First, recall that for any input sequence $x \in \Sigma^*$, $\pi(x)$ is defined as $\delta(\texttt{Update}(c_0, x))$, where $\delta$ is applied to the configuration resulting from updating the initial configuration with the sequence $x$.

**Base Case** ($k = 0$): For any configuration $c = (q, \sigma', p) \in C$, we need to show $\texttt{Update}(c_0, \texttt{embed}(c)) \sim c$.

Let's denote $m_{\min} = \texttt{min\_pos}(\sigma')$ and $m_{\max} = \texttt{max\_pos}(\sigma')$. By Definition D.8, $\texttt{embed}(c)$ is a sequence $(x_1, x_2, \ldots, x_n)$ where each token $x_i = (q_i, a_i, d_i)$ encodes the state $q$, the symbol at a specific position, and a movement direction.

When we apply this sequence to the initial configuration $c_0 = (q_0, b^{\mathbb{Z}}, 0)$, we perform the following operations:

1. The embedding first writes all non-blank symbols from the leftmost position $m_{\min}$ to the rightmost position $m_{\max}$ by moving right. 2. If needed, additional movements are generated to ensure the head ends at the correct position $p$. 3. All tokens share the same control state $q$.

After applying the entire sequence $\texttt{embed}(c)$ to $c_0$, we obtain a configuration $c' = (q', \sigma'', p')$ where:

- $q' = q$ (all tokens in the embedding share the same control state)
- $\sigma''[i] = \sigma'[i + m_{\min}]$ for all $i \in \{0, 1, \ldots, m_{\max} - m_{\min}\}$ (the tape contents are shifted)
- $\sigma''[i] = b$ for all other positions
- $p' = p - m_{\min}$ (the head position is shifted accordingly)

This defines a translational equivalence between $c'$ and $c$ with translation constant $k = -m_{\min}$, as:

1. They have the same control state: $q' = q$

2. The tape contents are translated: $\sigma''[i] = \sigma'[i + k]$ for all $i \in \mathbb{Z}$

3. The head positions are translated: $p' = p + k$

Therefore, $\mathsf{Update}(c_0, \mathtt{embed}(c)) \sim c$, which proves the base case.

**Inductive Step:** Assume equation (19) holds for some $k \geq 0$, i.e., $\mathsf{Update}(c_0, f_\pi^k(\mathtt{embed}(c))) \sim g_\delta^k(c)$.

Let $c_k' = \mathsf{Update}(c_0, f_\pi^k(\mathtt{embed}(c)))$. By the induction hypothesis, $c_k' \sim g_\delta^k(c)$.

For the $(k + 1)$-th step, we have:

$$f_\pi^{k+1}(\mathtt{embed}(c)) = (f_\pi^k(\mathtt{embed}(c)), \pi(f_\pi^k(\mathtt{embed}(c)))) = (f_\pi^k(\mathtt{embed}(c)), \delta(c_k')) \tag{20}$$

Therefore:

$$\mathsf{Update}(c_0, f_\pi^{k+1}(\mathtt{embed}(c))) = \mathsf{Update}(c_k', \delta(c_k')) = g_\delta(c_k') \tag{21}$$

By Lemma D.5, since $c_k' \sim g_\delta^k(c)$, we have:

$$g_\delta(c_k') \sim g_\delta(g_\delta^k(c)) = g_\delta^{k+1}(c) \tag{22}$$

This proves that $\mathsf{Update}(c_0, f_\pi^{k+1}(\mathtt{embed}(c))) \sim g_\delta^{k+1}(c)$, completing the induction.

Since acceptance or rejection depends only on the final state (which is preserved exactly in the relation $\sim$), $\mathcal{M}_{\mathsf{TM}}$ accepts $x$ if and only if TM accepts $x$. $\square$

### D.3. Construction of State Function $s_{\mathsf{TM}}$

Although $\mathcal{M}$ writes out every Turing step, we can define a state function $s_{\mathsf{TM}}$ that condenses the final sequence into a minimal representation of the tape.

**Definition D.11** (State Function $s_{\mathsf{TM}}$). Let $\mathcal{M}_{\mathsf{TM}} = (\Sigma, \pi, \Sigma_{\text{accept}}, \Sigma_{\text{reject}})$ be the autoregressive machine representation of Turing machine from Definition D.6. We define its state function $s_{\mathsf{TM}} : \Sigma^* \to \Sigma^*$ as the following

$$s_{\mathsf{TM}}(x) = \mathtt{embed}(\mathsf{Update}(c_0, x)), \quad \forall x \in \Sigma^*, \tag{23}$$

where $c_0 = (q_0, b^{\mathbb{Z}}, 0)$ is the initial configuration.

We claim that the constructed $s_{\mathsf{TM}}$ satisfies all three properties in Definition 5.4:

(1) **Next-Token Preservation** ($\pi \circ s_{\mathsf{TM}} = \pi$): We need to prove that for any $x \in \Sigma^*$, $\pi(s_{\mathsf{TM}}(x)) = \pi(x)$. Let $c = \mathsf{Update}(c_0, x)$ be the configuration after applying sequence $x$ to the initial configuration $c_0$. By definition of $s_{\mathsf{TM}}$, we have $s_{\mathsf{TM}}(x) = \mathtt{embed}(c)$. By the definition of $\pi$ in Definition D.6, $\pi(x) = \delta(\mathsf{Update}(c_0, x)) = \delta(c)$. Similarly, $\pi(s_{\mathsf{TM}}(x)) = \pi(\mathtt{embed}(c)) = \delta(\mathsf{Update}(c_0, \mathtt{embed}(c)))$. From Theorem D.10, Equation (19) with $k = 0$, we have $\mathsf{Update}(c_0, \mathtt{embed}(c)) \sim c$. Since $\delta$ is invariant under translational equivalence (Lemma D.5), we have $\delta(\mathsf{Update}(c_0, \mathtt{embed}(c))) = \delta(c)$. Therefore, $\pi(s_{\mathsf{TM}}(x)) = \delta(c) = \pi(x)$, which proves the property.

(2) **Future-Trace Preservation**: We need to prove that for any $x, x' \in \Sigma^*$ and $y \in \Sigma^*$, if $s_{\mathsf{TM}}(x) = s_{\mathsf{TM}}(x')$, then $s_{\mathsf{TM}}((x, y)) = s_{\mathsf{TM}}((x', y))$. Let $c = \mathsf{Update}(c_0, x)$ and $c' = \mathsf{Update}(c_0, x')$. By definition of $s_{\mathsf{TM}}$, $s_{\mathsf{TM}}(x) = \mathtt{embed}(c)$ and $s_{\mathsf{TM}}(x') = \mathtt{embed}(c')$. Since $s_{\mathsf{TM}}(x) = s_{\mathsf{TM}}(x')$, we have $\mathtt{embed}(c) = \mathtt{embed}(c')$. This implies that $c \sim c'$, as $\mathtt{embed}$ maps translationally equivalent configurations to identical sequences. For any sequence of tokens $y = (y_1, \ldots, y_m) \in \Sigma^*$, let $c_y = \mathsf{Update}(c, y)$ and $c_y' = \mathsf{Update}(c', y)$. By Lemma D.5, since $c \sim c'$, we have $c_y \sim c_y'$. Therefore, $s_{\mathsf{TM}}((x, y)) = \mathtt{embed}(c_y) = \mathtt{embed}(c_y') = s_{\mathsf{TM}}((x', y))$, which proves the property.

(3) **Idempotence** ($s_{\mathsf{TM}}^2 = s_{\mathsf{TM}}$): We need to prove that for any $x \in \Sigma^*$, $s_{\mathsf{TM}}(s_{\mathsf{TM}}(x)) = s_{\mathsf{TM}}(x)$. Let $c = \mathsf{Update}(c_0, x)$. By definition, $s_{\mathsf{TM}}(x) = \mathtt{embed}(c)$. Now, $s_{\mathsf{TM}}(s_{\mathsf{TM}}(x)) = s_{\mathsf{TM}}(\mathtt{embed}(c)) = \mathtt{embed}(\mathsf{Update}(c_0, \mathtt{embed}(c)))$. From Theorem D.10, Equation (19) with $k = 0$, we have $\mathsf{Update}(c_0, \mathtt{embed}(c)) \sim c$. Since $\mathtt{embed}$ maps translationally equivalent configurations to identical sequences by Lemma D.9, we have: $s_{\mathsf{TM}}(s_{\mathsf{TM}}(x)) = \mathtt{embed}(\mathsf{Update}(c_0, \mathtt{embed}(c))) = \mathtt{embed}(c) = s_{\mathsf{TM}}(x)$. This completes the proof.

**Proof of Lemma 5.5** (Time and Space Preservation). By construction, each Turing step of TM corresponds to precisely one token generation under the next-token predictor $\pi$ in $\mathcal{M}_{\text{TM}}$. Consequently, the total number of tokens generated before halting matches the Turing machine's step count, ensuring *time complexity* is preserved exactly. Moreover, the state function $s_{\text{TM}}$ "compresses" the entire history of tokens into a short sequence that encodes only the currently used tape cells plus head position. Since a Turing machine at most needs space proportional to the number of non-blank cells and the head's location, the maximum length $\max_k |s_{\text{TM}}(f_\pi^k(x))|$ is bounded by the tape usage of TM. This shows *space complexity* is also preserved. Hence, the constructed $\mathcal{M}_{\text{TM}}$ and $s_{\text{TM}}$ simulate TM *optimally* in both time and space.

# E. Notations and Transformer Architecture

Let $\Sigma$ be a finite vocabulary. A decoder-only transformer $\pi_\theta : \Sigma^* \to \Sigma$ with $h$ heads, $L$ layers, hidden dimension $d$, and feed-forward width $w$ is defined as follows, with all parameters and operations in $\mathbb{R}$. We will first introduce the standard transformer architecture then list all non-standard architectural modifications useful for proving the main theorem.

## E.1. Standard Notations

**Definition E.1** (Seq-to-Embedding Function Space). $\mathcal{H}(B)$ is defined as the class of all functions mapping from $\Sigma^* \to B$. We also define $\mathcal{H} = \cup_{d \in \mathbb{N}^+} \mathcal{H}(\mathbb{R}^d)$ as the union of all such classes across real spaces of all output dimensions.

**Definition E.2** (Canonical Extension to Seq-to-Seq Function). Let $A, B$ be two arbitrary sets and function $\psi : A^* \to B$ be a mapping from sequences to elements from $A$ to $B$. We define its canonical sequence-to-sequence extension $\overline{\psi} : A^* \to B^*$ as follows: for any input sequence $x = (x_1, \ldots, x_n) \in A^*$ of length $n$ to an output sequence constructed iteratively as

$$[\overline{\psi}(x)]_i = \psi(x_1, \ldots, x_i) \quad \text{for } i = 1, \ldots, n \tag{24}$$

where $x_1, \ldots, x_i$ is the prefix of length $i$ of sequence $x$.

**Definition E.3** (Probability Simplex). For any natural number $n$, the $n$-dimensional probability simplex (with $n + 1$ coordinates) is defined as

$$\Delta^n = \left\{ (x_1, x_1, \ldots, x_{n+1}) \in \mathbb{R}^{n+1} \;\middle|\; x_i \geq 0, \forall i \in [n+1] \wedge \sum_{i=1}^{n+1} x_i = 1 \right\}. \tag{25}$$

**Definition E.4** (Softmax). For any vector $x \in \mathbb{R}^m$ and temperature parameter $\beta > 0$, the *softmax* function $\text{softmax}_\beta : \mathbb{R}^m \to \Delta^{m-1}$ is defined as:

$$[\text{softmax}_\beta(x)]_i = \frac{\exp(x_i/\beta)}{\sum_{j=1}^m \exp(x_j/\beta)} \quad \text{for } i = 1, \ldots, m \tag{26}$$

where $\Delta^{m-1}$ denotes the $(m-1)$-dimensional probability simplex. When $\beta = 1$, we simply write $\text{softmax}$ without the subscript.

In our analysis we will consider the instance-wise limit when $\beta \to 0$, which leads to the Average-Hard Attention (AHA) (Merrill et al., 2022) or Hardmax:

**Definition E.5** (Hardmax). For any vector $x \in \mathbb{R}^n$, we define the *hardmax* function, $\text{softmax}_0 : \mathbb{R}^n \to \Delta^{n-1}$, as the instance-wise limit of 0 temperature limit of softmax

$$\text{softmax}_0(x) \triangleq \lim_{\beta \to 0} \text{softmax}_\beta(x). \tag{27}$$

The following lemma shows the explicit form of the hardmax function. Its proof is deferred to Appendix J.1.

**Lemma E.6** (Hardmax Explicit Form). For any vector $x \in \mathbb{R}^n$, the zero-temperature softmax function outputs a uniform distribution over the set of indices achieving the maximum value:

$$[\text{softmax}_0(x)]_i = \begin{cases} \frac{1}{|\arg\max_j x_j|} & \text{if } i \in \arg\max_j x_j \\ 0 & \text{otherwise} \end{cases} \tag{28}$$

### E.2. Transformer Layers

Below we define the modules used standard transformer architecture. For simplicity, we define each module as a parametrized function mapping from sequences to embeddings (Definition E.1), which can be extended to sequences-to-sequences by the canonical extension (Definition E.2).

1. **Token Embeddings (TE)**  A *Token Embedding* layer parametrized by parameters $\theta_{\mathrm{TE}} \in \mathbb{R}^{d \times |\Sigma|}$ is a function $\mathrm{TE}_{\theta_{\mathrm{TE}}} : \Sigma \to \mathbb{R}^d$, which maps each element $x \in \Sigma$ to a $d$-dimensional vector $\theta_{\mathrm{TE}}(x) \in \mathbb{R}^d$. We abuse the notation and extend the definition to sequences, that is, $\mathrm{TE}_{\theta_{\mathrm{TE}}} : \Sigma^* \to \mathbb{R}^d$ where $\mathrm{TE}(x_1, \ldots, x_n) = \mathrm{TE}(x_n)$ for any positive integer $n$ and $x_1, \ldots, x_n \in \Sigma$.

2. **Positional Embedding (PE)**  For $d_{\mathrm{PE}} \in \mathbb{N}$, let $\phi_{\mathrm{PE}} : \mathbb{N}^+ \to \mathbb{R}^{d_{\mathrm{PE}}}$ be a parameter-free feature function.[4] A *positional embedding* layer parametrized by parameters $\theta_{\mathrm{PE}} \in \mathbb{R}^{d \times d_{\mathrm{PE}}}$, $\mathrm{PE}_{\theta_{\mathrm{PE}}} : \mathbb{N}^+ \to \mathbb{R}^d$ maps each position $i \in \mathbb{N}^+$ to a $d$-dimensional vector $\mathrm{PE}(i) \triangleq \theta_{\mathrm{PE}} \cdot \phi_{\mathrm{PE}}(i)$. We abuse the notation and extend the definition to sequences, that is, $\mathrm{PE}_{\theta_{\mathrm{PE}}} : \Sigma^* \to \mathbb{R}^d$ where $\mathrm{PE}(x_1, \ldots, x_n) = \mathrm{PE}(n) = \theta_{\mathrm{PE}} \cdot \phi_{\mathrm{PE}}(n)$ for any positive integer $n$ and $x_1, \ldots, x_n \in \Sigma$.

3. **Attention**  A (parameter-free) *Attention* mechanism with temperature parameter $\beta \geq 0$ is a function $\mathrm{ATTN}_\beta : (\mathbb{R}^{d_{\mathrm{ATTN}}} \times \mathbb{R}^{d_{\mathrm{ATTN}}} \times \mathbb{R}^{d'_{\mathrm{ATTN}}})^* \to \mathbb{R}^{d'_{\mathrm{ATTN}}}$ for $d_{\mathrm{ATTN}}, d'_{\mathrm{ATTN}} \in \mathbb{N}^+$. For a sequence of tuples of query/key/value vectors $(q_i, k_i, v_i)_{i=1}^n \in (\mathbb{R}^{d_{\mathrm{ATTN}}} \times \mathbb{R}^{d_{\mathrm{ATTN}}} \times \mathbb{R}^{d'_{\mathrm{ATTN}}})^n$, the attention mechanism computes:

$$\alpha = \mathrm{softmax}_\beta \left( (q_n \cdot k_j)_{j=1}^n \right) \in \mathbb{R}^n, \tag{29}$$

where $\beta$ is the temperature parameter. In our analysis we will use $\beta \to 0$ (see Definition E.5) and we denote $\mathrm{ATTN}_0$ by AHA (Average-Hard Attention). The output of attention is then computed as a weighted sum of value vectors:

$$\mathrm{ATTN}_\beta((q_i, k_i, v_i)_{i=1}^n) = \sum_{j=1}^n \alpha_j v_j. \tag{30}$$

4. **Single-Head Self-Attention Layer (SA)**  A *Single-Head Self-Attention* layer parametrized by parameters $\theta_{\mathrm{SA}} = (W_Q, W_K, W_V, W_O)$ is a function $\mathrm{SA}_{\theta_{\mathrm{SA}}} : (\mathbb{R}^d)^* \to \mathbb{R}^d$. For a sequence of embeddings $(h_1, h_2, \ldots, h_n)$, the projection matrices $W_Q, W_K, W_V, W_O \in \mathbb{R}^{d_{\mathrm{SA}} \times d}$ map each embedding to query, key, and value vectors:

$$q = W_Q \cdot h_n, \quad k_j = W_K \cdot h_j, \quad v_j = W_V \cdot h_j. \tag{31}$$

For a decoder-only (causal) transformer, the last position $n$ can only attend to positions $j \leq n$. The output is computed using the attention mechanism:

$$\mathrm{SA}_{\theta_{\mathrm{SA}}}(h_1, h_2, \ldots, h_n) = W_O^\top \cdot \mathrm{ATTN}_\beta((q_i, k_i, v_i)_{i=1}^n). \tag{32}$$

5. **Multi-Head Self-Attention Layer (MHA)**  A *Multi-Head Self-Attention* layer parametrized by parameters $\theta_{\mathrm{MHA}} = (\theta_{\mathrm{SA}}^1, \theta_{\mathrm{SA}}^2, \ldots, \theta_{\mathrm{SA}}^h)$ is a function $\mathrm{MHA}_{\theta_{\mathrm{MHA}}} : (\mathbb{R}^d)^* \to \mathbb{R}^d$, where each $\theta_{\mathrm{SA}}^k = (W_Q^k, W_K^k, W_V^k, W_O^k)$ for $k = 1, \ldots, H$ parametrizes a separate single-head attention. For a sequence of embeddings $(h_1, h_2, \ldots, h_n) \in (\mathbb{R}^d)^n$, the multi-head attention output is defined as the concatenation of outputs from all individual attention heads:[5]

$$\mathrm{MHA}_{\theta_{\mathrm{MHA}}}(h_1, h_2, \ldots, h_n) = \sum_{i=1}^H \mathrm{SA}_{\theta_{\mathrm{SA}}^i}(h_1, h_2, \ldots, h_n). \tag{33}$$

This formulation allows the model to jointly attend to information from different representation subspaces.

6. **Feed-Forward (FF)**  A *Feed-Forward* layer with single activation function $\sigma : \mathbb{R}^k \to \mathbb{R}$ and parametrized by parameters $\theta_{\mathrm{FF}, \sigma} = (W_0, W_1, \ldots, W_k)$ is a function $\mathrm{FF}_{\theta_{\mathrm{FF}}}^\sigma : \mathbb{R}^d \to \mathbb{R}^d$, where $W_0, W_1, \ldots, W_k \in \mathbb{R}^{d_{\mathrm{FF}} \times d}$.

$$[\mathrm{FF}_{\theta_{\mathrm{FF}}}(h)]_i = \sum_{j=1}^{d_{\mathrm{FF}}} W_{0,ji} \cdot \sigma \left( \sum_{r=1}^d W_{1,jr} h_r, \sum_{r=1}^d W_{2,jr} h_r, \ldots, \sum_{r=1}^d W_{k,jr} h_r \right) \tag{34}$$

---

[4] A particular case which we will be interested in is the 1-dimensional feature $\phi_{\mathrm{PE}}(i) = i$.

[5] We note our definition of multi-head attention is slightly different from the most classic definition of transformer, where the dimension of each head is the model dimension divided by the number of heads. We inflate the head dimension to model dimension for each head to ensure more attention heads is always better so things are simplified.

We also extend our definition of *Feed-Forward* layer to the case with a finite set of activation functions, denoted by $\mathcal{T}_{\text{ACT}}$. In this case we create a copy of feedforward layer for each of the activation function and define $\text{FF}_{\theta_{\text{FF}}} = \sum_{\sigma \in \mathcal{T}_{\text{ACT}}} \text{FF}^{\sigma}_{\theta_{\text{FF},\sigma}}$ with $\theta_{\text{FF}} = (\theta_{\text{FF},\sigma})_{\sigma \in \mathcal{T}_{\text{ACT}}}$ where $\theta_{\text{FF},\sigma}$ is the parameter of the feedforward layer with activation function $\sigma$. Similar to token embedding, we extend the definition to sequences, that is, $\text{FF}_{\theta_{\text{FF}}} : \mathbb{R}^{d^*} \to \mathbb{R}^d$ where $\text{FF}_{\theta_{\text{FF}}}(h_1, \ldots, h_n) = \text{FF}_{\theta_{\text{FF}}}(h_n)$ for any positive integer $n$ and $h_1, \ldots, h_n \in \mathbb{R}^d$.

7. **Identity and Residual Connections**    For any embedding dimension $d \in \mathbb{N}^+$, we will use the identity function $\text{id}_d : (\mathbb{R}^d)^* \to \mathbb{R}^d$ to represent the residual connections in transformer layers. Similar to token embedding, we extend the definition to sequences, that is, $\text{id}_d : \mathbb{R}^{d^*} \to \mathbb{R}^d$ where $\text{id}_d(h_1, \ldots, h_n) = h_n$ for any positive integer $n$ and $h_1, \ldots, h_n \in \mathbb{R}^d$.

8. **Linear Projection Layer**    A *Linear Projection Layer* parametrized by parameters $\theta_{\text{PROJ}} \in \mathbb{R}^{d_{\text{PROJ}} \times d}$ is a function $\text{PROJ}_{\theta_{\text{PROJ}}} : (\mathbb{R}^d)^* \to \mathbb{R}^{d_{\text{PROJ}}}$. For a sequence of embeddings $(h_1, h_2, \ldots, h_n)$, the linear projection layer applies a linear transformation to the last embedding in the sequence:

$$\text{PROJ}_{\theta_{\text{PROJ}}}(h_1, h_2, \ldots, h_n) = \theta_{\text{PROJ}} \cdot h_n. \tag{35}$$

9. **Decoding Layer**    A *(Greedy) Decoding Layer* is a special projection layer followed by argmax, parametrized by $\theta_{\text{DEC}} \in \mathbb{R}^{|\Sigma| \times d}$, where $d_{\text{PROJ}} = |\Sigma|$. For a sequence of embeddings $(h_1, h_2, \ldots, h_n)$, the decoding layer first applies a linear projection to the last embedding:

$$\text{DEC}_{\theta_{\text{DEC}}}(h_1, h_2, \ldots, h_n) = \theta_{\text{DEC}} \cdot h_n \in \mathbb{R}^{|\Sigma|}. \tag{36}$$

Then, the next token is deterministically selected by taking the argmax:

$$x_{n+1} = \arg \max_{x \in \Sigma} [\text{DEC}_{\theta_{\text{DEC}}}(h_1, h_2, \ldots, h_n)]_x. \tag{37}$$

Here we assume the argmax is well-defined, i.e., the maximum is unique.

**Definition E.7** (Transformer Layer).  A single transformer layer $\mathcal{H}_{\theta_{\text{MHA}}, \theta_{\text{FF}}} : (\mathbb{R}^d)^* \to \mathbb{R}^d$ with residual connection and set of activation fucntions $\mathcal{T}_{\text{ACT}}$, and average-hard attention is defined as:

$$\text{TF}_{\theta_{\text{MHA}}, \theta_{\text{FF}}} = (\text{FF}_{\theta_{\text{FF}}} + \text{id}_d) \circ (\overline{\text{MHA}_{\theta_{\text{MHA}}}} + \overline{\text{id}_d}) \tag{38}$$

The sequence-to-sequence version of the layer is defined as:

$$\overline{\text{TF}_{\theta_{\text{MHA}}, \theta_{\text{FF}}}} = (\overline{\text{FF}_{\theta_{\text{FF}}}} + \overline{\text{id}_d}) \circ (\overline{\text{MHA}_{\theta_{\text{MHA}}}} + \overline{\text{id}_d}) \tag{39}$$

**Definition E.8** (Transformer as Next-Token Generator).  Let $\theta = (\theta_{\text{TE}}, (\theta^{(\ell)}_{\text{MHA}})^L_{\ell=1}, (\theta^{(\ell)}_{\text{FF}})^L_{\ell=1}, \theta_{\text{DEC}})$ be the parameters of the transformer. The end-to-end next token generator $\pi_\theta : \Sigma^* \to \Sigma$ is defined as:

$$\pi_\theta = \text{DEC}_{\theta_{\text{DEC}}} \circ \left( \bigcirc^L_{\ell=1} \overline{\text{TF}_{\theta^{(\ell)}_{\text{MHA}}, \theta^{(\ell)}_{\text{FF}}}} \right) \circ \left( \overline{\text{PE}_{\theta_{\text{PE}}}} + \overline{\text{TE}_{\theta_{\text{TE}}}} \right), \tag{40}$$

where $\bigcirc^L_{\ell=1} f_l$ means the composition of functions $f_L \circ f_{L-1} \circ \cdots \circ f_1$.

### E.3. Function Classes Implementable by Transformers

To understand what kind of next-token generator can be implemented by a transformer in the sense of Definition E.8, it is very useful to understand the class of seq-to-embedding functions implementable by transformers. After all, the next-token generator is a sequence-to-embedding function followed by a decoding layer. We define the class of seq-to-embedding functions implementable by transformers as follows:

**Definition E.9** (Class of Embedding Functions Implementable by Transformers).  For any positive integers $d_{\text{PROJ}}$, we define $\mathcal{H}_{\text{TF}[\phi_{\text{PE}}; \mathcal{T}_{\text{ACT}}]}(d_{\text{PROJ}})$ as the class of seq-to-embedding functions $\psi : \Sigma^* \to \mathbb{R}^{d_{\text{PROJ}}}$ that can be computed by fixed-size transformers (independent of the length of input sequence). That is, there exist positive integers $d, d_{\text{FF}}, d_{\text{SA}}, H, L$, and $\theta = (\theta_{\text{TE}}, (\theta^{(\ell)}_{\text{MHA}})^L_{\ell=1}, (\theta^{(\ell)}_{\text{FF}})^L_{\ell=1}, \theta_{\text{DEC}})$ with matching dimensions such that:

$$\psi = \text{PROJ}_{\theta_{\text{PROJ}}} \circ \left( \bigcirc^L_{\ell=1} \overline{\text{TF}_{\theta^{(\ell)}_{\text{MHA}}, \theta^{(\ell)}_{\text{FF}}}} \right) \circ \left( \overline{\text{PE}} + \overline{\text{TE}_{\theta_{\text{TE}}}} \right) \tag{41}$$

Finally we define $\mathcal{H}_{\text{TF}[\phi_{\text{PE}}; \mathcal{T}_{\text{ACT}}]} = \cup_{d_{\text{PROJ}} \in \mathbb{N}^+} \mathcal{H}_{\text{TF}[\phi_{\text{PE}}; \mathcal{T}_{\text{ACT}}]}(d_{\text{PROJ}})$.

Finally, we define the function class that can be implemented by a token embedding layer, a positional embedding layer, a single-head attention layer, a multi-head self-attention layer, a feed-forward layer, a linear projection layer, a transformer layer, and a decoding layer, with all possible input embedding dimensions and output embedding dimensions, as $\mathcal{H}_{\text{TE}}, \mathcal{H}_{\text{PE}}, \mathcal{T}_{\text{SA}}, \mathcal{T}_{\text{MHA}}, \mathcal{T}_{\text{FF}}, \mathcal{T}_{\text{PROJ}}, \mathcal{T}_{\text{TF}}, \mathcal{T}_{\text{DEC}}$ respectively.

For simplicity, we do not assume rounding like standard floating point arithmetics like (Li et al., 2024b) and forward pass of transformer is done in full precision. However, because we only use average-hard attention and do not use layernorm, all the intermediate computation in the forward at position $n$ only requires $O(\log(n))$ precision. More concretely, all the intermediate steps can be written exactly as ratio of two integers bounded by a polynomial of $n$ independent of the input (but depending on Turing Machine). In the later parts of paper, we will still use $\mathbb{R}^d$ to be the codomain of the seq-to-embedding funcitons, but it can easily be replaced by $\mathbb{Q}^d$ with polynomial upper bound (in terms of input length) for the denominators and numerators.

### E.4. Closed Operators

**Definition E.10** (Average Hard Attention Operator). For any $d, d' \in \mathbb{N}^+$, we define the *average-hard attention* operator $\texttt{aha} : \mathcal{H}(\mathbb{R}^d) \times \mathcal{H}(\mathbb{R}^d) \times \mathcal{H}(\mathbb{R}^{d'}) \to \mathcal{H}(\mathbb{R}^{d'})$ as the operator induced by average-hard attention AHA. Formally, for any three seq-to-embedding functions $q, k \in \mathcal{H}(\mathbb{R}^d)$ and $v \in \mathcal{H}(\mathbb{R}^{d'})$, and any integer $n$, and any sequence $x \in \Sigma^n$, we define

$$\texttt{aha}(q, k, v)(x) = \text{AHA}(\overline{(q, k, v)}(x)) = \sum_{j \leq n} \alpha_j v(x_{1:j}) \tag{42}$$

where $\alpha = \text{softmax}_0\left((q(x) \cdot k(x_{1:j}))_{j=1}^n\right)$ are the attention weights using the hardmax function from Definition E.5. $\overline{(q, k, v)}(x)$ is a sequence of length $n$ where the $i$th term is $(q(x_{1:i}), k(x_{1:i}), v(x_{1:i}))$ Specifically, $\alpha_j$ is non-zero only for positions $j$ that maximize the dot product $q(x) \cdot k(x_{1:j})$, with equal weight assigned to all such maximizing positions.

**Definition E.11** (Local Operator). We say an operator $\omega : \mathcal{H}(\mathbb{R}^{d_1}) \times \mathcal{H}(\mathbb{R}^{d_2}) \times \ldots \times \mathcal{H}(\mathbb{R}^{d_k}) \to \mathcal{H}(\mathbb{R}^{d'})$ is *local* for some positive integers $k, d'$, and $\{d_i\}_{i=1}^k$ iff there exists a function $\phi_\omega : \mathbb{R}^{\sum_{i=1}^k d_i} \to \mathbb{R}^{d'}$ such that for any $\psi_i \in \mathcal{H}(\mathbb{R}^{d_i})$, $\omega(\psi_1, \ldots, \psi_k) = \phi_\omega \circ [\psi_1, \ldots, \psi_k]$.

**Definition E.12** (Direct Sum and Concatenation). We use $[u, v]$ denotes the concatenation of vectors $u$ and $v$. For two real vector spaces $\mathbb{R}^{d_1}$ and $\mathbb{R}^{d_2}$, their direct sum $\mathbb{R}^{d_1} \oplus \mathbb{R}^{d_2}$ is defined as the set of the concatenation of their individual elements:

$$\mathbb{R}^{d_1} \oplus \mathbb{R}^{d_2} = \{[v_1, v_2] \mid v_1 \in \mathbb{R}^{d_1}, v_2 \in \mathbb{R}^{d_2}\} = \mathbb{R}^{d_1 + d_2}. \tag{43}$$

For two functions $\phi_1 : \mathbb{R}^{d_1} \to \mathbb{R}^{d'_1}$ and $\phi_2 : \mathbb{R}^{d_2} \to \mathbb{R}^{d'_2}$, their direct sum $\phi_1 \oplus \phi_2 : \mathbb{R}^{d_1} \oplus \mathbb{R}^{d_2} \to \mathbb{R}^{d'_1} \oplus \mathbb{R}^{d'_2}$ is defined as:

$$(\phi_1 \oplus \phi_2)([v_1, v_2]) = [\phi_1(v_1), \phi_2(v_2)] \quad \text{for all } v_1 \in \mathbb{R}^{d_1}, v_2 \in \mathbb{R}^{d_2}. \tag{44}$$

For two function spaces $\mathcal{T}_1 = \{f : \mathbb{R}^{d_1} \to \mathbb{R}^{d'_1}\}$ and $\mathcal{T}_2 = \{g : \mathbb{R}^{d_2} \to \mathbb{R}^{d'_2}\}$, their direct sum $\mathcal{T}_1 \oplus \mathcal{T}_2$ is defined as:

$$\mathcal{T}_1 \oplus \mathcal{T}_2 = \{f \oplus g \mid f \in \mathcal{T}_1, g \in \mathcal{T}_2\} \tag{45}$$

where each element is a function from $\mathbb{R}^{d_1} \oplus \mathbb{R}^{d_2}$ to $\mathbb{R}^{d'_1} \oplus \mathbb{R}^{d'_2}$.

For two seq-to-embedding functions $\psi_1 \in \mathcal{H}(\mathbb{R}^{d_1})$ and $\psi_2 \in \mathcal{H}(\mathbb{R}^{d_2})$, their concatenation $[\psi_1, \psi_2] : \Sigma^* \to \mathbb{R}^{d_1 + d_2}$ is defined as:

$$[\psi_1, \psi_2](x) = [\psi_1(x), \psi_2(x)] \quad \text{for all } x \in \Sigma^*. \tag{46}$$

**Definition E.13** (Closed Operators). A *closed operator* is a mapping $\omega : \mathcal{H}(\mathbb{R}^{d_1}) \times \mathcal{H}(\mathbb{R}^{d_2}) \times \ldots \times \mathcal{H}(\mathbb{R}^{d_k}) \to \mathcal{H}(\mathbb{R}^{d'})$, for some positive integer $k$, that is $\omega(\psi_1, \ldots, \psi_k) \in \mathcal{H}_{\text{TF}[\phi_{\text{PE}}; \mathcal{T}_{\text{ACT}}]}$ for any $\psi_1, \ldots, \psi_k \in \mathcal{H}_{\text{TF}[\phi_{\text{PE}}; \mathcal{T}_{\text{ACT}}]}$.

## F. Full-Access Sequence Processing

Following the footsteps of (Weiss et al., 2021; Yang & Chiang, 2024), we define a more powerful version of RASP, called *Full-Access Sequence Processing* language. Our language is poewrful than `RASP` and `C-RASP` in the following two senses: (1). `FASP` support sequence of vectors as opposed to sequence of numbers only. (2). We allow simulating standard hard attention mechanism, while `RASP` must decide whether to "select" (attend) some entry only based on the indivual pair of key and query, but not the comparison between the rest pairs. `FASP` is provably equivalent to the expressiveness of transformers with average-hard attention and casual masking.

**Definition F.1** (FASP). Let $\phi_{\text{PE}} : \mathbb{N}^+ \to \mathbb{R}^{\text{PE}}$ be a feature function for positional embedding and $\mathcal{T}_{\text{ACT}}$ be the class of activation functions. We define the $\text{FASP}[\phi_{\text{PE}}; \mathcal{T}_{\text{ACT}}]$ program as the process of defining a sequence of token-sequence-to-embedding $\psi_1, \ldots, \psi_n \in \mathcal{H}$ using $\text{FASP}[\phi_{\text{PE}}; \mathcal{T}_{\text{ACT}}]$ operators. The program is defined as follows: at each step $t \in [n]$, the program maintains a set of defineable seq-to-embedding functions $\mathcal{S}_t$, and defines a new function by concatenation functions in $\mathcal{S}_t$, or applying local operators (corresponding to MLP), or non-local operators (corresponding to average-hard attention) to some function in $\mathcal{S}_t$. Finally we add the newly defined function to $\mathcal{S}_t$, which yields $\mathcal{S}_{t+1}$. In detail, we define the defineable functions at step $t \in [n]$:

$$\mathcal{S}_t \triangleq \mathcal{H}_{\text{TE}} \cup \{\phi_{\text{PE}}\} \cup \{\psi_i \mid 1 \leq i \leq t - 1\}. \tag{47}$$

Note this also implies that $\mathcal{S}_t = \mathcal{S}_{t-1} \cup \{\psi_t\}$.

$\psi_t$ at step $t$ has to be defined by applying one of the following four *primitive* operators on already-defiend functions from $\mathcal{S}_t$:

1. **Concatenation**: $\psi_t = [\psi, \psi']$, where $\psi, \psi' \in \mathcal{S}_t$. This operator concatenates the output embedding vector of two functions into a longer vector.

2. **Average-Hard Attention**: $\psi_t = \text{aha}(\psi, \psi', \psi'')$, where $\psi, \psi', \psi'' \in \mathcal{S}_t$ and $\psi, \psi'$ have the same output dimension. This implements average-hard attention with query $\psi$, key $\psi'$, and value $\psi''$.

3. **Linear Projection**: $\psi_t = \phi \circ \psi$, where $\psi \in \mathcal{S}_t$ and $\phi$ is a linear transformation with arbitrary output dimension.

4. **Nonlinear Activation**: $\psi_t = \phi \circ \psi$, where $\phi : \mathbb{R}^k \to \mathbb{R} \in \mathcal{T}_{\text{ACT}}, \psi \in \mathcal{S}_t \cap \mathcal{H}(k)$ for some positive integer $k$. [6]

We denote the set of all such final outputed seq-to-embedding functions defineable by FASP as some $\psi_i$ with position embedding $\phi_{\text{PE}}$ and activation functions $\mathcal{T}_{\text{ACT}}$ as $\text{FASP}[\phi_{\text{PE}}; \mathcal{T}_{\text{ACT}}]$.

In particular, when we want to use FASP to define or represent a function mapping from a sequence of tokens $\Sigma^*$ to a single token in $\Sigma$, we could simply require to the embedding of dimension of $|\Sigma|$, assume an implicit order over $\Sigma$ so the index maps to a token in $\Sigma$ and return the index (token) with the largest value in the last function defined.[7]

**Theorem F.2.** For any positional encoding feature function $\phi_{\text{PE}}$ and activation function class $\mathcal{T}_{\text{ACT}}$, it holds that $\text{FASP}[\phi_{\text{PE}}; \mathcal{T}_{\text{ACT}}] = \mathcal{H}_{\text{TF}[\phi_{\text{PE}}; \mathcal{T}_{\text{ACT}}]}$.

The high-level idea towards the proof of Theorem F.2 is to show that the four operators that generates new functions in $\text{FASP}[\phi_{\text{PE}}; \mathcal{T}_{\text{ACT}}]$ are also *closed* under the class of embedding functions that can be implemented by transformers, namely $\mathcal{H}_{\text{TF}[\phi_{\text{PE}}; \mathcal{T}_{\text{ACT}}]}$. We defer its full proof to Appendix J.2 and only sketch the high-level idea via providing some key lemmas below.

As the base case, i.e., when the number of transformer layers is $0$, we know that the class of seq-to-embedding functions is simply the class of embedding functions, including both token embedding and positional embedding.

**Lemma F.3.** The function classes corresponding to token embedding and positional embeddings are subsets of $\mathcal{H}_{\text{TF}[\phi_{\text{PE}}; \mathcal{T}_{\text{ACT}}]}$. Formally, $\mathcal{H}_{\text{PE}}, \mathcal{H}_{\text{TE}} \subseteq \mathcal{H}_{\text{TF}[\phi_{\text{PE}}; \mathcal{T}_{\text{ACT}}]}$.

Next we will also identify two main types of closed operators: concatenation and transformer layer, where the latter includes local operators by feedforward networks with non-linear activation functions and non-local operators by average-hard attention.

**Lemma F.4** (Closedness Under Concatenation, Direct Sum, and Sum). We have the following closedness property for seq-to-embedding functions under concatenation, direct sum, and sum:

1. For any set $\mathcal{H} \in \{\mathcal{H}_{\text{PE}}, \mathcal{H}_{\text{TE}}\}$, for any $\psi_1, \psi_2 \in \mathcal{H}$, their concatenation $[\psi_1, \psi_2] \in \mathcal{H}$.

2. For any $d, d' \in \mathbb{N}$, let $0_{d,d'} : \mathbb{R}^d \to \mathbb{R}^{d'}$ be the zero function (mapping every input to $0 \in \mathbb{R}^{d'}$). For any set $\mathcal{T} \in \{\mathcal{T}_{\text{SA}}, \mathcal{T}_{\text{MHA}}, \mathcal{T}_{\text{FF}}, \mathcal{T}_{\text{PROJ}}\}$, (a). $0_{d,d'} \in \mathcal{T}$ and (b).for any $\phi \in \mathcal{T}$, the direct sum $\phi \oplus 0_{d,d'} \in \mathcal{T}$.

3. For any set $\mathcal{T} \in \{\mathcal{T}_{\text{MHA}}, \mathcal{T}_{\text{FF}}, \mathcal{T}_{\text{PROJ}}\}, \mathcal{T} = \mathcal{T} + \mathcal{T} \triangleq \{\phi_1 + \phi_2 \mid \phi_1, \phi_2 \in \mathcal{T}\}$. Moreover, $\mathcal{T}_{\text{MHA}}$ is the sum closure of $\mathcal{T}_{\text{SA}}$, that is, $\mathcal{T}_{\text{MHA}} = \{\sum_{j=1}^h \phi_j \mid \phi_j \in \mathcal{T}_{\text{SA}}, h \in \mathbb{N}^+\}$.

---

[6] We allow multi-variable activation functions like Gated ReLU (ReGLU), $x, y \mapsto x[y]_+$.

[7] We could assume an arbitrary order to break ties, but we omit this for simplicity. In our examples we always ensure the argmax is unique.

4. For any set $\mathcal{T} \in \{\mathcal{T}_{\mathrm{MHA}}, \mathcal{T}_{\mathrm{FF}}, \mathcal{T}_{\mathrm{PROJ}}, \mathcal{T}_{\mathrm{TF}}, \{\mathrm{id}_d \mid d \in \mathbb{N}\}\}$, for any $\phi_1, \phi_2 \in \mathcal{T}$, their direct sum $\phi_1 \oplus \phi_2 \in \mathcal{T}$.

**Lemma F.5.** The concatenation operator is closed over $\mathcal{H}_{\mathrm{TF}[\phi_{\mathrm{PE}};\mathcal{T}_{\mathrm{ACT}}]}$, that is, $[\cdot, \cdot] : \mathcal{H}^2_{\mathrm{TF}[\phi_{\mathrm{PE}};\mathcal{T}_{\mathrm{ACT}}]} \to \mathcal{H}_{\mathrm{TF}[\phi_{\mathrm{PE}};\mathcal{T}_{\mathrm{ACT}}]}$.

**Lemma F.6** (Local Closed Operators). A local operator $\omega$ is closed over $\mathcal{H}_{\mathrm{TF}[\phi_{\mathrm{PE}};\mathcal{T}_{\mathrm{ACT}}]}$, that is, $\omega : \mathcal{H}_{\mathrm{TF}[\phi_{\mathrm{PE}};\mathcal{T}_{\mathrm{ACT}}]}(d_1) \times \mathcal{H}_{\mathrm{TF}[\phi_{\mathrm{PE}};\mathcal{T}_{\mathrm{ACT}}]}(d_2) \times \ldots \times \mathcal{H}_{\mathrm{TF}[\phi_{\mathrm{PE}};\mathcal{T}_{\mathrm{ACT}}]}(d_k) \to \mathcal{H}_{\mathrm{TF}[\phi_{\mathrm{PE}};\mathcal{T}_{\mathrm{ACT}}]}(\mathbb{R}^{d'})$ for some positive integers $k, d'$, and $\{d_i\}_{i=1}^k$, if its equivalent local function $\phi_\omega$ can be implemented by a multi-layer network with activation functions in $\mathcal{T}_{\mathrm{ACT}}$

Besides the local operators induced feedforward networks, we also have the following non-local closed operator induced by attention (Lemma F.12).

**Lemma F.7** (AHA is a Closed Operator). Average-hard attention is a closed operator over $\mathcal{H}_{\mathrm{TF}[\phi_{\mathrm{PE}};\mathcal{T}_{\mathrm{ACT}}]}$, that is, for any $q, k \in \mathcal{H}_{\mathrm{TF}[\phi_{\mathrm{PE}};\mathcal{T}_{\mathrm{ACT}}]}(\mathbb{R}^d)$ and $v \in \mathcal{H}_{\mathrm{TF}[\phi_{\mathrm{PE}};\mathcal{T}_{\mathrm{ACT}}]}(\mathbb{R}^{d'})$, we have $\mathrm{aha}(q, k, v) \in \mathcal{H}_{\mathrm{TF}[\phi_{\mathrm{PE}};\mathcal{T}_{\mathrm{ACT}}]}(\mathbb{R}^{d'})$.

The proof of Lemma F.12 is similar to that of Lemma F.11, which uses the definition of $\mathcal{H}_{\mathrm{TF}[\phi_{\mathrm{PE}};\mathcal{T}_{\mathrm{ACT}}]}$ and the closedness property of concatenation (Lemma F.10). The proof is straightforward and omitted.

### F.1. Custom Operators in FASP

To further improve the convenience of coding in FASP and proving certain functions can be expressed by constant depth transformers uniformly, we introduce an extension to FASP, which instead of allowing the four primitive operators, we also allow other closed operators Definition E.13. Below we are going to introduce a specific grammar that allows us to build new custom operators that are commonly used in transformer models. These operators are not primitive operators in FASP, but can be easily implemented by composition of the primitive operators defined in Definition F.1. Those custom operators are closed under the class of embedding functions that can be implemented by transformers, namely $\mathcal{H}_{\mathrm{TF}[\phi_{\mathrm{PE}};\mathcal{T}_{\mathrm{ACT}}]}$, since each primitive operator is closed.

**Definition F.8** (Custom Closed Operators). Let $\omega : \mathcal{H}(\mathbb{R}^{d_1}) \times \mathcal{H}(\mathbb{R}^{d_2}) \times \ldots \times \mathcal{H}(\mathbb{R}^{d_k}) \to \mathcal{H}(\mathbb{R}^{d'})$ be an operator and let its input be $\tilde{\psi}_1, \ldots, \tilde{\psi}_k$. We say $\omega$ is a *custom closed operator* if it can be expressed as a composition of primitive operators in FASPand other previously defined custom closed operators[8].

In detail, the definition of $\omega$ via composition is similar to FASPand is as follows:

- at each step $t \in [n]$, the program maintains a set of defineable seq-to-embedding functions $\mathcal{S}_t \triangleq \mathcal{H}_{\mathrm{TE}} \cup \{\phi_{\mathrm{PE}}\} \cup \{\psi_i \mid 1 \le i \le t-1\} \cup \{\tilde{\psi}_j \mid 1 \le j \le k\}$.
- at each step $t \in [n]$, the program defines a new function $\psi_t$ by applying either one of the four primitive operators in FASP, or a previously defined custom closed operator to some functions in $\mathcal{S}_t$.
- the operator $\omega$ returns the last function defined in the program, i.e., $\psi_n$, on input of $\tilde{\psi}_1, \ldots, \tilde{\psi}_k$.

When the definition via composition is short, we also write them in an inline format without explicitly naming the intermediate $\psi_i$.

**Example F.9** (Addition). We define the *addition* operator $\mathrm{add} : \mathcal{H}(\mathbb{R}) \times \mathcal{H}(\mathbb{R}) \to \mathcal{H}(\mathbb{R})$ as the operator that takes two seq-to-embedding functions $\psi, \psi' \in \mathcal{H}(\mathbb{R}^d)$ and outputs their element-wise sum:

$$\mathrm{add}(\psi, \psi')(x) = \psi(x) + \psi'(x) \quad \text{for all } x \in \Sigma^*. \tag{48}$$

**Lemma F.10.** The concatenation operator is closed over $\mathcal{H}_{\mathrm{TF}[\phi_{\mathrm{PE}};\mathcal{T}_{\mathrm{ACT}}]}$, that is, $[\cdot, \cdot] : \mathcal{H}^2_{\mathrm{TF}[\phi_{\mathrm{PE}};\mathcal{T}_{\mathrm{ACT}}]} \to \mathcal{H}_{\mathrm{TF}[\phi_{\mathrm{PE}};\mathcal{T}_{\mathrm{ACT}}]}$.

**Lemma F.11** (Local Closed Operators). A local operator $\omega$ is closed over $\mathcal{H}_{\mathrm{TF}[\phi_{\mathrm{PE}};\mathcal{T}_{\mathrm{ACT}}]}$, that is, $\omega : \mathcal{H}_{\mathrm{TF}[\phi_{\mathrm{PE}};\mathcal{T}_{\mathrm{ACT}}]}(d_1) \times \mathcal{H}_{\mathrm{TF}[\phi_{\mathrm{PE}};\mathcal{T}_{\mathrm{ACT}}]}(d_2) \times \ldots \times \mathcal{H}_{\mathrm{TF}[\phi_{\mathrm{PE}};\mathcal{T}_{\mathrm{ACT}}]}(d_k) \to \mathcal{H}_{\mathrm{TF}[\phi_{\mathrm{PE}};\mathcal{T}_{\mathrm{ACT}}]}(\mathbb{R}^{d'})$ for some positive integers $k, d'$, and $\{d_i\}_{i=1}^k$, if its equivalent local function $\phi_\omega$ can be implemented by a multi-layer network with activation functions in $\mathcal{T}_{\mathrm{ACT}}$

Besides the local operators induced feedforward networks, we also have the following non-local closed operator induced by attention (Lemma F.12).

**Lemma F.12** (AHA is a Closed Operator). Average-hard attention is a closed operator over $\mathcal{H}_{\mathrm{TF}[\phi_{\mathrm{PE}};\mathcal{T}_{\mathrm{ACT}}]}$, that is, for any $q, k \in \mathcal{H}_{\mathrm{TF}[\phi_{\mathrm{PE}};\mathcal{T}_{\mathrm{ACT}}]}(\mathbb{R}^d)$ and $v \in \mathcal{H}_{\mathrm{TF}[\phi_{\mathrm{PE}};\mathcal{T}_{\mathrm{ACT}}]}(\mathbb{R}^{d'})$, we have $\mathrm{aha}(q, k, v) \in \mathcal{H}_{\mathrm{TF}[\phi_{\mathrm{PE}};\mathcal{T}_{\mathrm{ACT}}]}(\mathbb{R}^{d'})$.

---

[8]Those operators cannot be defined with $\omega$

The proof of Lemma F.12 is similar to that of Lemma F.11, which uses the definition of $\mathcal{H}_{\mathrm{TF}[\phi_{\mathrm{PE}};\mathcal{T}_{\mathrm{ACT}}]}$ and the closedness property of concatenation (Lemma F.10). The proof is straightforward and omitted.

The addition operator is a custom closed operator, as it can be expressed as a composition of the primitive operators in `FASP`:

---

**Algorithm 1** Implementation of addition operator, $\mathrm{add}(\psi_1, \psi_2)$

---

**Input** : Two seq-to-embedding functions $\psi_1, \psi_2 \in \mathcal{H}(\mathbb{R}^d)$
**Output** : A seq-to-embedding function $\psi^* \in \mathcal{H}(\mathbb{R})$
  $\psi_{\mathrm{cat}} \leftarrow [\psi_1, \psi_2]$    // Concatenate the two functions
  $\psi^* \leftarrow (\psi_{\mathrm{cat}})_1 + (\psi_{\mathrm{cat}})_2$    // Linear transformation – summation over both coordinates
  **return** $\psi^*$

---

Alternatively, in the inline format for composition, we can simply write: $\mathrm{add}(\psi, \psi') = \psi + \psi'$.

### F.2. Fine-Grained Types of Seq-to-Embedding Functions

So far we have been talking about seq-to-embedding functions whose ranges are $\mathbb{R}^d$. It turns out to be useful to consider more fine-grained types of seq-to-embedding functions whose range are only subset of $\mathbb{R}^d$. The main benefit of restricting output types is that it also simmplifies the construction of following operators, as they they only need to be defined on seq-to-embedding functions with smaller domains. In particular, we will be interested in and use the following three types:

- **Binary Seq-to-Embedding Functions**: These are seq-to-embedding functions whose range is $\{0,1\}^d$. We denote the set of all such functions as $\mathcal{H}(\{0,1\}^d)$.
- **Integer Seq-to-Embedding Functions**: These are seq-to-embedding functions whose range is $\mathbb{Z}^d$. We denote the set of all such functions as $\mathcal{H}(\mathbb{Z}^d)$.
- **One-Hot Seq-to-Embedding Functions**: Given a finite set $A$, we define $\mathcal{H}(\mathrm{onehot}(A))$ as the class of seq-to-embedding functions whose range is the set of one-hot encodings of elements in $A$. Specifically, for any $\psi \in \mathcal{H}(\mathrm{onehot}(A))$ and any input $x \in \Sigma^*$, $\psi(x) \in \{e_a : a \in A\}$ where $e_a \in \{0,1\}^{|A|}$ is the one-hot encoding of element $a \in A$. (See definition of $\mathrm{onehot}$ below, Definition F.13)
  One-hot embedding will be particularly useful at the last line of `FASP`, when we need to take argmax of the output embedding to get the final token. A recommended practice here for the readability of the code here is to ensure the last embedding before argmax computes the one-hot embedding of the desired output token.

**Definition F.13** (One-Hot Encoding). We define the one-hot encoding operator $\mathrm{onehot}_A : A \rightarrow \{0,1\}^{|A|}$ for any finite set $A$ as:

$$[\mathrm{onehot}_A(a)]_i = \begin{cases} 1 & \text{if } a \text{ is the } i\text{-th element of } A \text{ under some fixed ordering} \\ 0 & \text{otherwise} \end{cases} \tag{49}$$

We use $\mathrm{onehot}(A) \triangleq \{\mathrm{onehot}_A(a) \mid a \in A\}$ to denote the set of all one-hot encoding operators for all finite sets $A$.

The inverse operation, which maps a one-hot vector back to the corresponding element, is denoted as $\mathrm{onehot}_A^{-1} : \{0,1\}^{|A|} \rightarrow A$, defined as:

$$\mathrm{onehot}_A^{-1}(v) = a \text{ where } a \text{ is the } i\text{-th element of } A \text{ and } v_i = 1 \tag{50}$$

When the set $A$ is clear from context, we may simply write $\mathrm{onehot}$ and $\mathrm{onehot}^{-1}$ for brevity.

## G. Notable Special Cases of **FASP**

In this section, we would like to discuss some special cases of `FASP` that are of particular interest. We consider four special cases of `FASP`, from less expressive to more expressive (see Lemma G.1), that are of particular interest: $\mathrm{FASP}[0; [\cdot]_+]$ (Appendix G.1), $\mathrm{FASP}[0; [\cdot]_+, \times]$ (Appendix G.2), $\mathrm{FASP}[\texttt{is\_first}; [\cdot]_+, \times]$ (Appendix G.3) and $\mathrm{FASP}[\texttt{seq\_len}; [\cdot]_+, \times]$ (Appendix G.4).

We first formally define the above mentioned positional embeddings and activation functions. We start with positional embeddings.

- $0 : \mathbb{N}^+ \to \{0\}$. We use $0$ to denote the constant position embedding that always outputs $0$, which is equivalent to not having positional encoding.
- $\text{is\_first} : \mathbb{N}^+ \to \{0, 1\}$. We use $\text{is\_first}$ to denote the function that outputs $1$ if the input is the first position and $0$ otherwise. That is, $\text{is\_first}(n) = \mathbf{1}[n = 1]$.
- $\text{seq\_len} : \mathbb{N}^+ \to \mathbb{N}^+$. We use $\text{seq\_len}$ to denote the identity mapping over $\mathbb{N}^+$, which returns the position index itself. That is, $\text{seq\_len}(n) = n$. This allows the model to directly access the current sequence length.

Now we define the non-linear activation functions that will be used in this subsection.

- $\text{ReLU}(\text{or}[\cdot]_+) : \mathbb{R} \to \mathbb{R}$. We define $\text{ReLU}(x) = [x]_+ = \max(x, 0)$ to bethe ReLU activation function, which outputs the input if it is positive and $0$ otherwise.
- $\text{multiply}(\text{or}\times) : \mathbb{R} \times \mathbb{R} \to \mathbb{R}$. We use $\times$ to denote the multiplication function, which outputs the product of its two inputs.
- $\text{square} : \mathbb{R} \to \mathbb{R}$. We use this to denote the square function, which outputs the square of its input, i.e., $\text{square}(x) = x^2$.
- $\text{ReGLU} : \mathbb{R} \times \mathbb{R} \to \mathbb{R}$. We use this to denote the ReGLU (Rectified Gated Linear Unit) activation, which multiplies the first input by the rectified second input, that is, $\text{ReGLU}(x, y) = x[y]_+$.

**Lemma G.1.** Let $\phi_{\text{PE}}$ and $\phi'_{\text{PE}}$ be two feature functions for positional embedding, and $\mathcal{T}_{\text{ACT}}$ and $\mathcal{T}'_{\text{ACT}}$ be two sets of activation functions. If $\phi'_{\text{PE}} \in \text{FASP}[\phi_{\text{PE}}; \mathcal{T}_{\text{ACT}}]$ and $\mathcal{T}'_{\text{ACT}} \subseteq \text{FASP}[\phi_{\text{PE}}; \mathcal{T}_{\text{ACT}}]$, then $\text{FASP}[\phi'_{\text{PE}}; \mathcal{T}'_{\text{ACT}}] \subseteq \text{FASP}[\phi_{\text{PE}}; \mathcal{T}_{\text{ACT}}]$.

*Proof of Lemma G.1.* Since $\phi'_{\text{PE}} \in \text{FASP}[\phi_{\text{PE}}; \mathcal{T}_{\text{ACT}}]$, there exists a program in $\text{FASP}[\phi_{\text{PE}}; \mathcal{T}_{\text{ACT}}]$ that computes $\phi'_{\text{PE}}$. Similarly, for each activation function $\sigma' \in \mathcal{T}'_{\text{ACT}}$, there exists a program in $\text{FASP}[\phi_{\text{PE}}; \mathcal{T}_{\text{ACT}}]$ that computes $\sigma'$. Given any program in $\text{FASP}[\phi'_{\text{PE}}; \mathcal{T}'_{\text{ACT}}]$, we can transform it into a program in $\text{FASP}[\phi_{\text{PE}}; \mathcal{T}_{\text{ACT}}]$ by: (1) replacing each use of $\phi'_{\text{PE}}$ with its implementation in $\text{FASP}[\phi_{\text{PE}}; \mathcal{T}_{\text{ACT}}]$, and (2) replacing each activation function $\sigma' \in \mathcal{T}'_{\text{ACT}}$ with its implementation in $\text{FASP}[\phi_{\text{PE}}; \mathcal{T}_{\text{ACT}}]$. This transformation preserves the functionality of the original program, showing that $\text{FASP}[\phi'_{\text{PE}}; \mathcal{T}'_{\text{ACT}}] \subseteq \text{FASP}[\phi_{\text{PE}}; \mathcal{T}_{\text{ACT}}]$. $\qquad\square$

**Theorem G.2** (Hierarchy of FASP Variants)**.** The following containment relations hold between variants of FASP:

$$\text{FASP}[0; [\cdot]_+] \subseteq \text{FASP}[0; [\cdot]_+, \times] \subseteq \text{FASP}[\text{is\_first}; [\cdot]_+, \times] \subseteq \text{FASP}[\text{seq\_len}; [\cdot]_+, \times] \tag{51}$$

where each inclusion represents a strict increase in expressiveness.

**Why we care about FASP**$[0; [\cdot]_+, \times]$ Most modern LLM architectures use 2-layer MLP with gated linear units (GLU) (Dauphin et al., 2017) as the activation function (Equation (34)), such as SwishGLU (Shazeer, 2020), which is a variant of GLU with Swish activation (Ramachandran et al., 2017). For simplicity, we focus on ReGLU, which is a variant of GLU with ReLU activation (Dauphin et al., 2017), and also the limit of SwishGLU as the Swish activation approaches ReLU by letting $\beta \to \infty$.

**Theorem G.3** (Equivalent Expressiveness of Different Activation Sets)**.** The following function classes are equivalent: $\text{FASP}[0; [\cdot]_+, \times] = \text{FASP}[0; [\cdot]_+, \text{square}] = \text{FASP}[0; \text{ReGLU}]$.

*Proof of Theorem G.3.* We prove that $\text{FASP}[0; [\cdot]_+, \times] = \text{FASP}[0; [\cdot]_+, \text{square}] = \text{FASP}[0; \text{ReGLU}]$ by showing that both $\text{FASP}[0; [\cdot]_+, \text{square}]$ and $\text{FASP}[0; \text{ReGLU}]$ are equivalent to $\text{FASP}[0; [\cdot]_+, \times]$.

**Equivalence of FASP**$[0; [\cdot]_+, \times]$ **and FASP**$[0; [\cdot]_+, \textbf{square}]$**:** For the forward direction ($\text{FASP}[0; [\cdot]_+, \times] \subseteq \text{FASP}[0; [\cdot]_+, \text{square}]$), we show that multiplication can be expressed using square and ReLU:

$$\text{multiply}(x, y) = x \cdot y = \frac{(x + y)^2 - x^2 - y^2}{2} = \frac{\text{square}(x + y) - \text{square}(x) - \text{square}(y)}{2} \tag{52}$$

For the reverse direction ($\text{FASP}[0; [\cdot]_+, \text{square}] \subseteq \text{FASP}[0; [\cdot]_+, \times]$), we observe that square is simply multiplication with itself:

$$\text{square}(x) = x^2 = x \cdot x = \text{multiply}(x, x) \tag{53}$$

**Equivalence of FASP$[0; [\cdot]_+, \times]$ and FASP$[0; \text{ReGLU}]$:** For the forward direction (FASP$[0; [\cdot]_+, \times] \subseteq$ FASP$[0; \text{ReGLU}]$), we need to show that both ReLU and multiplication can be expressed using ReGLU:

$$\text{ReLU}(x) = [x]_+ = \text{ReGLU}(x, 1) \tag{54}$$

$$\text{multiply}(x, y) = x \cdot y = \text{ReGLU}(x, y) - \text{ReGLU}(x, -y) \tag{55}$$

For the reverse direction (FASP$[0; \text{ReGLU}] \subseteq$ FASP$[0; [\cdot]_+, \times]$), we can directly express ReGLU using ReLU and multiplication:

$$\text{ReGLU}(x, y) = x[y]_+ = x \cdot [y]_+ = \text{multiply}(x, \text{ReLU}(y)) \tag{56}$$

Therefore, FASP$[0; [\cdot]_+, \times] =$ FASP$[0; [\cdot]_+, \text{square}] =$ FASP$[0; \text{ReGLU}]$. $\qquad\square$

## G.1. Expressiveness of FASP$[0; [\cdot]_+]$

By Lemma F.11, all the local operators that can be written as MLP with ReLU activation are in FASP$[0; [\cdot]_+]$. This includes:

1. Arithmetic operators over reals (addition, subtraction, max, min):
    - add $: \mathcal{H}(\mathbb{R}) \times \mathcal{H}(\mathbb{R}) \to \mathcal{H}(\mathbb{R})$. See Example F.9. We also write $\psi_1 + \psi_2$ for add$(\psi_1, \psi_2)$.
    - minus $: \mathcal{H}(\mathbb{R}) \times \mathcal{H}(\mathbb{R}) \to \mathcal{H}(\mathbb{R})$, minus$(\psi_1, \psi_2) \triangleq$ add$(\psi_1, -\psi_2)$. We also write $\psi_1 - \psi_2$ for minus$(\psi_1, \psi_2)$.
    - max $: \mathcal{H}(\mathbb{R}) \times \mathcal{H}(\mathbb{R}) \to \mathcal{H}(\mathbb{R})$, max$(\psi_1, \psi_2) \triangleq [\psi_1 - \psi_2]_+ \psi_2$.
    - min $: \mathcal{H}(\mathbb{R}) \times \mathcal{H}(\mathbb{R}) \to \mathcal{H}(\mathbb{R})$, min$(\psi_1, \psi_2) \triangleq -[\psi_1 - \psi_2]_+ \psi_2$.

2. Boolean operators(AND, OR, NOT, XOR): For any $\psi_1, \psi_2 \in \mathcal{H}(\{0, 1\})$, boolean operators are defined as:
    - and $: \mathcal{H}(\{0, 1\}) \times \mathcal{H}(\{0, 1\}) \to \mathcal{H}(\{0, 1\})$, and$(\psi_1, \psi_2) \triangleq$ min$(\psi_1, \psi_2)$. We also denote it as $\psi_1 \wedge \psi_2$.
    - not $: \mathcal{H}(\{0, 1\}) \to \mathcal{H}(\{0, 1\})$, defined as: not$(\psi) \triangleq 1 - \psi$. We also denote it as $\neg\psi$.
    - or $: \mathcal{H}(\{0, 1\}) \times \mathcal{H}(\{0, 1\}) \to \mathcal{H}(\{0, 1\})$, defined as: or$(\psi_1, \psi_2) \triangleq \neg(\neg\psi_1 \wedge \neg\psi_2)$. We also denote it as $\psi_1 \vee \psi_2$.
    - xor $: \mathcal{H}(\{0, 1\}) \times \mathcal{H}(\{0, 1\}) \to \mathcal{H}(\{0, 1\})$, defined as: xor$(\psi_1, \psi_2) \triangleq (\psi_1 \vee \psi_2) \wedge \neg(\psi_1 \wedge \psi_2)$. We also denote it as $\psi_1 \veebar \psi_2$.

3. Comparison operators over integers (less than, equality, etc.);
    - leq$(\psi_1, \psi_2)$: For every input $x \in \Sigma^*$, the less-than-or-equal operator leq $: \mathcal{H}(\mathbb{Z}) \times \mathcal{H}(\mathbb{Z}) \to \mathcal{H}(\{0, 1\})$ returns 1 if the first argument $\psi_1$ is less than or equal to the second argument $\psi_2$, otherwise it returns 0. Because it is a comparison operator defined only over integers, it admits the following equivalent definition:

$$\text{leq}(\psi_1, \psi_2) \triangleq [\psi_2 - \psi_1 + 1]_+ - [\psi_2 - \psi_1]_+ \quad (\text{written as } \psi_1 \le \psi_2) \tag{57}$$

    - The remaining comparison operators can be derived from less, which all have type $\mathcal{H}(\mathbb{Z}) \times \mathcal{H}(\mathbb{Z}) \to \mathcal{H}(\{0, 1\})$:

$$\text{geq}(\psi_1, \psi_2) \triangleq \text{leq}(\psi_2, \psi_1) \quad (\text{written as } \psi_1 \ge \psi_2) \tag{58}$$

$$\text{equal}(\psi_1, \psi_2) \triangleq \text{leq}(\psi_1, \psi_2) \wedge \text{leq}(\psi_2, \psi_1) \quad (\text{written as } \psi_1 = \psi_2) \tag{59}$$

$$\text{less}(\psi_1, \psi_2) \triangleq \text{leq}(\psi_1, \psi_2 - 1) \quad (\text{written as } \psi_1 < \psi_2) \tag{60}$$

$$\text{greater}(\psi_1, \psi_2) \triangleq \text{less}(\psi_2, \psi_1) \quad (\text{written as } \psi_1 > \psi_2) \tag{61}$$

$$\text{neq}(\psi_1, \psi_2) \triangleq \text{not}(\text{equal}(\psi_1, \psi_2)) \quad (\text{written as } \psi_1 \ne \psi_2) \tag{62}$$

    It is worth noting that equal can be extended to vector inputs, $\cup_{d \in \mathbb{N}^+} \mathcal{H}(\mathbb{Z}^d) \times \mathcal{H}(\mathbb{Z}^d)$ by comparing each coordinate of the two vectors and take the logical AND of all the results. Similarly we can extend neq to vector inputs by still setting it to be not $\circ$ equal.

4. All operators on finite discrete inputs (with one-hot encoding). Namely all operators with signature $\mathcal{H}(\text{onehot}(A_1)) \times \mathcal{H}(\text{onehot}(A_2)) \times \ldots \times \mathcal{H}(\text{onehot}(A_n)) \to \mathcal{H}$ for finite sets $A_1, A_2, \ldots, A_n$. In particular this includes the kronecker-product operator $\otimes : \mathcal{H}(\text{onehot}(A_1)) \times \mathcal{H}(\text{onehot}(A_2)) \to \mathcal{H}(\text{onehot}(A_1 \times A_2)$, where

$$\otimes(\psi_1, \psi_2)(x) = (\psi_1 \otimes \psi_2)(x) = \psi_1(x) \otimes \psi_2(x), \tag{63}$$

for any $x \in \Sigma^*$. Here $\otimes$ on RHS is just the usual kronecker product in on vector space. For simplicity, we will use $a_1 \in A_1$ and $a_2 \in A_2$ to denote the coordinates of $\psi_1$ and $\psi_2$ respectively, and use $(a_1, a_2)$ to denote the coordinate of $\psi_1 \otimes \psi_2$. We can construct $\psi_1 \otimes \psi_2$ by setting $(\psi_1 \otimes \psi_2)_{(a_1,a_2)} = \psi_{1_{a_1}} \text{and} \psi_{2_{a_2}}$ for all $a_1 \in A_1$ and $a_2 \in A_2$. [9]

We can also define the following non-local closed operators:

1. **Running Average**: For any $\psi \in \mathcal{H}(\mathbb{R})$, the running average operator $\texttt{average} : \mathcal{H}(\mathbb{R}) \to \mathcal{H}(\mathbb{R})$ computes the average of all prefix function values. For any input $x \in \Sigma^*$ of length $n$:

$$\texttt{average}(\psi)(x) = \frac{1}{n} \sum_{j=1}^{n} \psi(x_{1:j}) \tag{64}$$

This can be constructed using average-hard attention with constant queries and keys: $\texttt{average}(\psi) \triangleq \texttt{aha}(\mathbf{1}, \mathbf{1}, \psi)$.

2. **Running Maximum**: For any $\psi \in \mathcal{H}(\mathbb{R})$, the running maximum operator $\texttt{seq\_max} : \mathcal{H}(\mathbb{R}) \to \mathcal{H}(\mathbb{R})$ returns the maximum value across all prefixes. For any input $x \in \Sigma^*$ of length $n$:

$$\texttt{seq\_max}(\psi)(x) = \max_{j=1,\ldots,n} \psi(x_{1:j}) \tag{65}$$

This can be constructed as $\texttt{seq\_max}(\psi) \triangleq \texttt{aha}(\psi, \psi, \psi)$, where the position with maximum value receives all attention.

3. **Running Minimum**: For any $\psi \in \mathcal{H}(\mathbb{R})$, the running minimum operator $\texttt{seq\_min} : \mathcal{H}(\mathbb{R}) \to \mathcal{H}(\mathbb{R})$ returns the minimum value across all prefixes. For any input $x \in \Sigma^*$ of length $n$:

$$\texttt{seq\_min}(\psi)(x) = \min_{j=1,\ldots,n} \psi(x_{1:j}) \tag{66}$$

This can be implemented by negating the maximum of the negated function: $\texttt{seq\_min}(\psi) \triangleq -\texttt{seq\_max}(-\psi)$.

4. **Running Logical AND**: For any $\psi \in \mathcal{H}(\{0,1\})$, the running logical AND operator $\texttt{seq\_and} : \mathcal{H}(\{0,1\}) \to \mathcal{H}(\{0,1\})$ computes the conjunction of all prefix values. For any input $x \in \Sigma^*$ of length $n$:

$$\texttt{seq\_and}(\psi)(x) = \bigwedge_{j=1}^{n} \psi(x_{1:j}) \tag{67}$$

Since binary values are used, this is equivalent to the running minimum: $\texttt{seq\_and}(\psi) \triangleq \texttt{seq\_min}(\psi)$.

5. **Running Logical OR**: For any $\psi \in \mathcal{H}(\{0,1\})$, the running logical OR operator $\texttt{seq\_or} : \mathcal{H}(\{0,1\}) \to \mathcal{H}(\{0,1\})$ computes the disjunction of all prefix values. For any input $x \in \Sigma^*$ of length $n$:

$$\texttt{seq\_or}(\psi)(x) = \bigvee_{j=1}^{n} \psi(x_{1:j}) \tag{68}$$

Since binary values are used, this is equivalent to the running maximum: $\texttt{seq\_or}(\psi) \triangleq \texttt{seq\_max}(\psi)$.

### G.2. Expressiveness of $\texttt{FASP}[0; [\cdot]_+, \times]$

$\texttt{FASP}[0; [\cdot]_+, \times]$ allows one more activation function, $x, y \mapsto x \times y$ on top of $\texttt{FASP}[0; [\cdot]_+]$ discussed in the previous section. We first recall that multiplication activation induces the following multiplication operator $\texttt{multiply} : \mathcal{H}(\mathbb{R}) \times \mathcal{H}(\mathbb{R}) \to \mathcal{H}(\mathbb{R})$, which is defined as:

$$\texttt{multiply}(\psi_1, \psi_2)(x) \triangleq \psi_1(x) \cdot \psi_2(x) \tag{69}$$

for any $x \in \Sigma^*$. We will use $\psi_1 \cdot \psi_2$, $\psi_1 \psi_2$, or $\psi_1 \times \psi_2$ to denote $\texttt{multiply}(\psi_1, \psi_2)$ hereafter.

In $\texttt{FASP}[0; [\cdot]_+, \times]$ we have the following closed operator:

---

[9]Note this construction uses the fact that each coordinate of $\psi_1, \psi_2$ is boolean. We cannot define the kronecker product on infinite domains integers or reals without multiplication/square/gated ReLu activation. Will return to this in next subsection (Appendix G.2).

**Conditional Operator**   We define a conditional operator $\texttt{if\_then\_else} : \mathcal{H}(\{0,1\}) \times \mathcal{H}(\mathbb{R}^d) \times \mathcal{H}(\mathbb{R}^d) \to \mathcal{H}(\mathbb{R}^d)$ for control flow, which selects between values based on a condition:

$$\texttt{if\_then\_else}(\psi_{\text{cond}}, \psi_{\text{true}}, \psi_{\text{false}})(x) = \begin{cases} \psi_{\text{true}}(x) & \text{if } \psi_{\text{cond}}(x) = 1 \\ \psi_{\text{false}}(x) & \text{if } \psi_{\text{cond}}(x) = 0 \end{cases} \tag{70}$$

This can be constructed directly from previously defined closed operators:

$$\texttt{if\_then\_else}(\psi_{\text{cond}}, \psi_{\text{true}}, \psi_{\text{false}}) \triangleq \psi_{\text{cond}} \cdot \psi_{\text{true}} + (\neg \psi_{\text{cond}}) \cdot \psi_{\text{false}}. \tag{71}$$

## G.3. Expressiveness of $\texttt{FASP}[\texttt{is\_first}; [\cdot]_+, \times]$

We first recall the definition of $\texttt{is\_first}$:

$$\texttt{is\_first}(n) = \mathbf{1}[n = 1]. \tag{72}$$

where $\mathbf{1}[\cdot]$ is the indicator function. In practice, it is important for language model to know whether the current position is the first position, and it is standard to use [BOS] token to indicate the beginning of the sequence. By using $\texttt{is\_first}$ position embedding, we achieve the similar effect as using [BOS] token. It is easy to prove that LLM cannot count without any positional embedding, even with softmax attention. Concretely, without positional encoding, for any parameter $\theta$, any token $a \in \Sigma$, any integer $n$, $\pi_\theta(a^n) = \pi_\theta(a)$. So in some sense $\texttt{is\_first}$ is the minimal positional embedding that allows LLM to count.

Simply adding $\texttt{is\_first}$ position embedding allows us to define the following closed operators in $\texttt{FASP}[\texttt{is\_first}; [\cdot]_+]$, and thus also in $\texttt{FASP}[\texttt{is\_first}; [\cdot]_+, \times]$:

- $\texttt{inv\_seq\_len}$: We define $\texttt{inv\_seq\_len}(n) = 1/n$ as the inverse of sequence length by constructing

$$\texttt{inv\_seq\_len} \triangleq \texttt{average}(\texttt{seq\_len} = \mathbf{1}). \tag{73}$$

  This operator computes the inverse of the current sequence length, which is useful for normalizing operations that depend on sequence length.
- $\texttt{is\_pos\_k}$: We define $\texttt{is\_pos\_k}(n) = \mathbf{1}[n = k]$ as the indicator function for the $k$-th position. This can be constructed as:

$$\texttt{is\_pos\_k} = \texttt{geq}_0(k + 1 - k(k+1) \cdot \texttt{inv\_seq\_len}) \wedge \texttt{geq}_0(k(k+1) \cdot \texttt{inv\_seq\_len} - k - 1) \tag{74}$$

  where $\texttt{geq}_0 : \mathcal{H}((-\infty, -1] \cup [0, \infty)) \to \mathcal{H}(\{0, 1\})$ is defined as $\texttt{geq}_0(\psi) = [\psi + 1]_+ - [\psi]_+$. $\texttt{geq}_0$ satisfies that for any $x \in \Sigma^*$, $\texttt{geq}_0(\psi)(x) = 1$ if $\psi(x) \geq 0$ and $0$ if $\psi(x) \leq -1$.
  This works because at position $n$, we have $\texttt{inv\_seq\_len}(n) = 1/n$. When $n = k$, both $k + 1 - k(k+1)/n = k + 1 - k(k+1)/k = k + 1 - (k+1) = 0$ and $k(k+1)/n - k - 1 = k(k+1)/k - k - 1 = (k+1) - k - 1 = 0$, so both terms are $\leq 0$. When $n \neq k$, at least one of the expressions will be $> 0$, making the result false.
- $\texttt{rha}$: We define Rightmost-Hard Attention $\texttt{rha} : \mathcal{H}(\mathbb{Z}^{d'}) \times \mathcal{H}(\mathbb{Z}^{d'}) \times \mathcal{H}(\mathbb{R}^d) \to \mathcal{H}(\mathbb{R}^d)$ as the hard-attention which breaks tie by picking most recent argmax of attention score for any positive integer $d, d'$. That is, for any $x \in \Sigma^n$:

$$\texttt{rha}(\psi_q, \psi_k, \psi_v)(x) = \psi_v(x_{1:j^*}) \tag{75}$$

  where $j^*$ is the rightmost position with maximal query-key match:

$$j^* = \max\{j \mid \psi_q(x) \cdot \psi_k(x_{1:j}) = \max_{k \leq n} \psi_q(x) \cdot \psi_k(x_{1:k})\}. \tag{76}$$

  This can be implemented using the $\texttt{aha}$ primitive with augmented query and key vectors:

$$\texttt{rha}(\psi_q, \psi_k, \psi_v) \triangleq \texttt{aha}\left([\psi_q, \mathbf{1}], [\psi_k, \texttt{inv\_seq\_len}], \psi_v\right). \tag{77}$$

  For any two positions $j < j'$ with identical query-key match scores in the original space, the augmented scores will differ by $-1/j + 1/j'$, which is always positive since $-1/j > -1/j'$ when $j < j'$. This ensures that when multiple positions have the same original match score, the rightmost position (largest $j$) will achieve the highest augmented score, making $\texttt{rha}$ select it as the unique maximum.

We also have the following variant of rightmost hard attention $\texttt{rha}$ which relies on the multiplication activation, $\texttt{rightmost\_best\_match}$:

**Rightmost Best Match**   For any positive integer $d, d'$, we define $\texttt{rightmost\_best\_match} : \mathcal{H}(\mathbb{Z}^{d'}) \times \mathcal{H}(\mathbb{Z}^{d'}) \times \mathcal{H}(\mathbb{R}^d) \to \mathcal{H}(\mathbb{R}^d)$ as the variant of rightmost hard attention which minimizes the $\ell_2$ distance between key and query, as supposed to maximize their inner product. That is, for any $x \in \Sigma^n$:

$$\texttt{rightmost\_best\_match}(\psi_q, \psi_k, \psi_v)(x) \;=\; \psi_v(x_{1:j^*}) \tag{78}$$

where $j^*$ is the rightmost position with maximal query-key match quantified by the $\ell_2$ norm:

$$j^* = \max\left(\arg\min_{j \leq n} \|\psi_q(x) - \psi_k(x_{1:j})\|_2\right), \tag{79}$$

This can be implemented using the $\texttt{rha}$ and the multiplication operator:

$$\texttt{rightmost\_best\_match}(\psi_q, \psi_k, \psi_v) \;\triangleq\; \texttt{rha}\left([\psi_q, \mathbf{1}], [2\psi_k, -\psi_k^\top \psi_k], \psi_v\right), \tag{80}$$

For ant input $x$, this definition retrieves the value at the position $k$ that maximizes $2\psi_q(x)^\top \psi_k(x_{1:k}) - \psi_k(x_{1:k})^\top \psi_k(x_{1:k})$, or equivalently, minimizes $\|\psi_q(x) - \psi_k(x_{1:k})\|_2^2$.

**Rightmost Exact Match**   For any positive integer $d, d'$, we define $\texttt{rightmost\_exact\_match} : \mathcal{H}(\mathbb{Z}^{d'}) \times \mathcal{H}(\mathbb{Z}^{d'}) \times \mathcal{H}(\mathbb{R}^d) \times \mathcal{H}(\mathbb{R}^d) \to \mathcal{H}(\mathbb{R}^d)$ as the variant of rightmost best match (and thus variant of rightmost hard attention) which returns the value $\psi_v$ associated with the rightmost key $\psi_k$ that exactly matches the query $\psi_q$, and otherwise returns the default value $\psi_d$. That is, for any $x \in \Sigma^n$:

$$\begin{aligned}
&\texttt{rightmost\_exact\_match}(\psi_q, \psi_k, \psi_v, \psi_d)(x) \\
&\triangleq \texttt{if\_then\_else}(\texttt{rightmost\_best\_match}(\psi_q, \psi_k, \psi_k) = \psi_q, \psi_v, \psi_d).
\end{aligned} \tag{81}$$

### G.4. Expressiveness of $\texttt{FASP}[\texttt{seq\_len}; [\cdot]_+, \times]$

We end this section by considering the most expressive case $\texttt{FASP}[\texttt{seq\_len}; [\cdot]_+, \times]$ so far, where $\texttt{seq\_len}$ is the identity mapping over $\mathbb{N}^+$. With positional embedding $\texttt{seq\_len}$, we can define the following partial sum operator, $\sum$, which is closed in $\texttt{FASP}[\texttt{seq\_len}; \times]$, and thus also $\texttt{FASP}[\texttt{seq\_len}; [\cdot]_+, \times]$.

**Partial Sum:**   We define $\texttt{sum} : \mathcal{H}(\mathbb{R}) \to \mathcal{H}(\mathbb{R})$ as the operator that computes the running sum. That is, for any $x \in \Sigma^n$:

$$\texttt{sum}(\psi)(x) \;=\; \sum_{j=1}^{n} \psi(x_{1:j}) \tag{82}$$

This can be constructed by scaling the average operator, i.e., $\texttt{sum}(\psi) = \texttt{average}(\psi) \cdot \texttt{seq\_len}$.

We note that the ability of transformer to express or compute $\texttt{seq\_len}$ (e.g., in terms of precision) is necessary to define the partial sum operator, as the sum of the constant token embedding of value 1 immediately gives the sequence length, which implies that any transformer class that can compute partial sum necessarily can also compute $\texttt{seq\_len}$, even without any non-linear actiation function.

We also note that with $\texttt{sum}$ as a closed operator in $\texttt{FASP}[\texttt{seq\_len}; [\cdot]_+, \times]$, it is clear that $\texttt{FASP}[\texttt{seq\_len}; [\cdot]_+, \times]$ is a superset of C-RASP (Yang & Chiang, 2024).

## H. Proof of Theorem 5.1: Main Result

We state the formal version of Theorem 5.1 as follows:

**Theorem H.1** (Main). Let $\textsf{TM} = (\mathcal{A}, b, Q, q_0, \delta, Q_{\text{accept}}, Q_{\text{reject}})$ be any single-tape Turing machine that has time complexity $T(x)$ and space complexity $S(x)$ on input $x \in (\mathcal{A} \setminus \{b\})^*$. There exists a transformer with constant depth, constant embedding dimension, Gated ReLU activation, and positional embedding $n \mapsto n$, average hard attention, such that for the next-token predictor $\pi_\theta$ implemented by this transformer and the reduction rule $\phi'$ defined in (9), the following holds:

1. $\text{PENCIL}_{f_{\pi_\theta}, \phi'}$ produces the same output (accept or reject) as $\textsf{TM}$ on $x$.

2. The total number of tokens generated by $\text{PENCIL}_{f_{\pi_\theta}, \phi'}$ is $\mathcal{O}(T(x))$.

3. The maximal context length used by $\text{PENCIL}_{f_{\pi_\theta}, \phi'}$ during generation is at most $\mathcal{O}(S(x))$.

**Problem Setup** Our goal is to construct a learnable model that can replicate PENCIL's model generation process, since the reduction process can be realized by the reduction rule. Specifically, at each iteration $i$, starting from a compressed state

$$x^{(i-0.5)} \triangleq s \circ f_\pi^{t_{i-1}}(x) \in \Sigma^*, \tag{83}$$

we need to construct a model that can autoregressively produce the extended sequence

$$x^{(i)} \triangleq \left( f_\pi^{t_i - t_{i-1}} \circ s \circ f_\pi^{t_{i-1}}(x), \; [\text{SEP}], \; s \circ f_\pi^{t_i}(x), \; [\text{RETURN}] \right) \in \Sigma^*. \tag{84}$$

Intuitively, $x^{(i)}$ includes a newly generated block of uncompressed tokens representing the computations of Turing machine, followed by a separator $[\text{SEP}]$, followed by an updated compressed state representing Turing machine's current memory, and finally the token $[\text{RETURN}]$.

The base case $x^{(0.5)} \triangleq x$ serves as the initial prompt. Iteration $i$ then starts from $x^{(i-0.5)}$ and ends with $x^{(i)}$. Here $\pi : \hat{\Sigma}^* \to \hat{\Sigma}$ is the next-token generator in the autoregressive machine that simulates Turing Machine, where $\hat{\Sigma} = \mathcal{Q} \times \mathcal{A} \times \{-1, 0, 1\}$. To implement this mapping, PENCIL uses a transformer as the next-token generator $\pi_\theta : \Sigma^* \to \Sigma$ where transformer vocabulary is $\Sigma \triangleq \hat{\Sigma} \cup \{[\text{SEP}], [\text{RETURN}]\}$ and $\theta$ is the transformer parameter. It suffices to show that there is a next-token generator $\pi' \in \text{FASP}[n; [\cdot]_+, \times]$ (or equivalently, expressible by a transformer with $n \mapsto n$ positional embedding, average-hard attention and Gated ReLU activation) that can

1. simulate the next-token generator in the autoregressive machine that simulates Turing Machine.
2. generate the special token $[\text{SEP}]$ at the earliest time that the length will be halved after summarization.
3. simulate the summarization process.

**Transformer Construction as FASP Program:** The construction of the transformer is defined by the following FASP program where each line uses a close operator to construct a new transformer model in the desired class. For readability, we use colored keywords: orange for primitive functions, red for non-local closed operators, and blue for local closed operators. We below clarify the new primitive seq-to-embedding functions used here. [10]

1. `get_symbol` : $\Sigma \to \text{onehot}(A)$ - Maps a token to a one-hot encoding of the symbol part of the token, extracting the symbol from state-symbol-move triples. Returns a one-hot vector in the symbol alphabet space.

2. `get_move` : $\Sigma \to \{-1, 0, 1\}$ - Maps a token to a scalar value representing the move direction (-1 for left, 0 for stay, 1 for right) extracted from state-symbol-move triples.

3. `get_state` : $\Sigma \to \text{onehot}(Q)$ - Maps a token to a one-hot vector of the state part, extracting the state information from state-symbol-move triples.

Most of the closed operators used in the program below are all already defined in Appendix G, except `transition`, which maps one hot embedding of state and symbol to the onehot embedding of (next state, next symbol, next move) in $\Sigma$. The following program thus completes the proof of Theorem 5.1

---

[10]We are proving for the simplified reduction rule only. The proof extends to the original PENCIL reduction rule in a straight-forward manner.

```
# Detect separator token
is_sep = (get_token = onehot([SEP]))
exist_sep = seq_or(is_sep)

# Phase masks to distinguish between simulation and summarization phases
sim_phase_mask = not exist_sep
sum_phase_mask = exist_sep and (not is_sep)

# Position tracking for Simulation, frozen in SUMMARIZATION (after [SEP] is generated)
next_sim_pos = seq_sum(get_move and sim_phase_mask)
current_sim_pos = next_sim_pos - (get_move and sim_phase_mask)
max_pos = seq_max(current_sim_pos)
min_pos = seq_min(current_sim_pos)
expected_sum_len = max_pos - min_pos + ReLU(max_pos- next_sim_pos -1) + 1

# SIMULATION Phase
# Get current symbol at head position
current_symbol = rightmost_exact_match(next_sim_pos,current_sim_pos,get_symbol,onehot(b))
# Compute next step based on transition function
simulation_step = transition(get_state, current_symbol)

# Decide whether to continue simulation or switch to summarization
end_simulation = sequence_len >= 2* expected_sum_len
simulation=if_then_else(end_simulation, onehot([SEP]), simulation_step)

# SUMMARIZATION Phase
current_sum_pos = seq_sum(get_move and sum_phase_mask)
current_sum_len = seq_sum(sum_phase_mask)

# Decide the next move in SUMMARIZATION PHASE
next_move = compute_move(current_sum_len, next_sim_pos, max_pos, min_pos)

# By construction, exact match always happens.
summary_symbol=rightmost_best_match(current_sum_pos+min_pos,current_sim_pos,get_symbol)
summary_step = get_state ⊗ summary_symbol ⊗ onehot(next_move)

# Check if we've reached the final position in summarization
end_summary = (current_sum_len = expected_sum_len)
summary = if_then_else(end_summary, onehot([RETURN], summary_step))

# MAIN - Select appropriate action based on current phase
result = if_then_else(exist_sep, summary, simulation)
```

# I. Omitted Proofs from Section 5 for Genreal Autoregressive Machines

**Lemma I.1.** Let $s$ be a state function of a autoregressive machine $\mathcal{M} = (\Sigma, \pi, \Sigma_{\text{accept}}, \Sigma_{\text{reject}})$. It holds that $s \circ f_\pi^k \circ s = s \circ f_\pi^k$ and that $\pi^{k+1} = \pi^{k+1} \circ s$ for any $k \geq 0$.

*Proof of Lemma I.1.* For any $z \in \Sigma^*$, we have that $s^2(z) = s(z)$. Now let $x = s(z), x' = z$ and $y = \pi(z) = \pi(s(z))$, since $s(x) = s(x')$, we have $s((x, y)) = s((x', y))$, which further implies that

$$s(f_\pi(s(z))) = s((x, y)) = s((x', y)) = s(f_\pi(z)). \tag{85}$$

Therefore, $s \circ f_\pi \circ s = s \circ f_\pi$. Now we use induction to prove that $s \circ f_\pi^k \circ s = s \circ f_\pi^k$ for all $k \in \mathbb{N}^+$. The base case $k = 1$ is already proved. Now suppose $s \circ f_\pi^k \circ s = s \circ f_\pi^k$, we have

$$s \circ f_\pi^{k+1} \circ s = s \circ f_\pi \circ f_\pi^k \circ s = s \circ f_\pi \circ s \circ f_\pi^k \circ s = s \circ f_\pi \circ s \circ f_\pi^k = s \circ f_\pi \circ f_\pi^k \tag{86}$$

which completes the induction.

Now we turn to the second part, which is a simple consequence of the first part. Note that for $k \geq 1$, $\pi^k = \pi \circ f_\pi^{k-1} = \pi \circ s \circ f_\pi^{k-1}$. By first part, $s \circ f_\pi^{k-1} = s \circ f_\pi^{k-1} \circ s$. This completes the proof of the second part. $\qquad\square$

## I.1. Proof of Proposition 5.6

Recall that we partition the full generation into segments indexed by $i \in [I]$ where $I$ is the total number of iterations and each iteration corresponds to one effective reduction. Let $t_0 = 0$, and for each $i \geq 1$, define $t_i$ to be the smallest integer greater than $t_{i-1}$ such that

$$\left| s \circ f_\pi^{t_i}(x) \right| \leq \frac{1}{2} \left| f_\pi^{t_i - t_{i-1}} \circ s \circ f_\pi^{t_{i-1}}(x) \right|, \tag{87}$$

where $|\cdot|$ denotes sequence length. In words, $t_i$ is the next time step at which the (compressed) state is at most half the length of the newly generated segment. Each iteration $i$ therefore covers times from $t_{i-1} + 1$ to $t_i$.

We let $x^{(i)}$ denote the sequence

$$x^{(i)} \triangleq \left( f_\pi^{t_i - t_{i-1}} \circ s \circ f_\pi^{t_{i-1}}(x), \ [\text{SEP}], \ s \circ f_\pi^{t_i}(x), \ [\text{RETURN}] \right). \tag{88}$$

Applying $\phi_{\text{scroll}}$ then discards all tokens except the final compressed state

$$x^{(i+0.5)} \triangleq s \circ f_\pi^{t_i}(x) \tag{89}$$

which is treated as the initial sequence for the next iteration.

**Bounding the Maximum Sequence Length (Space)** Consider any point immediately before the $[\text{RETURN}]$ of iteration $i$. By definition of $t_i$, we have

$$\left| s \circ f_\pi^{t_i - 1}(x) \right| > \frac{1}{2} \left| f_\pi^{t_i - 1 - t_{i-1}} \circ s \circ f_\pi^{t_{i-1}}(x) \right|. \tag{90}$$

Hence, if we look at the entire sequence (88) its length is at most

$$2 \left| s \circ f_\pi^{t_i - 1}(x) \right| + 2 + \left| s \circ f_\pi^{t_i}(x) \right| + 2 = \mathcal{O}\big(S(\mathcal{M}, s, x)\big). \tag{91}$$

Here the additional "+2" accounts for the two special tokens $[\text{SEP}]$ and $[\text{RETURN}]$, plus a small constant overhead. Because $s \circ f_\pi^{t_i - 1}(x)$ (and also $s \circ f_\pi^{t_i}(x)$) is at most $S(\mathcal{M}, s, x)$ in length, we conclude that at every $[\text{RETURN}]$, the sequence is $\mathcal{O}(S(\mathcal{M}, s, x))$ long. This implies the maximum context length under PENCIL never exceeds $\mathcal{O}(S(\mathcal{M}, s, x))$.

**Bounding the Total Number of Tokens (Time)** Next, we show the total tokens generated (summing over all iterations) is $\mathcal{O}(T(\mathcal{M}, x))$. The critical point is that our reduction rule does not trigger too frequently: if we were to compress immediately after every single token (e.g. each Turing-machine step), we would incur an excessive time overhead. By only reducing when the sequence grows sufficiently large relative to the state size, we avoid inflating the total time cost. Formally, define

$$\ell_i \triangleq (t_i - t_{i-1}) + \left| s \circ f_\pi^{t_i}(x) \right| + 2, \tag{92}$$

which represents the cost (length) of generating the new tokens in iteration $i$, plus the two special tokens (such as `[SEP]` and `[RETURN]`). We wish to bound $\sum_{i=1}^{I} \ell_i$. From the definition of $t_i$, it follows that

$$\left(t_i - t_{i-1}\right) + \left|s \circ f_\pi^{t_i}(x)\right| \geq 2\left|s \circ f_\pi^{t_{i-1}}(x)\right|. \tag{93}$$

Summing up (93) from $i = 1$ to $I$ gives us

$$\left(t_I - t_0\right) + \left|s \circ f_\pi^{t_I}(x)\right| \geq \sum_{i=1}^{I} \left|s \circ f_\pi^{t_{i-1}}(x)\right|. \tag{94}$$

where $|s \circ f_\pi^{t_0}(x)| = 0$. Since $t_I \leq T(\mathcal{M}, x)$ (the total number of steps for $\mathcal{M}$), each iteration's generation cost can be bounded by a linear function of $t_I$ plus the space used by the states. Concretely, summing up $\ell_i$ over $i$ yields

$$\sum_{i=1}^{I} \ell_i \leq \sum_{i=1}^{I} \left[\left(t_i - t_{i-1}\right) + \left|s \circ f_\pi^{t_i}(x)\right| + 2\right] \leq 2\,t_I + 2\,I + \left|s \circ f_\pi^{t_I}(x)\right|. \tag{95}$$

Since $I \leq t_I$ (each iteration covers at least one time step) and $t_I \leq T(\mathcal{M}, x)$, we conclude $\sum_{i=1}^{I} \ell_i = \mathcal{O}\big(T(\mathcal{M}, x)\big)$.

**Conclusion** Together with our bound on the maximum sequence length, this shows that PENCIL simulates $\mathcal{M}$ using both *optimal space* $S(\mathcal{M}, s, x)$ and *optimal time* $T(\mathcal{M}, x)$. Thus, we complete the proof of Proposition 5.6.

## J. Omitted Proofs

### J.1. Omitted Proofs in Appendix E

*Proof of Lemma E.6.* Let $M = \arg\max_j x_j$ be the set of indices achieving the maximum value, and let $x_{\max} = \max_j x_j$. For any $i \in M$, we have $x_i = x_{\max}$, and for any $i \notin M$, we have $x_i < x_{\max}$. Consider the softmax function with temperature $\beta$:

$$[\text{softmax}_\beta(x)]_i = \frac{\exp(x_i/\beta)}{\sum_{j=1}^{n} \exp(x_j/\beta)}$$

$$= \frac{\exp(x_i/\beta)}{\sum_{j \in M} \exp(x_{\max}/\beta) + \sum_{j \notin M} \exp(x_j/\beta)}$$

For $i \in M$, as $\beta \to 0$:

$$\lim_{\beta \to 0}[\text{softmax}_\beta(x)]_i = \lim_{\beta \to 0} \frac{\exp(x_{\max}/\beta)}{|M|\exp(x_{\max}/\beta) + \sum_{j \notin M} \exp(x_j/\beta)}$$

$$= \lim_{\beta \to 0} \frac{1}{|M| + \sum_{j \notin M} \exp((x_j - x_{\max})/\beta)}$$

Since $x_j < x_{\max}$ for all $j \notin M$, we have $(x_j - x_{\max})/\beta \to -\infty$ as $\beta \to 0$, and thus $\exp((x_j - x_{\max})/\beta) \to 0$. This gives:

$$\lim_{\beta \to 0}[\text{softmax}_\beta(x)]_i = \frac{1}{|M|} \quad \text{for all } i \in M \tag{96}$$

For $i \notin M$, we have:

$$\lim_{\beta \to 0}[\text{softmax}_\beta(x)]_i = \lim_{\beta \to 0} \frac{\exp(x_i/\beta)}{|M|\exp(x_{\max}/\beta) + \sum_{j \notin M} \exp(x_j/\beta)}$$

$$= \lim_{\beta \to 0} \frac{\exp((x_i - x_{\max})/\beta)}{|M| + \sum_{j \notin M} \exp((x_j - x_{\max})/\beta)}$$

Since $x_i < x_{\max}$, we have $(x_i - x_{\max})/\beta \to -\infty$ as $\beta \to 0$, so $\exp((x_i - x_{\max})/\beta) \to 0$, giving:

$$\lim_{\beta \to 0}[\mathrm{softmax}_\beta(x)]_i = 0 \quad \text{for all } i \notin M \tag{97}$$

This proves that $\mathrm{softmax}_0(x)$ distributes probability mass uniformly over the indices achieving the maximum value of $x$. $\qquad\square$

### J.2. Omitted Proofs in Appendix F

*Proof of Theorem F.2.* We will prove the theorem by showing both directions of the inclusion: $\mathtt{FASP}[\phi_{\mathrm{PE}}; \mathcal{T}_{\mathrm{ACT}}] \subseteq \mathcal{H}_{\mathrm{TF}[\phi_{\mathrm{PE}}; \mathcal{T}_{\mathrm{ACT}}]}$ and $\mathcal{H}_{\mathrm{TF}[\phi_{\mathrm{PE}}; \mathcal{T}_{\mathrm{ACT}}]} \subseteq \mathtt{FASP}[\phi_{\mathrm{PE}}; \mathcal{T}_{\mathrm{ACT}}]$.

**Direction 1: $\mathtt{FASP}[\phi_{\mathrm{PE}}; \mathcal{T}_{\mathrm{ACT}}] \subseteq \mathcal{H}_{\mathrm{TF}[\phi_{\mathrm{PE}}; \mathcal{T}_{\mathrm{ACT}}]}$**   We show that any function definable in FASP can be implemented by a transformer. We prove this by induction on the number of steps in the FASP program. The base case is trivial as the initial set of definable functions $\mathcal{S}_0$ includes token embeddings $\mathcal{H}_{\mathrm{TE}}$ and positional embeddings $\phi_{\mathrm{PE}}$, which are directly implementable by transformer embedding layers, as established in Lemma F.3.

For the inductive step, assume that all functions in $\mathcal{S}_t$ can be implemented by transformers. Consider a new function $\psi_t$ defined at step $t$. We need to show that $\psi_t \in \mathcal{H}_{\mathrm{TF}[\phi_{\mathrm{PE}}; \mathcal{T}_{\mathrm{ACT}}]}$. There are four possible operators:

1. **Concatenation**: If $\psi_t = [\psi, \psi']$ where $\psi, \psi' \in \mathcal{S}_t$, then by the induction hypothesis, both $\psi$ and $\psi'$ can be implemented by transformers. By Lemma F.10, we know that concatenation is a closed operator over $\mathcal{H}_{\mathrm{TF}[\phi_{\mathrm{PE}}; \mathcal{T}_{\mathrm{ACT}}]}$, thus $\psi_t \in \mathcal{H}_{\mathrm{TF}[\phi_{\mathrm{PE}}; \mathcal{T}_{\mathrm{ACT}}]}$.

2. **Average-Hard Attention**: If $\psi_t = \mathtt{aha}(\psi, \psi', \psi'')$ where $\psi, \psi', \psi'' \in \mathcal{S}_t$, by Lemma F.12, average-hard attention is a closed operator over $\mathcal{H}_{\mathrm{TF}[\phi_{\mathrm{PE}}; \mathcal{T}_{\mathrm{ACT}}]}$. Therefore, $\psi_t \in \mathcal{H}_{\mathrm{TF}[\phi_{\mathrm{PE}}; \mathcal{T}_{\mathrm{ACT}}]}$.

3. **Linear Projection**: If $\psi_t = W \cdot \psi$ where $\psi \in \mathcal{S}_t$ and $W$ is a matrix, this defines a local operator as per Definition E.11. By Theorem F.11, any local operator implementable by a network with quadratic and ReLU activations is closed over $\mathcal{H}_{\mathrm{TF}[\phi_{\mathrm{PE}}; \mathcal{T}_{\mathrm{ACT}}]}$. Linear projection falls into this category, so $\psi_t \in \mathcal{H}_{\mathrm{TF}[\phi_{\mathrm{PE}}; \mathcal{T}_{\mathrm{ACT}}]}$.

4. **Nonlinear Activation**: If $\psi_t = \phi \circ \psi$ where $\phi \in \mathcal{T}_{\mathrm{ACT}}$ and $\psi \in \mathcal{S}_t$, this also defines a local operator. Since the activations in $\mathcal{T}_{\mathrm{ACT}}$ can be implemented by networks with quadratic and ReLU activations (as assumed in our framework), Theorem F.11 ensures that $\psi_t \in \mathcal{H}_{\mathrm{TF}[\phi_{\mathrm{PE}}; \mathcal{T}_{\mathrm{ACT}}]}$.

Thus, any function in $\mathtt{FASP}[\phi_{\mathrm{PE}}; \mathcal{T}_{\mathrm{ACT}}]$ can be implemented by a transformer, establishing that $\mathtt{FASP}[\phi_{\mathrm{PE}}; \mathcal{T}_{\mathrm{ACT}}] \subseteq \mathcal{H}_{\mathrm{TF}[\phi_{\mathrm{PE}}; \mathcal{T}_{\mathrm{ACT}}]}$.

**Direction 2: $\mathcal{H}_{\mathrm{TF}[\phi_{\mathrm{PE}}; \mathcal{T}_{\mathrm{ACT}}]} \subseteq \mathtt{FASP}[\phi_{\mathrm{PE}}; \mathcal{T}_{\mathrm{ACT}}]$**   We need to show that any transformer can be expressed as a FASP program. We prove this by induction on the number of layers in the transformer.

For the base case, a 0-layer transformer just consists of token and positional embeddings, which are already in the initial set of definable functions $\mathcal{S}_0$ in FASP.

For the inductive step, assume that any transformer with $L$ layers can be expressed in FASP. Consider a transformer with $L + 1$ layers. The first $L$ layers can be expressed in FASP by the induction hypothesis. Let's denote this as $\psi_L$. We need to show that adding the $(L + 1)$-th layer maintains expressibility in FASP.

The $(L + 1)$-th layer consists of a multi-head self-attention sublayer followed by a feed-forward network:

1. **Multi-Head Attention**: The multi-head attention can be decomposed into $h$ single-head attention, each of which can be expressed as $\mathtt{aha}(W_Q^i \cdot \psi_L, W_K^i \cdot \psi_L, W_V^i \cdot \psi_L)$ for $i \in \{1, \ldots, h\}$, where $W_Q^i, W_K^i$, and $W_V^i$ are the query, key, and value projection matrices for the $i$-th head. The outputs of these heads are concatenated and projected through $W_O$, which can be represented as a linear projection in FASP.

2. **Feed-Forward Network**: The feed-forward network applies a linear transformation followed by a nonlinear activation and another linear transformation. This can be directly expressed in FASP using the linear projection and nonlinear activation.

3. **Residual Connections**: The residual connections simply add the input to the output of each sublayer, which can be expressed as addition (which is a linear transformation) in FASP.

Therefore, any transformer with $L + 1$ layers can be expressed in FASP, establishing that $\mathcal{H}_{\text{TF}[\phi_{\text{PE}}; \mathcal{T}_{\text{ACT}}]} \subseteq \text{FASP}[\phi_{\text{PE}}; \mathcal{T}_{\text{ACT}}]$.

Combining the two directions, we have $\text{FASP}[\phi_{\text{PE}}; \mathcal{T}_{\text{ACT}}] = \mathcal{H}_{\text{TF}[\phi_{\text{PE}}; \mathcal{T}_{\text{ACT}}]}$, which completes the proof. $\qquad \square$

*Proof of Lemma F.4.* We prove each claim separately:

**(1) Token and Positional Embeddings:** For any $\psi_1, \psi_2 \in \mathcal{H}_{\text{TE}}$, let $\psi_1 : \Sigma \to \mathbb{R}^{d_1}$ and $\psi_2 : \Sigma \to \mathbb{R}^{d_2}$ be parameterized by $\theta_{\text{TE}}^1 \in (\mathbb{R}^{d_1})^\Sigma$ and $\theta_{\text{TE}}^2 \in (\mathbb{R}^{d_2})^\Sigma$ respectively. We define $[\psi_1, \psi_2] : \Sigma \to \mathbb{R}^{d_1 + d_2}$ parameterized by $\theta_{\text{TE}} \in (\mathbb{R}^{d_1 + d_2})^\Sigma$ where for each $\sigma \in \Sigma$, $\theta_{\text{TE}}(\sigma) = [\theta_{\text{TE}}^1(\sigma), \theta_{\text{TE}}^2(\sigma)]$. This directly implements the concatenation, showing that $[\psi_1, \psi_2] \in \mathcal{H}_{\text{TE}}$.

The case for positional embeddings follows similarly. For any $\psi_1, \psi_2 \in \mathcal{H}_{\text{PE}}$ with parameters $\theta_{\text{PE}}^1 \in \mathbb{R}^{d_1 \times d_{\text{PE}}}$ and $\theta_{\text{PE}}^2 \in \mathbb{R}^{d_2 \times d_{\text{PE}}}$, we can define $[\psi_1, \psi_2] \in \mathcal{H}_{\text{PE}}$ with parameters $\theta_{\text{PE}} = [\theta_{\text{PE}}^1; \theta_{\text{PE}}^2] \in \mathbb{R}^{(d_1 + d_2) \times d_{\text{PE}}}$.

**(2) Zero Function and Direct Sum with Zero:** The statement that $0 \in \mathcal{T}$ is straightforward as each operator allows setting all parameters (weight matrices and biases) to zero.

For $\phi \oplus 0_{d,d'} \in \mathcal{T}$, consider any $\phi \in \mathcal{T}$:

- For $\mathcal{T}_{\text{SA}}$: Given $\phi = \text{SA}_{\theta_{\text{SA}}}$ with parameters $\theta_{\text{SA}} = (W_Q, W_K, W_V, W_O)$, we define $\phi \oplus 0$ as $\text{SA}_{\theta'_{\text{SA}}}$ with parameters $\theta'_{\text{SA}} = (W'_Q, W'_K, W'_V, W'_O)$ where:

$$W'_Q = \begin{bmatrix} W_Q \\ \mathbf{0} \end{bmatrix}, \quad W'_K = \begin{bmatrix} W_K \\ \mathbf{0} \end{bmatrix}, \quad W'_V = \begin{bmatrix} W_V \\ \mathbf{0} \end{bmatrix}, \quad W'_O = \begin{bmatrix} W_O & \mathbf{0} \end{bmatrix}$$

- For $\mathcal{T}_{\text{MHA}}$: The proof follows from the fact that $\mathcal{T}_{\text{MHA}}$ is composed of multiple $\mathcal{T}_{\text{SA}}$ attention heads.
- For $\mathcal{T}_{\text{FF}}$: it suffices to prove for the sub feedforward network corresponding to each activation $\sigma \in \mathcal{T}_{\text{ACT}}$. Given $\sigma : \mathbb{R}^k \to \mathbb{R}$ and $\phi = \text{FF}^\sigma_{\theta_{\text{FF},\sigma}}$ with parameters $\theta_{\text{FF},\sigma} = (W_i)_{i=0}^k$, we define $\phi \oplus 0$ as $\text{FF}_{\theta'_{\text{FF}}}$ with parameters $W'_i = \begin{bmatrix} W_i & \mathbf{0} \end{bmatrix}$.
- For $\mathcal{T}_{\text{PROJ}}$: Given $\phi = \text{PROJ}_{\theta_{\text{PROJ}}}$ with parameter $\theta_{\text{PROJ}} \in \mathbb{R}^{d_{\text{PROJ}} \times d}$, we define $\phi \oplus 0_{d,d'}$ as $\text{PROJ}_{\theta'_{\text{PROJ}}}$ with $\theta'_{\text{PROJ}} = \begin{bmatrix} \theta_{\text{PROJ}} & \mathbf{0} \end{bmatrix}$.

**(3) Closure Under Addition:** For any $\mathcal{T} \in \{\mathcal{T}_{\text{MHA}}, \mathcal{T}_{\text{FF}}, \mathcal{T}_{\text{PROJ}}\}$, we have $\mathcal{T} = \mathcal{T} + \mathcal{T}$:

- For $\mathcal{T}_{\text{MHA}}$: The sum $\phi_1 + \phi_2$ of two multi-head attention modules can be implemented by concatenating their attention heads into a single module with $h_1 + h_2$ heads.
- For $\mathcal{T}_{\text{FF}}$: The sum of two feed-forward networks can be implemented by doubling the intermediate dimension and summing their outputs through appropriate matrix concatenation.
- For $\mathcal{T}_{\text{PROJ}}$: The sum of two projection layers is simply implemented by adding their parameter matrices.
- By definition, $\mathcal{T}_{\text{MHA}}$ is the sum closure of $\mathcal{T}_{\text{SA}}$ since multi-head attention is the sum of outputs from single-head attention modules.

**(4) Direct Sum Closure:** For any set $\mathcal{T} \in \{\mathcal{T}_{\text{MHA}}, \mathcal{T}_{\text{FF}}, \mathcal{T}_{\text{PROJ}}, \{\text{id}_d \mid d \in \mathbb{N}\}\}$, for any $\phi_1 \in \mathcal{T}$ with input dimension $d_1$ and output dimension $d'_1$, and $\phi_2 \in \mathcal{T}$ with input dimension $d_2$ and output dimension $d'_2$, their direct sum $\phi_1 \oplus \phi_2 \in \mathcal{T}$. This can be proved by decomposing the direct sum as:

$$\phi_1 \oplus \phi_2 = (\phi_1 \oplus 0) + (0 \oplus \phi_2) \tag{98}$$

where $0$ represents the appropriate zero function. From claim (2), we know that $\phi_1 \oplus 0, 0 \oplus \phi_2 \in \mathcal{T}$, and from claim (3), we know that $\mathcal{T} = \mathcal{T} + \mathcal{T}$. Therefore, $\phi_1 \oplus \phi_2 \in \mathcal{T}$.

For the identity function, note that $\text{id}_d : (\mathbb{R}^d)^* \to \mathbb{R}^d$ can be implemented by any of the above operators with appropriate parameter choices. For instance, in $\mathcal{T}_{\text{MHA}}$, we can set each head to implement identity by using $W_Q = W_K = W_V = I$ and $W_O = I/h$ where $h$ is the number of heads. For $\mathcal{T}_{\text{FF}}$, we can set $W_0 = W_1 = 0$, $W_2 = 0$, $b_0 = b_1 = 0$, and $b_2 = 0$. The direct sum of identity functions remains an identity function: $\text{id}_{d_1} \oplus \text{id}_{d_2} = \text{id}_{d_1 + d_2}$, which is again implementable by

the same operators with appropriately sized parameters. For $\mathcal{T}_{\mathrm{TF}}$: Given any two transformer layers $\phi_1, \phi_2 \in \mathcal{T}_{\mathrm{TF}}$, where $\phi_1 : (\mathbb{R}^{d_1})^* \to \mathbb{R}^{d_1}$ and $\phi_2 : (\mathbb{R}^{d_2})^* \to \mathbb{R}^{d_2}$ with parameters $\theta_{\mathrm{MHA}}^{(1)}, \theta_{\mathrm{FF}}^{(1)}$ and $\theta_{\mathrm{MHA}}^{(2)}, \theta_{\mathrm{FF}}^{(2)}$ respectively, we need to show $\phi_1 \oplus \phi_2 \in \mathcal{T}_{\mathrm{TF}}$.

By definition of $\mathcal{T}_{\mathrm{TF}}$ and transformer layers (Definition 3.16), we have:

$$\phi_1 = \left( \mathrm{FF}_{\theta_{\mathrm{FF}}^{(1)}} + \mathrm{id}_{d_1} \right) \circ \left( \overline{\mathrm{MHA}_{\theta_{\mathrm{MHA}}^{(1)}}} + \overline{\mathrm{id}_{d_1}} \right) \tag{99}$$

$$\phi_2 = \left( \mathrm{FF}_{\theta_{\mathrm{FF}}^{(2)}} + \mathrm{id}_{d_2} \right) \circ \left( \overline{\mathrm{MHA}_{\theta_{\mathrm{MHA}}^{(2)}}} + \overline{\mathrm{id}_{d_2}} \right) \tag{100}$$

For the direct sum $\phi_1 \oplus \phi_2$, we have:

$$\phi_1 \oplus \phi_2 = \left( \left( \mathrm{FF}_{\theta_{\mathrm{FF}}^{(1)}} \oplus \mathrm{FF}_{\theta_{\mathrm{FF}}^{(2)}} \right) + \left( \mathrm{id}_{d_1} \oplus \mathrm{id}_{d_2} \right) \right) \circ \left( \left( \overline{\mathrm{MHA}_{\theta_{\mathrm{MHA}}^{(1)}}} \oplus \overline{\mathrm{MHA}_{\theta_{\mathrm{MHA}}^{(2)}}} \right) + \left( \overline{\mathrm{id}_{d_1}} \oplus \overline{\mathrm{id}_{d_2}} \right) \right) \tag{101}$$

$$= \left( \left( \mathrm{FF}_{\theta_{\mathrm{FF}}^{(1)}} \oplus \mathrm{FF}_{\theta_{\mathrm{FF}}^{(2)}} \right) + \mathrm{id}_{d_1 + d_2} \right) \circ \left( \left( \overline{\mathrm{MHA}_{\theta_{\mathrm{MHA}}^{(1)}}} \oplus \overline{\mathrm{MHA}_{\theta_{\mathrm{MHA}}^{(2)}}} \right) + \overline{\mathrm{id}_{d_1 + d_2}} \right) \tag{102}$$

From our earlier results: 1. $\mathrm{FF}_{\theta_{\mathrm{FF}}^{(1)}} \oplus \mathrm{FF}_{\theta_{\mathrm{FF}}^{(2)}} \in \mathcal{T}_{\mathrm{FF}}$ (claim 4) 2. $\mathrm{id}_{d_1} \oplus \mathrm{id}_{d_2} = \mathrm{id}_{d_1 + d_2}$ (claim 4) 3. $\mathrm{MHA}_{\theta_{\mathrm{MHA}}^{(1)}} \oplus \mathrm{MHA}_{\theta_{\mathrm{MHA}}^{(2)}} \in \mathcal{T}_{\mathrm{MHA}}$ (claim 4)

Therefore, $\phi_1 \oplus \phi_2$ can be expressed as a transformer layer, which means $\phi_1 \oplus \phi_2 \in \mathcal{T}_{\mathrm{TF}}$. $\qquad\square$

*Proof of Lemma F.10.* We need to prove that for any $\psi_1, \psi_2 \in \mathcal{H}_{\mathrm{TF}[\phi_{\mathrm{PE}}; \mathcal{T}_{\mathrm{ACT}}]}$, their concatenation $[\psi_1, \psi_2] \in \mathcal{H}_{\mathrm{TF}[\phi_{\mathrm{PE}}; \mathcal{T}_{\mathrm{ACT}}]}$. Let $\psi_1 : \Sigma^* \to \mathbb{R}^{d_1}$ and $\psi_2 : \Sigma^* \to \mathbb{R}^{d_2}$ be two sequence-to-embedding functions in $\mathcal{H}_{\mathrm{TF}[\phi_{\mathrm{PE}}; \mathcal{T}_{\mathrm{ACT}}]}$. By definition, for $i \in \{1, 2\}$, there exist token embedding $\mathrm{TE}_i \in \mathcal{H}_{\mathrm{TE}}$, positional embedding $\mathrm{PE}_i \in \mathcal{H}_{\mathrm{PE}}$, transformer layers $\mathrm{TF}_{i,\ell} \in \mathcal{T}_{\mathrm{TF}}$ for $\ell \in \{1, \dots, L_i\}$, and projection $\mathrm{PROJ}_i \in \mathcal{T}_{\mathrm{PROJ}}$ such that:

$$\psi_i = \mathrm{PROJ}_i \circ \left( \bigcirc_{\ell=1}^{L_i} \overline{\mathrm{TF}_{i,\ell}} \right) \circ \left( \overline{\mathrm{PE}_i} + \overline{\mathrm{TE}_i} \right) \tag{103}$$

Without loss of generality, we can assume $L_1 = L_2 = L$ (if not, we can pad the shallower transformer with identity layers since $\mathrm{id}_d \in \mathcal{T}_{\mathrm{TF}}$). We construct a transformer that computes $[\psi_1, \psi_2]$ as follows:

1. **Initial embedding layer:** By Lemma F.4(1), we construct token and positional embeddings $\mathrm{TE} = [\mathrm{TE}_1, \mathrm{TE}_2] \in \mathcal{H}_{\mathrm{TE}}$ and $\mathrm{PE} = [\mathrm{PE}_1, \mathrm{PE}_2] \in \mathcal{H}_{\mathrm{PE}}$.

2. **Transformer layers:** For each $\ell \in \{1, \dots, L\}$, we define $\mathrm{TF}_\ell = \mathrm{TF}_{1,\ell} \oplus \mathrm{TF}_{2,\ell} \in \mathcal{T}_{\mathrm{TF}}$ by Lemma F.4(4).

3. **Projection layer:** We define $\mathrm{PROJ} = \mathrm{PROJ}_1 \oplus \mathrm{PROJ}_2 \in \mathcal{T}_{\mathrm{PROJ}}$ by Lemma F.4(4).

Thus, $[\psi_1, \psi_2] = \mathrm{PROJ} \circ \left( \bigcirc_{\ell=1}^{L} \overline{\mathrm{TF}_\ell} \right) \circ \left( \overline{\mathrm{PE}} + \overline{\mathrm{TE}} \right)$ is expressible by a valid transformer with a constant number of layers, which proves $[\psi_1, \psi_2] \in \mathcal{H}_{\mathrm{TF}[\phi_{\mathrm{PE}}; \mathcal{T}_{\mathrm{ACT}}]}$. $\qquad\square$

*Proof of Lemma F.11.* First we claim that if $\phi_\omega$ can be implemented by a 2-layer feedforward network with ReGLU activation, then $\omega$ is closed over $\mathcal{H}_{\mathrm{TF}[\phi_{\mathrm{PE}}; \mathcal{T}_{\mathrm{ACT}}]}$. This is because for any $\psi_i \in \mathcal{H}_{\mathrm{TF}[\phi_{\mathrm{PE}}; \mathcal{T}_{\mathrm{ACT}}]}(d_i)$, we have $[\psi_1, \dots, \psi_k] \in \mathcal{H}_{\mathrm{TF}[\phi_{\mathrm{PE}}; \mathcal{T}_{\mathrm{ACT}}]}$ since concatenation is closed. Suppose $[\psi_1, \dots, \psi_k]$ can be expressed as:

$$[\psi_1, \dots, \psi_k] = \mathrm{PROJ}_{\theta_{\mathrm{PROJ}}} \circ \left( \bigcirc_{\ell=1}^{L} \overline{\mathrm{TF}_{\theta_{\mathrm{MHA}}^\ell, \theta_{\mathrm{FF}}^\ell}} \right) \circ \left( \overline{\mathrm{PE}} + \overline{\mathrm{TE}_{\theta_{\mathrm{TE}}}} \right) \tag{104}$$

Now, applying a 2-layer feedforward network $\phi_\omega$ to this concatenated output means:

$$\omega(\psi_1, \dots, \psi_k) = \phi_\omega([\psi_1, \dots, \psi_k]) \tag{105}$$

Adding a 2-layer feedforward network $\phi_\omega$ after this means:

$$\omega(\psi_1, \dots, \psi_k) = \phi_\omega \circ \mathrm{PROJ}_{\theta_{\mathrm{PROJ}}} \circ \left( \bigcirc_{\ell=1}^{L} \overline{\mathrm{TF}_{\theta_{\mathrm{MHA}}^\ell, \theta_{\mathrm{FF}}^\ell}} \right) \circ \left( \overline{\mathrm{PE}} + \overline{\mathrm{TE}_{\theta_{\mathrm{TE}}}} \right) \tag{106}$$

To prove this remains in $\mathcal{H}_{\mathrm{TF}[\phi_{\mathrm{PE}};\mathcal{T}_{\mathrm{ACT}}]}$, we can construct an additional transformer layer $\mathrm{TF}_{\theta_{\mathrm{MHA}}^{L+1},\theta_{\mathrm{FF}}^{L+1}}$ where: 1. $\theta_{\mathrm{MHA}}^{L+1}$ implements zero attention (all weights set to 0) 2. $\theta_{\mathrm{FF}}^{L+1}$ implements $\phi_\omega \circ \mathrm{PROJ}_{\theta_{\mathrm{PROJ}}}$

This is valid because the linear projection $\mathrm{PROJ}_{\theta_{\mathrm{PROJ}}}$ can be absorbed into the first layer of the feedforward network in $\phi_\omega$. Specifically, if $\phi_\omega$ has parameters $(W_0, W_1, W_2, b_0, b_1, b_2)$ and $\mathrm{PROJ}_{\theta_{\mathrm{PROJ}}}$ has parameter matrix $\theta_{\mathrm{PROJ}}$, then $\phi_\omega \circ \mathrm{PROJ}_{\theta_{\mathrm{PROJ}}}$ is equivalent to a feedforward network with parameters: $(W_0', W_1', W_2', b_0', b_1', b_2') = (W_0\theta_{\mathrm{PROJ}}, W_1\theta_{\mathrm{PROJ}}, W_2, b_0, b_1, b_2)$.

Therefore, $\omega(\psi_1, \ldots, \psi_k) = \mathrm{PROJ}_{\theta_{\mathrm{PROJ}}'} \circ \left(\bigcirc_{\ell=1}^{L+1} \overline{\mathrm{TF}_{\theta_{\mathrm{MHA}}^\ell,\theta_{\mathrm{FF}}^\ell}}\right) \circ \left(\overline{\mathrm{PE}} + \overline{\mathrm{TE}_{\theta_{\mathrm{TE}}}}\right) \in \mathcal{H}_{\mathrm{TF}[\phi_{\mathrm{PE}};\mathcal{T}_{\mathrm{ACT}}]}$ where $\theta_{\mathrm{FF}}^{L+1}$ implements the combined function $\phi_\omega \circ \mathrm{PROJ}_{\theta_{\mathrm{PROJ}}}$ and $\mathrm{PROJ}_{\theta_{\mathrm{PROJ}}'}$ is the identity projection. This completes the proof of the claim.

Since composition of closed operators remains closed, the above claim extends to any number of layers, which are just composition of 2-layer networks. This completes the proof of Lemma F.11. $\qquad\square$

# K. Example: Boolean Satisfiability (SAT)

## K.1. Chain-of-Thought

• **Prompt:**

```
<|startoftext|> ( 4 ∨ ¬ 3 ∨ ¬ 2 ) ∧ ( ¬ 4 ∨ ¬ 2 ∨ 1 ) ∧ ( ¬ 2 ∨ ¬ 1 ∨ 3 ) ∧ ( 3 ∨ 4 ∨ 2 ) ∧ ( 3 ∨ ¬ 2
    ↪  ∨ 4 ) ∧ ( ¬ 1 ∨ 4 ∨ 2 ) ∧ ( 1 ∨ ¬ 4 ∨ 2 ) ∧ ( 4 ∨ ¬ 2 ∨ 3 ) ∧ ( ¬ 4 ∨ 1 ∨ ¬ 3 ) ∧ ( 3 ∨ 2 ∨
    ↪  4 ) ∧ ( ¬ 3 ∨ 1 ∨ ¬ 4 ) ∧ ( ¬ 4 ∨ 2 ∨ ¬ 3 ) ∧ ( 2 ∨ 4 ∨ ¬ 3 ) ∧ ( 3 ∨ 4 ∨ 2 ) ∧ ( 3 ∨ ¬ 1 ∨ ¬
    ↪  4 ) ∧ ( ¬ 1 ∨ ¬ 4 ∨ ¬ 2 ) ∧ ( 2 ∨ 3 ∨ 4 ) <|endofprompt|>
```

• **Response:**

```
[CALL] Question: ( 4 ∨ ¬ 3 ∨ ¬ 2 ) ∧ ( ¬ 4 ∨ ¬ 2 ∨ 1 ) ∧ ( ¬ 2 ∨ ¬ 1 ∨ 3 ) ∧ ( 3 ∨ 4 ∨ 2 ) ∧ ( 3 ∨ ¬
    ↪  2 ∨ 4 ) ∧ ( ¬ 1 ∨ 4 ∨ 2 ) ∧ ( 1 ∨ ¬ 4 ∨ 2 ) ∧ ( 4 ∨ ¬ 2 ∨ 3 ) ∧ ( ¬ 4 ∨ 1 ∨ ¬ 3 ) ∧ ( 3 ∨ 2 ∨
    ↪  4 ) ∧ ( ¬ 3 ∨ 1 ∨ ¬ 4 ) ∧ ( ¬ 4 ∨ 2 ∨ ¬ 3 ) ∧ ( 2 ∨ 4 ∨ ¬ 3 ) ∧ ( 3 ∨ 4 ∨ 2 ) ∧ ( 3 ∨ ¬ 1 ∨
    ↪  ¬ 4 ) ∧ ( ¬ 1 ∨ ¬ 4 ∨ ¬ 2 ) ∧ ( 2 ∨ 3 ∨ 4 ) Try 1 = True [CALL] Question: ( 4 ∨ ¬ 3 ∨ ¬ 2 ) ∧
    ↪  ( ¬ 2 ∨ 3 ) ∧ ( 3 ∨ 4 ∨ 2 ) ∧ ( 3 ∨ ¬ 2 ∨ 4 ) ∧ ( 4 ∨ 2 ) ∧ ( 4 ∨ ¬ 2 ∨ 3 ) ∧ ( 3 ∨ 2 ∨ 4 )
    ↪  ∧ ( ¬ 4 ∨ 2 ∨ ¬ 3 ) ∧ ( 2 ∨ 4 ∨ ¬ 3 ) ∧ ( 3 ∨ 4 ∨ 2 ) ∧ ( 3 ∨ ¬ 4 ) ∧ ( ¬ 4 ∨ ¬ 2 ) ∧ ( 2 ∨ 3
    ↪  ∨ 4 ) Try 2 = True [CALL] Question: ( 4 ∨ ¬ 3 ) ∧ ( 3 ) ∧ ( 3 ∨ 4 ) ∧ ( 4 ∨ 3 ) ∧ ( 3 ∨ ¬ 4
    ↪  ) ∧ ( ¬ 4 ) Found 3 Let 3 = True [CALL] Question: ( 4 ) ∧ ( ¬ 4 ) Found 4 Let 4 = True [SEP]
    ↪  Answer: False [RETURN] [SEP] Answer: False [RETURN] Try 2 = False [CALL] Question: ( 3 ∨ 4 )
    ↪  ∧ ( 4 ) ∧ ( 3 ∨ 4 ) ∧ ( ¬ 4 ∨ ¬ 3 ) ∧ ( 4 ∨ ¬ 3 ) ∧ ( 3 ∨ 4 ) ∧ ( 3 ∨ ¬ 4 ) ∧ ( 3 ∨ 4 ) Found
    ↪  4 Let 4 = True [CALL] Question: ( ¬ 3 ) ∧ ( 3 ) Found ¬ 3 Let 3 = False [SEP] Answer: False
    ↪  [RETURN] [SEP] Answer: False [RETURN] [SEP] Answer: False [RETURN] Try 1 = False [CALL]
    ↪  Question: ( 4 ∨ ¬ 3 ∨ ¬ 2 ) ∧ ( ¬ 4 ∨ ¬ 2 ) ∧ ( 3 ∨ 4 ∨ 2 ) ∧ ( 3 ∨ ¬ 2 ∨ 4 ) ∧ ( ¬ 4 ∨ 2 ) ∧
    ↪  ( 4 ∨ ¬ 2 ∨ 3 ) ∧ ( ¬ 4 ∨ ¬ 3 ) ∧ ( 3 ∨ 2 ∨ 4 ) ∧ ( ¬ 3 ∨ ¬ 4 ) ∧ ( ¬ 4 ∨ 2 ∨ ¬ 3 ) ∧ ( 2 ∨
    ↪  4 ∨ ¬ 3 ) ∧ ( 3 ∨ 4 ∨ 2 ) ∧ ( 2 ∨ 3 ∨ 4 ) Try 2 = True [CALL] Question: ( 4 ∨ ¬ 3 ) ∧ ( ¬ 4 )
    ↪  ∧ ( 3 ∨ 4 ) ∧ ( 4 ∨ 3 ) ∧ ( ¬ 4 ∨ ¬ 3 ) ∧ ( ¬ 3 ∨ ¬ 4 ) Found ¬ 4 Let 4 = False [CALL]
    ↪  Question: ( ¬ 3 ) ∧ ( 3 ) ∧ ( 3 ) Found ¬ 3 Let 3 = False [SEP] Answer: False [RETURN] [SEP]
    ↪  Answer: False [RETURN] Try 2 = False [CALL] Question: ( 3 ∨ 4 ) ∧ ( ¬ 4 ) ∧ ( ¬ 4 ∨ ¬ 3 ) ∧ (
    ↪  3 ∨ 4 ) ∧ ( ¬ 3 ∨ ¬ 4 ) ∧ ( ¬ 4 ∨ ¬ 3 ) ∧ ( 4 ∨ ¬ 3 ) ∧ ( 3 ∨ 4 ) ∧ ( 3 ∨ 4 ) Found ¬ 4 Let
    ↪  4 = False [CALL] Question: ( 3 ) ∧ ( 3 ) ∧ ( ¬ 3 ) ∧ ( 3 ) ∧ ( 3 ) Found 3 Let 3 = True [SEP]
    ↪   Answer: False [RETURN] [SEP] Answer: False [RETURN] [SEP] Answer: False [RETURN] [SEP]
    ↪  Answer: False [RETURN] <|endoftext|>
```

## K.2. PENCIL

• **Prompt:**

```
<|startoftext|> ( 4 ∨ ¬ 3 ∨ ¬ 2 ) ∧ ( ¬ 4 ∨ ¬ 2 ∨ 1 ) ∧ ( ¬ 2 ∨ ¬ 1 ∨ 3 ) ∧ ( 3 ∨ 4 ∨ 2 ) ∧ ( 3 ∨ ¬ 2
    ↪  ∨ 4 ) ∧ ( ¬ 1 ∨ 4 ∨ 2 ) ∧ ( 1 ∨ ¬ 4 ∨ 2 ) ∧ ( 4 ∨ ¬ 2 ∨ 3 ) ∧ ( ¬ 4 ∨ 1 ∨ ¬ 3 ) ∧ ( 3 ∨ 2 ∨
    ↪  4 ) ∧ ( ¬ 3 ∨ 1 ∨ ¬ 4 ) ∧ ( ¬ 4 ∨ 2 ∨ ¬ 3 ) ∧ ( 2 ∨ 4 ∨ ¬ 3 ) ∧ ( 3 ∨ 4 ∨ 2 ) ∧ ( 3 ∨ ¬ 1 ∨ ¬
    ↪  4 ) ∧ ( ¬ 1 ∨ ¬ 4 ∨ ¬ 2 ) ∧ ( 2 ∨ 3 ∨ 4 ) <|endofprompt|>
```

• **Response:**

```
Answer: False <|endoftext|>
```

## K.3. Internal Thinking Process of PENCIL

### Model Generation (1)

```
[CALL] Question: ( ¬ 3 ∨ 4 ∨ 1 ) ∧ ( 1 ∨ 3 ∨ 2 ) ∧ ( ¬ 4 ∨ ¬ 3 ∨ ¬ 1 ) ∧ ( ¬ 3 ∨ ¬ 1 ∨ 2 ) ∧ ( 4 ∨ 1
    ↪  ∨ 3 ) ∧ ( 4 ∨ 1 ∨ ¬ 3 ) ∧ ( ¬ 3 ∨ ¬ 1 ∨ 4 ) ∧ ( ¬ 1 ∨ 2 ∨ ¬ 4 ) ∧ ( ¬ 3 ∨ ¬ 1 ∨ 4 ) ∧ ( ¬ 3 ∨
    ↪  2 ∨ ¬ 4 ) ∧ ( ¬ 4 ∨ ¬ 1 ∨ 3 ) ∧ ( 2 ∨ 1 ∨ ¬ 3 ) ∧ ( 1 ∨ 4 ∨ 3 ) ∧ ( 2 ∨ ¬ 3 ∨ 4 ) ∧ ( 2 ∨ ¬
    ↪  4 ∨ 1 ) ∧ ( 1 ∨ 3 ∨ 2 ) ∧ ( 4 ∨ 2 ∨ ¬ 3 ) Try 1 = True [CALL] Question: ( ¬ 4 ∨ ¬ 3 ) ∧ ( ¬ 3
    ↪  ∨ 2 ) ∧ ( ¬ 3 ∨ 4 ) ∧ ( 2 ∨ ¬ 4 ) ∧ ( ¬ 3 ∨ 4 ) ∧ ( ¬ 3 ∨ 2 ∨ ¬ 4 ) ∧ ( ¬ 4 ∨ 3 ) ∧ ( 2 ∨ ¬
    ↪  3 ∨ 4 ) ∧ ( 4 ∨ 2 ∨ ¬ 3 ) Try 2 = True [CALL] Question: ( ¬ 4 ∨ ¬ 3 ) ∧ ( ¬ 3 ∨ 4 ) ∧ ( ¬ 3 ∨
    ↪  4 ) ∧ ( ¬ 4 ∨ 3 ) Try 3 = True [CALL] Question: ( ¬ 4 ) ∧ ( 4 ) ∧ ( 4 ) Found ¬ 4 Let 4 =
    ↪  False [SEP] Answer: False [RETURN]
```

### Reduction Rule (1)

```
[CALL] Question: ( ¬ 3 ∨ 4 ∨ 1 ) ∧ ( 1 ∨ 3 ∨ 2 ) ∧ ( ¬ 4 ∨ ¬ 3 ∨ ¬ 1 ) ∧ ( ¬ 3 ∨ ¬ 1 ∨ 2 ) ∧ ( 4 ∨ 1
↪  ∨ 3 ) ∧ ( 4 ∨ 1 ∨ ¬ 3 ) ∧ ( ¬ 3 ∨ ¬ 1 ∨ 4 ) ∧ ( ¬ 1 ∨ 2 ∨ ¬ 4 ) ∧ ( ¬ 3 ∨ ¬ 1 ∨ 4 ) ∧ ( ¬ 3 ∨
↪   2 ∨ ¬ 4 ) ∧ ( ¬ 4 ∨ ¬ 1 ∨ 3 ) ∧ ( 2 ∨ 1 ∨ ¬ 3 ) ∧ ( 1 ∨ 4 ∨ 3 ) ∧ ( 2 ∨ ¬ 3 ∨ 4 ) ∧ ( 2 ∨ ¬
↪  4 ∨ 1 ) ∧ ( 1 ∨ 3 ∨ 2 ) ∧ ( 4 ∨ 2 ∨ ¬ 3 ) Try 1 = True [CALL] Question: ( ¬ 4 ∨ ¬ 3 ) ∧ ( ¬ 3
↪   ∨ 2 ) ∧ ( ¬ 3 ∨ 4 ) ∧ ( 2 ∨ ¬ 4 ) ∧ ( ¬ 3 ∨ 4 ) ∧ ( ¬ 3 ∨ 2 ∨ ¬ 4 ) ∧ ( ¬ 4 ∨ 3 ) ∧ ( 2 ∨ ¬
↪  3 ∨ 4 ) ∧ ( 4 ∨ 2 ∨ ¬ 3 ) Try 2 = True [CALL] Question: ( ¬ 4 ∨ ¬ 3 ) ∧ ( ¬ 3 ∨ 4 ) ∧ ( ¬ 3 ∨
↪   4 ) ∧ ( ¬ 4 ∨ 3 ) Try 3 = True Answer: False
```

## Model Generation (2)

```
[CALL] Question: ( ¬ 3 ∨ 4 ∨ 1 ) ∧ ( 1 ∨ 3 ∨ 2 ) ∧ ( ¬ 4 ∨ ¬ 3 ∨ ¬ 1 ) ∧ ( ¬ 3 ∨ ¬ 1 ∨ 2 ) ∧ ( 4 ∨ 1
↪  ∨ 3 ) ∧ ( 4 ∨ 1 ∨ ¬ 3 ) ∧ ( ¬ 3 ∨ ¬ 1 ∨ 4 ) ∧ ( ¬ 1 ∨ 2 ∨ ¬ 4 ) ∧ ( ¬ 3 ∨ ¬ 1 ∨ 4 ) ∧ ( ¬ 3 ∨
↪   2 ∨ ¬ 4 ) ∧ ( ¬ 4 ∨ ¬ 1 ∨ 3 ) ∧ ( 2 ∨ 1 ∨ ¬ 3 ) ∧ ( 1 ∨ 4 ∨ 3 ) ∧ ( 2 ∨ ¬ 3 ∨ 4 ) ∧ ( 2 ∨ ¬
↪  4 ∨ 1 ) ∧ ( 1 ∨ 3 ∨ 2 ) ∧ ( 4 ∨ 2 ∨ ¬ 3 ) Try 1 = True [CALL] Question: ( ¬ 4 ∨ ¬ 3 ) ∧ ( ¬ 3
↪   ∨ 2 ) ∧ ( ¬ 3 ∨ 4 ) ∧ ( 2 ∨ ¬ 4 ) ∧ ( ¬ 3 ∨ 4 ) ∧ ( ¬ 3 ∨ 2 ∨ ¬ 4 ) ∧ ( ¬ 4 ∨ 3 ) ∧ ( 2 ∨ ¬
↪  3 ∨ 4 ) ∧ ( 4 ∨ 2 ∨ ¬ 3 ) Try 2 = True [CALL] Question: ( ¬ 4 ∨ ¬ 3 ) ∧ ( ¬ 3 ∨ 4 ) ∧ ( ¬ 3 ∨
↪   4 ) ∧ ( ¬ 4 ∨ 3 ) Try 3 = True Answer: False Try 3 = False [CALL] Question: ( ¬ 4 ) Found ¬
↪  4 Let 4 = False [SEP] Answer: True [RETURN]
```

## Reduction Rule (2)

```
[CALL] Question: ( ¬ 3 ∨ 4 ∨ 1 ) ∧ ( 1 ∨ 3 ∨ 2 ) ∧ ( ¬ 4 ∨ ¬ 3 ∨ ¬ 1 ) ∧ ( ¬ 3 ∨ ¬ 1 ∨ 2 ) ∧ ( 4 ∨ 1
↪  ∨ 3 ) ∧ ( 4 ∨ 1 ∨ ¬ 3 ) ∧ ( ¬ 3 ∨ ¬ 1 ∨ 4 ) ∧ ( ¬ 1 ∨ 2 ∨ ¬ 4 ) ∧ ( ¬ 3 ∨ ¬ 1 ∨ 4 ) ∧ ( ¬ 3 ∨
↪   2 ∨ ¬ 4 ) ∧ ( ¬ 4 ∨ ¬ 1 ∨ 3 ) ∧ ( 2 ∨ 1 ∨ ¬ 3 ) ∧ ( 1 ∨ 4 ∨ 3 ) ∧ ( 2 ∨ ¬ 3 ∨ 4 ) ∧ ( 2 ∨ ¬
↪  4 ∨ 1 ) ∧ ( 1 ∨ 3 ∨ 2 ) ∧ ( 4 ∨ 2 ∨ ¬ 3 ) Try 1 = True [CALL] Question: ( ¬ 4 ∨ ¬ 3 ) ∧ ( ¬ 3
↪   ∨ 2 ) ∧ ( ¬ 3 ∨ 4 ) ∧ ( 2 ∨ ¬ 4 ) ∧ ( ¬ 3 ∨ 4 ) ∧ ( ¬ 3 ∨ 2 ∨ ¬ 4 ) ∧ ( ¬ 4 ∨ 3 ) ∧ ( 2 ∨ ¬
↪  3 ∨ 4 ) ∧ ( 4 ∨ 2 ∨ ¬ 3 ) Try 2 = True [CALL] Question: ( ¬ 4 ∨ ¬ 3 ) ∧ ( ¬ 3 ∨ 4 ) ∧ ( ¬ 3 ∨
↪   4 ) ∧ ( ¬ 4 ∨ 3 ) Try 3 = True Answer: False Try 3 = False Answer: True
```

## Model Generation (3)

```
[CALL] Question: ( ¬ 3 ∨ 4 ∨ 1 ) ∧ ( 1 ∨ 3 ∨ 2 ) ∧ ( ¬ 4 ∨ ¬ 3 ∨ ¬ 1 ) ∧ ( ¬ 3 ∨ ¬ 1 ∨ 2 ) ∧ ( 4 ∨ 1
↪  ∨ 3 ) ∧ ( 4 ∨ 1 ∨ ¬ 3 ) ∧ ( ¬ 3 ∨ ¬ 1 ∨ 4 ) ∧ ( ¬ 1 ∨ 2 ∨ ¬ 4 ) ∧ ( ¬ 3 ∨ ¬ 1 ∨ 4 ) ∧ ( ¬ 3 ∨
↪   2 ∨ ¬ 4 ) ∧ ( ¬ 4 ∨ ¬ 1 ∨ 3 ) ∧ ( 2 ∨ 1 ∨ ¬ 3 ) ∧ ( 1 ∨ 4 ∨ 3 ) ∧ ( 2 ∨ ¬ 3 ∨ 4 ) ∧ ( 2 ∨ ¬
↪  4 ∨ 1 ) ∧ ( 1 ∨ 3 ∨ 2 ) ∧ ( 4 ∨ 2 ∨ ¬ 3 ) Try 1 = True [CALL] Question: ( ¬ 4 ∨ ¬ 3 ) ∧ ( ¬ 3
↪   ∨ 2 ) ∧ ( ¬ 3 ∨ 4 ) ∧ ( 2 ∨ ¬ 4 ) ∧ ( ¬ 3 ∨ 4 ) ∧ ( ¬ 3 ∨ 2 ∨ ¬ 4 ) ∧ ( ¬ 4 ∨ 3 ) ∧ ( 2 ∨ ¬
↪  3 ∨ 4 ) ∧ ( 4 ∨ 2 ∨ ¬ 3 ) Try 2 = True [CALL] Question: ( ¬ 4 ∨ ¬ 3 ) ∧ ( ¬ 3 ∨ 4 ) ∧ ( ¬ 3 ∨
↪   4 ) ∧ ( ¬ 4 ∨ 3 ) Try 3 = True Answer: False Try 3 = False Answer: True [SEP] Answer: True [
↪  RETURN]
```

## Reduction Rule (3)

```
[CALL] Question: ( ¬ 3 ∨ 4 ∨ 1 ) ∧ ( 1 ∨ 3 ∨ 2 ) ∧ ( ¬ 4 ∨ ¬ 3 ∨ ¬ 1 ) ∧ ( ¬ 3 ∨ ¬ 1 ∨ 2 ) ∧ ( 4 ∨ 1
↪  ∨ 3 ) ∧ ( 4 ∨ 1 ∨ ¬ 3 ) ∧ ( ¬ 3 ∨ ¬ 1 ∨ 4 ) ∧ ( ¬ 1 ∨ 2 ∨ ¬ 4 ) ∧ ( ¬ 3 ∨ ¬ 1 ∨ 4 ) ∧ ( ¬ 3 ∨
↪   2 ∨ ¬ 4 ) ∧ ( ¬ 4 ∨ ¬ 1 ∨ 3 ) ∧ ( 2 ∨ 1 ∨ ¬ 3 ) ∧ ( 1 ∨ 4 ∨ 3 ) ∧ ( 2 ∨ ¬ 3 ∨ 4 ) ∧ ( 2 ∨ ¬
↪  4 ∨ 1 ) ∧ ( 1 ∨ 3 ∨ 2 ) ∧ ( 4 ∨ 2 ∨ ¬ 3 ) Try 1 = True [CALL] Question: ( ¬ 4 ∨ ¬ 3 ) ∧ ( ¬ 3
↪   ∨ 2 ) ∧ ( ¬ 3 ∨ 4 ) ∧ ( 2 ∨ ¬ 4 ) ∧ ( ¬ 3 ∨ 4 ) ∧ ( ¬ 3 ∨ 2 ∨ ¬ 4 ) ∧ ( ¬ 4 ∨ 3 ) ∧ ( 2 ∨ ¬
↪  3 ∨ 4 ) ∧ ( 4 ∨ 2 ∨ ¬ 3 ) Try 2 = True Answer: True
```

## Model Generation (4)

```
[CALL] Question: ( ¬ 3 ∨ 4 ∨ 1 ) ∧ ( 1 ∨ 3 ∨ 2 ) ∧ ( ¬ 4 ∨ ¬ 3 ∨ ¬ 1 ) ∧ ( ¬ 3 ∨ ¬ 1 ∨ 2 ) ∧ ( 4 ∨ 1
↪  ∨ 3 ) ∧ ( 4 ∨ 1 ∨ ¬ 3 ) ∧ ( ¬ 3 ∨ ¬ 1 ∨ 4 ) ∧ ( ¬ 1 ∨ 2 ∨ ¬ 4 ) ∧ ( ¬ 3 ∨ ¬ 1 ∨ 4 ) ∧ ( ¬ 3 ∨
↪   2 ∨ ¬ 4 ) ∧ ( ¬ 4 ∨ ¬ 1 ∨ 3 ) ∧ ( 2 ∨ 1 ∨ ¬ 3 ) ∧ ( 1 ∨ 4 ∨ 3 ) ∧ ( 2 ∨ ¬ 3 ∨ 4 ) ∧ ( 2 ∨ ¬
↪  4 ∨ 1 ) ∧ ( 1 ∨ 3 ∨ 2 ) ∧ ( 4 ∨ 2 ∨ ¬ 3 ) Try 1 = True [CALL] Question: ( ¬ 4 ∨ ¬ 3 ) ∧ ( ¬ 3
↪   ∨ 2 ) ∧ ( ¬ 3 ∨ 4 ) ∧ ( 2 ∨ ¬ 4 ) ∧ ( ¬ 3 ∨ 4 ) ∧ ( ¬ 3 ∨ 2 ∨ ¬ 4 ) ∧ ( ¬ 4 ∨ 3 ) ∧ ( 2 ∨ ¬
↪  3 ∨ 4 ) ∧ ( 4 ∨ 2 ∨ ¬ 3 ) Try 2 = True Answer: True [SEP] Answer: True [RETURN]
```

## Reduction Rule (4)

```
[CALL] Question: ( ¬ 3 ∨ 4 ∨ 1 ) ∧ ( 1 ∨ 3 ∨ 2 ) ∧ ( ¬ 4 ∨ ¬ 3 ∨ ¬ 1 ) ∧ ( ¬ 3 ∨ ¬ 1 ∨ 2 ) ∧ ( 4 ∨ 1
↪  ∨ 3 ) ∧ ( 4 ∨ 1 ∨ ¬ 3 ) ∧ ( ¬ 3 ∨ ¬ 1 ∨ 4 ) ∧ ( ¬ 1 ∨ 2 ∨ ¬ 4 ) ∧ ( ¬ 3 ∨ ¬ 1 ∨ 4 ) ∧ ( ¬ 3 ∨
↪   2 ∨ ¬ 4 ) ∧ ( ¬ 4 ∨ ¬ 1 ∨ 3 ) ∧ ( 2 ∨ 1 ∨ ¬ 3 ) ∧ ( 1 ∨ 4 ∨ 3 ) ∧ ( 2 ∨ ¬ 3 ∨ 4 ) ∧ ( 2 ∨ ¬
↪  4 ∨ 1 ) ∧ ( 1 ∨ 3 ∨ 2 ) ∧ ( 4 ∨ 2 ∨ ¬ 3 ) Try 1 = True Answer: True
```

**Model Generation (5)**

```
[CALL] Question: ( ¬ 3 ∨ 4 ∨ 1 ) ∧ ( 1 ∨ 3 ∨ 2 ) ∧ ( ¬ 4 ∨ ¬ 3 ∨ ¬ 1 ) ∧ ( ¬ 3 ∨ ¬ 1 ∨ 2 ) ∧ ( 4 ∨ 1
   ↪   ∨ 3 ) ∧ ( 4 ∨ 1 ∨ ¬ 3 ) ∧ ( ¬ 3 ∨ ¬ 1 ∨ 4 ) ∧ ( ¬ 1 ∨ 2 ∨ ¬ 4 ) ∧ ( ¬ 3 ∨ ¬ 1 ∨ 4 ) ∧ ( ¬ 3 ∨
   ↪   2 ∨ ¬ 4 ) ∧ ( ¬ 4 ∨ ¬ 1 ∨ 3 ) ∧ ( 2 ∨ 1 ∨ ¬ 3 ) ∧ ( 1 ∨ 4 ∨ 3 ) ∧ ( 2 ∨ ¬ 3 ∨ 4 ) ∧ ( 2 ∨ ¬
   ↪   4 ∨ 1 ) ∧ ( 1 ∨ 3 ∨ 2 ) ∧ ( 4 ∨ 2 ∨ ¬ 3 ) Try 1 = True Answer: True [SEP] Answer: True [
   ↪   RETURN]
```

**Reduction Rule (5)**

```
Answer: True
```

**Model Generation (Final Response)**

```
Answer: True <|endoftext|>
```

# L. Example: Quantified Boolean Formula (QBF)

## L.1. Chain-of-Thought

● **Prompt:**

```
<|startoftext|> ∀ 3 ∀ 4 ∃ 1 ∀ 2 : #1 ( 3 ∨ ¬ 3 ) #2 ( ¬ 3 ∨ 3 ∨ 1 ) #3 ( 3 ∨ 1 ) #4 ( ¬ 1 ∨ ¬ 3 ∨ ¬ 4
   ↪   ) #5 ( 1 ∨ ¬ 1 ) #6 ( ¬ 4 ∨ 4 ∨ 3 ) #7 ( ¬ 4 ∨ 4 ) #8 ( 4 ∨ 1 ) <|endofprompt|>
```

● **Response:**

```
[CALL] Question: prefix_from ∀ 3 Try 3 = False [CALL] Question: prefix_from ∀ 4 Try 4 = False [CALL] Question: prefix_from ∃ 1 Try 1 = False [CALL]
   ↪   Question: prefix_from ∀ 2 Try 2 = False [CALL] Question: evaluate 1 = False 2 = False 3 = False 4 = False Check #0 ( 3 ∨ ¬ 3 ) True
   ↪   Check #1 ( ¬ 3 ∨ 3 ∨ 1 ) True Check #2 ( 3 ∨ 1 ) False [SEP] Answer: False [RETURN] [SEP] Answer: False [RETURN] Try 1 = True [CALL]
   ↪   Question: prefix_from ∀ 2 Try 2 = False [CALL] Question: evaluate 1 = True 2 = False 3 = False 4 = False Check #0 ( 3 ∨ ¬ 3 ) True Check
   ↪   #1 ( ¬ 3 ∨ 3 ∨ 1 ) True Check #2 ( 3 ∨ 1 ) True Check #3 ( ¬ 1 ∨ ¬ 3 ∨ ¬ 4 ) True Check #4 ( 1 ∨ ¬ 1 ) True Check #5 ( ¬ 4 ∨ 4 ∨ 3 ) True
   ↪   Check #6 ( ¬ 4 ∨ 4 ) Check #7 ( 4 ∨ 1 ) True Formula = True [SEP] Answer: True [RETURN] Try 2 = True [CALL] Question: evaluate 1 =
   ↪   True 2 = True 3 = False 4 = False Check #0 ( 3 ∨ ¬ 3 ) True Check #1 ( ¬ 3 ∨ 3 ∨ 1 ) True Check #2 ( 3 ∨ 1 ) True Check #3 ( ¬ 1 ∨ ¬ 3 ∨
   ↪   ¬ 4 ) True Check #4 ( 1 ∨ ¬ 1 ) True Check #5 ( ¬ 4 ∨ 4 ∨ 3 ) True Check #6 ( ¬ 4 ∨ 4 ) True Check #7 ( 4 ∨ 1 ) True Formula = True [SEP]
   ↪   Answer: True [RETURN] [SEP] Answer: True [RETURN] [SEP] Answer: True [RETURN] Try 4 = True [CALL] Question: prefix_from ∃ 1 Try 1 =
   ↪   False [CALL] Question: prefix_from ∀ 2 Try 2 = False [CALL] Question: evaluate 1 = False 2 = False 3 = False 4 = True Check #0 ( 3 ∨ ¬ 3
   ↪   ) True Check #1 ( ¬ 3 ∨ 3 ∨ 1 ) True Check #2 ( 3 ∨ 1 ) False [SEP] Answer: False [RETURN] [SEP] Answer: False [RETURN] Try 1 = True [
   ↪   CALL] Question: prefix_from ∀ 2 Try 2 = False [CALL] Question: evaluate 1 = True 2 = False 3 = False 4 = True Check #0 ( 3 ∨ ¬ 3 ) True
   ↪   Check #1 ( ¬ 3 ∨ 3 ∨ 1 ) True Check #2 ( 3 ∨ 1 ) True Check #3 ( ¬ 1 ∨ ¬ 3 ∨ ¬ 4 ) True Check #4 ( 1 ∨ ¬ 1 ) True Check #5 ( ¬ 4 ∨ 4 ∨ 3
   ↪   ) True Check #6 ( ¬ 4 ∨ 4 ) True Check #7 ( 4 ∨ 1 ) True Formula = True [SEP] Answer: True [RETURN] Try 2 = True [CALL] Question:
   ↪   evaluate 1 = True 2 = True 3 = False 4 = True Check #0 ( 3 ∨ ¬ 3 ) True Check #1 ( ¬ 3 ∨ 3 ∨ 1 ) True Check #2 ( 3 ∨ 1 ) True Check #3 (
   ↪   ¬ 1 ∨ ¬ 3 ∨ ¬ 4 ) True Check #4 ( 1 ∨ ¬ 1 ) True Check #5 ( ¬ 4 ∨ 4 ∨ 3 ) True Check #6 ( ¬ 4 ∨ 4 ) True Check #7 ( 4 ∨ 1 ) True Formula
   ↪   = True [SEP] Answer: True [RETURN] [SEP] Answer: True [RETURN] [SEP] Answer: True [RETURN] [SEP] Answer: True [RETURN] Try 3 = True [CALL
   ↪   ] Question: prefix_from ∀ 4 Try 4 = False [CALL] Question: prefix_from ∃ 1 Try 1 = False [CALL] Question: prefix_from ∀ 2 Try 2 = False [
   ↪   CALL] Question: evaluate 1 = False 2 = False 3 = True 4 = False Check #0 ( 3 ∨ ¬ 3 ) True Check #1 ( ¬ 3 ∨ 3 ∨ 1 ) True Check #2 ( 3 ∨ 1
   ↪   ) True Check #3 ( ¬ 1 ∨ ¬ 3 ∨ ¬ 4 ) True Check #4 ( 1 ∨ ¬ 1 ) True Check #5 ( ¬ 4 ∨ 4 ∨ 3 ) True Check #6 ( ¬ 4 ∨ 4 ) True Check #7 ( 4 ∨
   ↪   1 ) False [SEP] Answer: False [RETURN] Try 1 = True [CALL] Question: prefix_from ∀ 2 Try 2 = False [CALL]
   ↪   Question: evaluate 1 = True 2 = False 3 = True 4 = False Check #0 ( 3 ∨ ¬ 3 ) True Check #1 ( ¬ 3 ∨ 3 ∨ 1 ) True Check #2 ( 3 ∨ 1 ) True
   ↪   Check #3 ( ¬ 1 ∨ ¬ 3 ∨ ¬ 4 ) True Check #4 ( 1 ∨ ¬ 1 ) True Check #5 ( ¬ 4 ∨ 4 ∨ 3 ) True Check #6 ( ¬ 4 ∨ 4 ) True Check #7 ( 4 ∨ 1 )
   ↪   True Formula = True [SEP] Answer: True [RETURN] Try 2 = True [CALL] Question: evaluate 1 = True 2 = True 3 = True 4 = False Check #0 ( 3
   ↪   ∨ ¬ 3 ) True Check #1 ( ¬ 3 ∨ 3 ∨ 1 ) True Check #2 ( 3 ∨ 1 ) True Check #3 ( ¬ 1 ∨ ¬ 3 ∨ ¬ 4 ) True Check #4 ( 1 ∨ ¬ 1 ) True Check #5 (
   ↪   ¬ 4 ∨ 4 ∨ 3 ) True Check #6 ( ¬ 4 ∨ 4 ) True Check #7 ( 4 ∨ 1 ) True Formula = True [SEP] Answer: True [RETURN] [SEP] Answer: True [
   ↪   RETURN] [SEP] Answer: True [RETURN] Try 4 = True [CALL] Question: prefix_from ∃ 1 Try 1 = False [CALL] Question: prefix_from ∀ 2 Try 2 =
   ↪   False [CALL] Question: evaluate 1 = False 2 = False 3 = True 4 = True Check #0 ( 3 ∨ ¬ 3 ) True Check #1 ( ¬ 3 ∨ 3 ∨ 1 ) True Check #2 (
   ↪   3 ∨ 1 ) True Check #3 ( ¬ 1 ∨ ¬ 3 ∨ ¬ 4 ) True Check #4 ( 1 ∨ ¬ 1 ) True Check #5 ( ¬ 4 ∨ 4 ∨ 3 ) True Check #6 ( ¬ 4 ∨ 4 ) True Check #7
   ↪   ( 4 ∨ 1 ) True Formula = True [SEP] Answer: True [RETURN] Try 1 = True [CALL] Question: evaluate 1 = False 2 = True 3 = True 4 = True
   ↪   Check #0 ( 3 ∨ ¬ 3 ) True Check #1 ( ¬ 3 ∨ 3 ∨ 1 ) True Check #2 ( 3 ∨ 1 ) True Check #3 ( ¬ 1 ∨ ¬ 3 ∨ ¬ 4 ) True Check #4 ( 1 ∨ ¬ 1 )
   ↪   True Check #5 ( ¬ 4 ∨ 4 ∨ 3 ) True Check #6 ( ¬ 4 ∨ 4 ) True Check #7 ( 4 ∨ 1 ) True Formula = True [SEP] Answer: True [RETURN] [SEP]
   ↪   Answer: True [RETURN] [SEP] Answer: True [RETURN] [SEP] Answer: True [RETURN] [SEP] Answer: True [RETURN] <|endoftext|>
```

## L.2. PENCIL

● **Prompt:**

```
<|startoftext|> ∀ 3 ∀ 4 ∃ 1 ∀ 2 : #1 ( 3 ∨ ¬ 3 ) #2 ( ¬ 3 ∨ 3 ∨ 1 ) #3 ( 3 ∨ 1 ) #4 ( ¬ 1 ∨ ¬ 3 ∨ ¬ 4
   ↪   ) #5 ( 1 ∨ ¬ 1 ) #6 ( ¬ 4 ∨ 4 ∨ 3 ) #7 ( ¬ 4 ∨ 4 ) #8 ( 4 ∨ 1 ) <|endofprompt|>
```

● **Response:**

```
Answer: True <|endoftext|>
```

## L.3. Internal Thinking Process of PENCIL

**Model Generation (1)**

```
[CALL] Question: prefix_from ∀ 3 Try 3 = False [CALL] Question: prefix_from ∀ 4 Try 4 = False [CALL]
  ↪ Question: prefix_from ∃ 1 Try 1 = False [CALL] Question: prefix_from ∀ 2 Try 2 = False [CALL]
  ↪  Question: evaluate 1 = False 2 = False 3 = False 4 = False Check #0 ( 3 ∨ ¬ 3 ) True Check
  ↪ #1 ( ¬ 3 ∨ 3 ∨ 1 ) True Check #2 ( 3 ∨ 1 ) False [SEP] Answer: False [RETURN]
```

**Reduction Rule (1)**

```
[CALL] Question: prefix_from ∀ 3 Try 3 = False [CALL] Question: prefix_from ∀ 4 Try 4 = False [CALL]
  ↪ Question: prefix_from ∃ 1 Try 1 = False [CALL] Question: prefix_from ∀ 2 Try 2 = False Answer
  ↪ : False
```

**Model Generation (2)**

```
[CALL] Question: prefix_from ∀ 3 Try 3 = False [CALL] Question: prefix_from ∀ 4 Try 4 = False [CALL]
  ↪ Question: prefix_from ∃ 1 Try 1 = False [CALL] Question: prefix_from ∀ 2 Try 2 = False Answer
  ↪ : False [SEP] Answer: False [RETURN]
```

**Reduction Rule (2)**

```
[CALL] Question: prefix_from ∀ 3 Try 3 = False [CALL] Question: prefix_from ∀ 4 Try 4 = False [CALL]
  ↪ Question: prefix_from ∃ 1 Try 1 = False Answer: False
```

**Model Generation (3)**

```
[CALL] Question: prefix_from ∀ 3 Try 3 = False [CALL] Question: prefix_from ∀ 4 Try 4 = False [CALL]
  ↪ Question: prefix_from ∃ 1 Try 1 = False Answer: False Try 1 = True [CALL] Question:
  ↪ prefix_from ∀ 2 Try 2 = False [CALL] Question: evaluate 1 = True 2 = False 3 = False 4 =
  ↪ False Check #0 ( 3 ∨ ¬ 3 ) True Check #1 ( ¬ 3 ∨ 3 ∨ 1 ) True Check #2 ( 3 ∨ 1 ) True Check
  ↪ #3 ( ¬ 1 ∨ ¬ 3 ∨ ¬ 4 ) True Check #4 ( 1 ∨ ¬ 1 ) True Check #5 ( ¬ 4 ∨ 4 ∨ 3 ) True Check #6
  ↪ ( ¬ 4 ∨ 4 ) True Check #7 ( 4 ∨ 1 ) True Formula = True [SEP] Answer: True [RETURN]
```

**Reduction Rule (3)**

```
[CALL] Question: prefix_from ∀ 3 Try 3 = False [CALL] Question: prefix_from ∀ 4 Try 4 = False [CALL]
  ↪ Question: prefix_from ∃ 1 Try 1 = False Answer: False Try 1 = True [CALL] Question:
  ↪ prefix_from ∀ 2 Try 2 = False Answer: True
```

**Model Generation (4)**

```
[CALL] Question: prefix_from ∀ 3 Try 3 = False [CALL] Question: prefix_from ∀ 4 Try 4 = False [CALL]
  ↪ Question: prefix_from ∃ 1 Try 1 = False Answer: False Try 1 = True [CALL] Question:
  ↪ prefix_from ∀ 2 Try 2 = False Answer: True Try 2 = True [CALL] Question: evaluate 1 = True 2
  ↪ = True 3 = False 4 = False Check #0 ( 3 ∨ ¬ 3 ) True Check #1 ( ¬ 3 ∨ 3 ∨ 1 ) True Check #2 (
  ↪  3 ∨ 1 ) True Check #3 ( ¬ 1 ∨ ¬ 3 ∨ ¬ 4 ) True Check #4 ( 1 ∨ ¬ 1 ) True Check #5 ( ¬ 4 ∨ 4
  ↪ ∨ 3 ) True Check #6 ( ¬ 4 ∨ 4 ) True Check #7 ( 4 ∨ 1 ) True Formula = True [SEP] Answer:
  ↪ True [RETURN]
```

**Reduction Rule (4)**

```
[CALL] Question: prefix_from ∀ 3 Try 3 = False [CALL] Question: prefix_from ∀ 4 Try 4 = False [CALL]
  ↪ Question: prefix_from ∃ 1 Try 1 = False Answer: False Try 1 = True [CALL] Question:
  ↪ prefix_from ∀ 2 Try 2 = False Answer: True Try 2 = True Answer: True
```

**Model Generation (5)**

```
[CALL] Question: prefix_from ∀ 3 Try 3 = False [CALL] Question: prefix_from ∀ 4 Try 4 = False [CALL]
  ↪ Question: prefix_from ∃ 1 Try 1 = False Answer: False Try 1 = True [CALL] Question:
  ↪ prefix_from ∀ 2 Try 2 = False Answer: True Try 2 = True Answer: True [SEP] Answer: True [
  ↪ RETURN]
```

**Reduction Rule (5)**

```
[CALL] Question: prefix_from ∀ 3 Try 3 = False [CALL] Question: prefix_from ∀ 4 Try 4 = False [CALL]
  ↪ Question: prefix_from ∃ 1 Try 1 = False Answer: False Try 1 = True Answer: True
```

**Model Generation (6)**

```
[CALL] Question: prefix_from ∀ 3 Try 3 = False [CALL] Question: prefix_from ∀ 4 Try 4 = False [CALL]
  ↪ Question: prefix_from ∃ 1 Try 1 = False Answer: False Try 1 = True Answer: True [SEP] Answer:
  ↪  True [RETURN]
```

**Reduction Rule (6)**

```
[CALL] Question: prefix_from ∀ 3 Try 3 = False [CALL] Question: prefix_from ∀ 4 Try 4 = False Answer:
  ↪  True
```

**Model Generation (7)**

```
[CALL] Question: prefix_from ∀ 3 Try 3 = False [CALL] Question: prefix_from ∀ 4 Try 4 = False Answer:
  ↪  True Try 4 = True [CALL] Question: prefix_from ∃ 1 Try 1 = False [CALL] Question:
  ↪ prefix_from ∀ 2 Try 2 = False [CALL] Question: evaluate 1 = False 2 = False 3 = False 4 =
  ↪ True Check #0 ( 3 ∨ ¬ 3 ) True Check #1 ( ¬ 3 ∨ 3 ∨ 1 ) True Check #2 ( 3 ∨ 1 ) False [SEP]
  ↪ Answer: False [RETURN]
```

**Reduction Rule (7)**

```
[CALL] Question: prefix_from ∀ 3 Try 3 = False [CALL] Question: prefix_from ∀ 4 Try 4 = False Answer:
  ↪  True Try 4 = True [CALL] Question: prefix_from ∃ 1 Try 1 = False [CALL] Question:
  ↪ prefix_from ∀ 2 Try 2 = False Answer: False
```

**Model Generation (8)**

```
[CALL] Question: prefix_from ∀ 3 Try 3 = False [CALL] Question: prefix_from ∀ 4 Try 4 = False Answer:
  ↪  True Try 4 = True [CALL] Question: prefix_from ∃ 1 Try 1 = False [CALL] Question:
  ↪ prefix_from ∀ 2 Try 2 = False Answer: False [SEP] Answer: False [RETURN]
```

**Reduction Rule (8)**

```
[CALL] Question: prefix_from ∀ 3 Try 3 = False [CALL] Question: prefix_from ∀ 4 Try 4 = False Answer:
  ↪  True Try 4 = True [CALL] Question: prefix_from ∃ 1 Try 1 = False Answer: False
```

**Model Generation (9)**

```
[CALL] Question: prefix_from ∀ 3 Try 3 = False [CALL] Question: prefix_from ∀ 4 Try 4 = False Answer:
  ↪  True Try 4 = True [CALL] Question: prefix_from ∃ 1 Try 1 = False Answer: False Try 1 = True
  ↪ [CALL] Question: prefix_from ∀ 2 Try 2 = False [CALL] Question: evaluate 1 = True 2 = False 3
  ↪  = False 4 = True Check #0 ( 3 ∨ ¬ 3 ) True Check #1 ( ¬ 3 ∨ 3 ∨ 1 ) True Check #2 ( 3 ∨ 1 )
  ↪ True Check #3 ( ¬ 1 ∨ ¬ 3 ∨ ¬ 4 ) True Check #4 ( 1 ∨ ¬ 1 ) True Check #5 ( ¬ 4 ∨ 4 ∨ 3 )
  ↪ True Check #6 ( ¬ 4 ∨ 4 ) True Check #7 ( 4 ∨ 1 ) True Formula = True [SEP] Answer: True [
  ↪ RETURN]
```

**Reduction Rule (9)**

```
[CALL] Question: prefix_from ∀ 3 Try 3 = False [CALL] Question: prefix_from ∀ 4 Try 4 = False Answer:
  ↪  True Try 4 = True [CALL] Question: prefix_from ∃ 1 Try 1 = False Answer: False Try 1 = True
  ↪ [CALL] Question: prefix_from ∀ 2 Try 2 = False Answer: True
```

**Model Generation (10)**

```
[CALL] Question: prefix_from ∀ 3 Try 3 = False [CALL] Question: prefix_from ∀ 4 Try 4 = False Answer:
  ↪  True Try 4 = True [CALL] Question: prefix_from ∃ 1 Try 1 = False Answer: False Try 1 = True
  ↪ [CALL] Question: prefix_from ∀ 2 Try 2 = False Answer: True Try 2 = True [CALL] Question:
  ↪ evaluate 1 = True 2 = True 3 = False 4 = True Check #0 ( 3 ∨ ¬ 3 ) True Check #1 ( ¬ 3 ∨ 3 ∨
  ↪ 1 ) True Check #2 ( 3 ∨ 1 ) True Check #3 ( ¬ 1 ∨ ¬ 3 ∨ ¬ 4 ) True Check #4 ( 1 ∨ ¬ 1 ) True
  ↪ Check #5 ( ¬ 4 ∨ 4 ∨ 3 ) True Check #6 ( ¬ 4 ∨ 4 ) True Check #7 ( 4 ∨ 1 ) True Formula =
  ↪ True [SEP] Answer: True [RETURN]
```

**Reduction Rule (10)**

```
[CALL] Question: prefix_from ∀ 3 Try 3 = False [CALL] Question: prefix_from ∀ 4 Try 4 = False Answer:
    ↪  True Try 4 = True [CALL] Question: prefix_from ∃ 1 Try 1 = False Answer: False Try 1 = True
    ↪  [CALL] Question: prefix_from ∀ 2 Try 2 = False Answer: True Try 2 = True Answer: True
```

**Model Generation (11)**

```
[CALL] Question: prefix_from ∀ 3 Try 3 = False [CALL] Question: prefix_from ∀ 4 Try 4 = False Answer:
    ↪  True Try 4 = True [CALL] Question: prefix_from ∃ 1 Try 1 = False Answer: False Try 1 = True
    ↪  [CALL] Question: prefix_from ∀ 2 Try 2 = False Answer: True Try 2 = True Answer: True [SEP]
    ↪  Answer: True [RETURN]
```

**Reduction Rule (11)**

```
[CALL] Question: prefix_from ∀ 3 Try 3 = False [CALL] Question: prefix_from ∀ 4 Try 4 = False Answer:
    ↪  True Try 4 = True [CALL] Question: prefix_from ∃ 1 Try 1 = False Answer: False Try 1 = True
    ↪  Answer: True
```

**Model Generation (12)**

```
[CALL] Question: prefix_from ∀ 3 Try 3 = False [CALL] Question: prefix_from ∀ 4 Try 4 = False Answer:
    ↪  True Try 4 = True [CALL] Question: prefix_from ∃ 1 Try 1 = False Answer: False Try 1 = True
    ↪  Answer: True [SEP] Answer: True [RETURN]
```

**Reduction Rule (12)**

```
[CALL] Question: prefix_from ∀ 3 Try 3 = False [CALL] Question: prefix_from ∀ 4 Try 4 = False Answer:
    ↪  True Try 4 = True Answer: True
```

**Model Generation (13)**

```
[CALL] Question: prefix_from ∀ 3 Try 3 = False [CALL] Question: prefix_from ∀ 4 Try 4 = False Answer:
    ↪  True Try 4 = True Answer: True [SEP] Answer: True [RETURN]
```

**Reduction Rule (13)**

```
[CALL] Question: prefix_from ∀ 3 Try 3 = False Answer: True
```

**Model Generation (14)**

```
[CALL] Question: prefix_from ∀ 3 Try 3 = False Answer: True Try 3 = True [CALL] Question: prefix_from
    ↪  ∀ 4 Try 4 = False [CALL] Question: prefix_from ∃ 1 Try 1 = False [CALL] Question:
    ↪  prefix_from ∀ 2 Try 2 = False [CALL] Question: evaluate 1 = False 2 = False 3 = True 4 =
    ↪  False Check #0 ( 3 ∨ ¬ 3 ) True Check #1 ( ¬ 3 ∨ 3 ∨ 1 ) True Check #2 ( 3 ∨ 1 ) True Check
    ↪  #3 ( ¬ 1 ∨ ¬ 3 ∨ ¬ 4 ) True Check #4 ( 1 ∨ ¬ 1 ) True Check #5 ( ¬ 4 ∨ 4 ∨ 3 ) True Check #6
    ↪  ( ¬ 4 ∨ 4 ) True Check #7 ( 4 ∨ 1 ) False [SEP] Answer: False [RETURN]
```

**Reduction Rule (14)**

```
[CALL] Question: prefix_from ∀ 3 Try 3 = False Answer: True Try 3 = True [CALL] Question: prefix_from
    ↪  ∀ 4 Try 4 = False [CALL] Question: prefix_from ∃ 1 Try 1 = False [CALL] Question:
    ↪  prefix_from ∀ 2 Try 2 = False Answer: False
```

**Model Generation (15)**

```
[CALL] Question: prefix_from ∀ 3 Try 3 = False Answer: True Try 3 = True [CALL] Question: prefix_from
    ↪  ∀ 4 Try 4 = False [CALL] Question: prefix_from ∃ 1 Try 1 = False [CALL] Question:
    ↪  prefix_from ∀ 2 Try 2 = False Answer: False [SEP] Answer: False [RETURN]
```

**Reduction Rule (15)**

```
[CALL] Question: prefix_from ∀ 3 Try 3 = False Answer: True Try 3 = True [CALL] Question: prefix_from
    ↪  ∀ 4 Try 4 = False [CALL] Question: prefix_from ∃ 1 Try 1 = False Answer: False
```

## Model Generation (16)

```
[CALL] Question: prefix_from ∀ 3 Try 3 = False Answer: True Try 3 = True [CALL] Question: prefix_from
    ↪  ∀ 4 Try 4 = False [CALL] Question: prefix_from ∃ 1 Try 1 = False Answer: False Try 1 = True
    ↪  [CALL] Question: prefix_from ∀ 2 Try 2 = False [CALL] Question: evaluate 1 = True 2 = False 3
    ↪   = True 4 = False Check #0 ( 3 ∨ ¬ 3 ) True Check #1 ( ¬ 3 ∨ 3 ∨ 1 ) True Check #2 ( 3 ∨ 1 )
    ↪  True Check #3 ( ¬ 1 ∨ ¬ 3 ∨ ¬ 4 ) True Check #4 ( 1 ∨ ¬ 1 ) True Check #5 ( ¬ 4 ∨ 4 ∨ 3 )
    ↪  True Check #6 ( ¬ 4 ∨ 4 ) True Check #7 ( 4 ∨ 1 ) True Formula = True [SEP] Answer: True [
    ↪  RETURN]
```

## Reduction Rule (16)

```
[CALL] Question: prefix_from ∀ 3 Try 3 = False Answer: True Try 3 = True [CALL] Question: prefix_from
    ↪  ∀ 4 Try 4 = False [CALL] Question: prefix_from ∃ 1 Try 1 = False Answer: False Try 1 = True
    ↪  [CALL] Question: prefix_from ∀ 2 Try 2 = False Answer: True
```

## Model Generation (17)

```
[CALL] Question: prefix_from ∀ 3 Try 3 = False Answer: True Try 3 = True [CALL] Question: prefix_from
    ↪  ∀ 4 Try 4 = False [CALL] Question: prefix_from ∃ 1 Try 1 = False Answer: False Try 1 = True
    ↪  [CALL] Question: prefix_from ∀ 2 Try 2 = False Answer: True Try 2 = True [CALL] Question:
    ↪  evaluate 1 = True 2 = True 3 = True 4 = False Check #0 ( 3 ∨ ¬ 3 ) True Check #1 ( ¬ 3 ∨ 3 ∨
    ↪  1 ) True Check #2 ( 3 ∨ 1 ) True Check #3 ( ¬ 1 ∨ ¬ 3 ∨ ¬ 4 ) True Check #4 ( 1 ∨ ¬ 1 ) True
    ↪  Check #5 ( ¬ 4 ∨ 4 ∨ 3 ) True Check #6 ( ¬ 4 ∨ 4 ) True Check #7 ( 4 ∨ 1 ) True Formula =
    ↪  True [SEP] Answer: True [RETURN]
```

## Reduction Rule (17)

```
[CALL] Question: prefix_from ∀ 3 Try 3 = False Answer: True Try 3 = True [CALL] Question: prefix_from
    ↪  ∀ 4 Try 4 = False [CALL] Question: prefix_from ∃ 1 Try 1 = False Answer: False Try 1 = True
    ↪  [CALL] Question: prefix_from ∀ 2 Try 2 = False Answer: True Try 2 = True Answer: True
```

## Model Generation (18)

```
[CALL] Question: prefix_from ∀ 3 Try 3 = False Answer: True Try 3 = True [CALL] Question: prefix_from
    ↪  ∀ 4 Try 4 = False [CALL] Question: prefix_from ∃ 1 Try 1 = False Answer: False Try 1 = True
    ↪  [CALL] Question: prefix_from ∀ 2 Try 2 = False Answer: True Try 2 = True Answer: True [SEP]
    ↪  Answer: True [RETURN]
```

## Reduction Rule (18)

```
[CALL] Question: prefix_from ∀ 3 Try 3 = False Answer: True Try 3 = True [CALL] Question: prefix_from
    ↪  ∀ 4 Try 4 = False [CALL] Question: prefix_from ∃ 1 Try 1 = False Answer: False Try 1 = True
    ↪  Answer: True
```

## Model Generation (19)

```
[CALL] Question: prefix_from ∀ 3 Try 3 = False Answer: True Try 3 = True [CALL] Question: prefix_from
    ↪  ∀ 4 Try 4 = False [CALL] Question: prefix_from ∃ 1 Try 1 = False Answer: False Try 1 = True
    ↪  Answer: True [SEP] Answer: True [RETURN]
```

## Reduction Rule (19)

```
[CALL] Question: prefix_from ∀ 3 Try 3 = False Answer: True Try 3 = True [CALL] Question: prefix_from
    ↪  ∀ 4 Try 4 = False Answer: True
```

## Model Generation (20)

```
[CALL] Question: prefix_from ∀ 3 Try 3 = False Answer: True Try 3 = True [CALL] Question: prefix_from
    ↪  ∀ 4 Try 4 = False Answer: True Try 4 = True [CALL] Question: prefix_from ∃ 1 Try 1 = False [
    ↪ CALL] Question: prefix_from ∀ 2 Try 2 = False [CALL] Question: evaluate 1 = False 2 = False 3
    ↪  = True 4 = True Check #0 ( 3 ∨ ¬ 3 ) True Check #1 ( ¬ 3 ∨ 3 ∨ 1 ) True Check #2 ( 3 ∨ 1 )
    ↪ True Check #3 ( ¬ 1 ∨ ¬ 3 ∨ ¬ 4 ) True Check #4 ( 1 ∨ ¬ 1 ) True Check #5 ( ¬ 4 ∨ 4 ∨ 3 )
    ↪ True Check #6 ( ¬ 4 ∨ 4 ) True Check #7 ( 4 ∨ 1 ) True Formula = True [SEP] Answer: True [
    ↪ RETURN]
```

## Reduction Rule (20)

```
[CALL] Question: prefix_from ∀ 3 Try 3 = False Answer: True Try 3 = True [CALL] Question: prefix_from
    ↪  ∀ 4 Try 4 = False Answer: True Try 4 = True [CALL] Question: prefix_from ∃ 1 Try 1 = False [
    ↪ CALL] Question: prefix_from ∀ 2 Try 2 = False Answer: True
```

## Model Generation (21)

```
[CALL] Question: prefix_from ∀ 3 Try 3 = False Answer: True Try 3 = True [CALL] Question: prefix_from
    ↪  ∀ 4 Try 4 = False Answer: True Try 4 = True [CALL] Question: prefix_from ∃ 1 Try 1 = False [
    ↪ CALL] Question: prefix_from ∀ 2 Try 2 = False Answer: True Try 2 = True [CALL] Question:
    ↪ evaluate 1 = False 2 = True 3 = True 4 = True Check #0 ( 3 ∨ ¬ 3 ) True Check #1 ( ¬ 3 ∨ 3 ∨
    ↪ 1 ) True Check #2 ( 3 ∨ 1 ) True Check #3 ( ¬ 1 ∨ ¬ 3 ∨ ¬ 4 ) True Check #4 ( 1 ∨ ¬ 1 ) True
    ↪ Check #5 ( ¬ 4 ∨ 4 ∨ 3 ) True Check #6 ( ¬ 4 ∨ 4 ) True Check #7 ( 4 ∨ 1 ) True Formula =
    ↪ True [SEP] Answer: True [RETURN]
```

## Reduction Rule (21)

```
[CALL] Question: prefix_from ∀ 3 Try 3 = False Answer: True Try 3 = True [CALL] Question: prefix_from
    ↪  ∀ 4 Try 4 = False Answer: True Try 4 = True [CALL] Question: prefix_from ∃ 1 Try 1 = False [
    ↪ CALL] Question: prefix_from ∀ 2 Try 2 = False Answer: True Try 2 = True Answer: True
```

## Model Generation (22)

```
[CALL] Question: prefix_from ∀ 3 Try 3 = False Answer: True Try 3 = True [CALL] Question: prefix_from
    ↪  ∀ 4 Try 4 = False Answer: True Try 4 = True [CALL] Question: prefix_from ∃ 1 Try 1 = False [
    ↪ CALL] Question: prefix_from ∀ 2 Try 2 = False Answer: True Try 2 = True Answer: True [SEP]
    ↪ Answer: True [RETURN]
```

## Reduction Rule (22)

```
[CALL] Question: prefix_from ∀ 3 Try 3 = False Answer: True Try 3 = True [CALL] Question: prefix_from
    ↪  ∀ 4 Try 4 = False Answer: True Try 4 = True [CALL] Question: prefix_from ∃ 1 Try 1 = False
    ↪  Answer: True
```

## Model Generation (23)

```
[CALL] Question: prefix_from ∀ 3 Try 3 = False Answer: True Try 3 = True [CALL] Question: prefix_from
    ↪  ∀ 4 Try 4 = False Answer: True Try 4 = True [CALL] Question: prefix_from ∃ 1 Try 1 = False
    ↪  Answer: True [SEP] Answer: True [RETURN]
```

## Reduction Rule (23)

```
[CALL] Question: prefix_from ∀ 3 Try 3 = False Answer: True Try 3 = True [CALL] Question: prefix_from
    ↪  ∀ 4 Try 4 = False Answer: True Try 4 = True Answer: True
```

## Model Generation (24)

```
[CALL] Question: prefix_from ∀ 3 Try 3 = False Answer: True Try 3 = True [CALL] Question: prefix_from
    ↪  ∀ 4 Try 4 = False Answer: True Try 4 = True Answer: True [SEP] Answer: True [RETURN]
```

## Reduction Rule (24)

```
[CALL] Question: prefix_from ∀ 3 Try 3 = False Answer: True Try 3 = True Answer: True
```

**Model Generation (25)**

```
[CALL] Question: prefix_from ∀ 3 Try 3 = False Answer: True Try 3 = True Answer: True [SEP] Answer:
    ↪ True [RETURN]
```

**Reduction Rule (25)**

```
Answer: True
```

**Model Generation (Final Response)**

```
Answer: True <|endoftext|>
```

# M. Example: Einstein's Puzzle

## M.1. Chain-of-Thought

● **Prompt:**

```
<|startoftext|> Constraint#1 : the Green house is immediately to the right of the one who keeps Birds
 Constraint#2 : the Brit is immediately to the right of the German
 Constraint#3 : the one who keeps Dogs is the same house as the Red house
 Constraint#4 : the one who keeps Birds is immediately to the right of the Swede <|endofprompt|>
```

● **Response:**

```
[CALL] ====== Possible Assignments ======
 House#1
 Color category have 3 possibilities Blue Green Red
 Nationality category have 3 possibilities Brit German Swede
 Pet category have 3 possibilities Birds Dogs Fish
 House#2
 Color category have 3 possibilities Blue Green Red
 Nationality category have 3 possibilities Brit German Swede
 Pet category have 3 possibilities Birds Dogs Fish
 House#3
 Color category have 3 possibilities Blue Green Red
 Nationality category have 3 possibilities Brit German Swede
 Pet category have 3 possibilities Birds Dogs Fish
 Unsatisfied constraints are Constraint#1 Constraint#2 Constraint#3 Constraint#4
 => Puzzle not solved yet
 ====== Propagation ======
 Applying Constraint#1 [CALL]
 PHASE 1: Single-value logic for Green and Birds under RIGHT constraint
 PHASE 2: Handling relation Green RIGHT Birds
 Green is immediately RIGHT of Birds
 Removing Green from House#1 because Green can't be in the leftmost house if it's to the RIGHT of
     ↪ Birds
 Removing Birds from House#3 can't be in the rightmost house if it's to the LEFT of Green
 [SEP] House#1 Color category changed from 3 possibilities Blue Green Red to 2 possibilities Blue Red
 House#3 Pet category changed from 3 possibilities Birds Dogs Fish to 2 possibilities Dogs Fish [
     ↪ RETURN]
 Applying Constraint#2 [CALL]
 PHASE 1: Single-value logic for Brit and German under RIGHT constraint
 PHASE 2: Handling relation Brit RIGHT German
 Brit is immediately RIGHT of German
 Removing Brit from House#1 because Brit can't be in the leftmost house if it's to the RIGHT of
     ↪ German
 Removing German from House#3 can't be in the rightmost house if it's to the LEFT of Brit
 [SEP] House#1 Nationality category changed from 3 possibilities Brit German Swede to 2 possibilities
     ↪ German Swede
 House#3 Nationality category changed from 3 possibilities Brit German Swede to 2 possibilities Brit
     ↪ Swede [RETURN]
 Applying Constraint#3 [CALL]
 PHASE 1: Single-value logic for Dogs and Red under SAME constraint
```

```
PHASE 2: Handling relation Dogs SAME Red
Dogs must be in the SAME house as Red
[SEP] No changes from this constraint [RETURN]
Applying Constraint#4 [CALL]
PHASE 1: Single-value logic for Birds and Swede under RIGHT constraint
PHASE 2: Handling relation Birds RIGHT Swede
Birds is immediately RIGHT of Swede
Removing Birds from House#1 because Birds can't be in the leftmost house if it's to the RIGHT of
    ↪ Swede
Removing Swede from House#3 can't be in the rightmost house if it's to the LEFT of Birds
[SEP] House#3 Nationality category changed from 2 possibilities Brit Swede to 1 possibilities Brit
House#1 Pet category changed from 3 possibilities Birds Dogs Fish to 2 possibilities Dogs Fish [
    ↪ RETURN]
[SEP] [CALL] ====== Possible Assignments After Propagation ======
House#1
Color category have 2 possibilities Blue Red
Nationality category have 2 possibilities German Swede
Pet category have 2 possibilities Dogs Fish
House#2
Color category have 3 possibilities Blue Green Red
Nationality category have 3 possibilities Brit German Swede
Pet category have 3 possibilities Birds Dogs Fish
House#3
Color category have 3 possibilities Blue Green Red
Nationality category is Brit
Pet category have 2 possibilities Dogs Fish
Unsatisfied constraints are Constraint#1 Constraint#2 Constraint#3 Constraint#4 [RETURN]
=> Puzzle not solved yet
====== Branch ======
Branching on House#1 Color category with 2 possibilities Blue Red
Trying possibility Blue in House#1 Color category
[CALL] ====== Possible Assignments ======
House#1
Color category is Blue
Nationality category have 2 possibilities German Swede
Pet category have 2 possibilities Dogs Fish
House#2
Color category have 3 possibilities Blue Green Red
Nationality category have 3 possibilities Brit German Swede
Pet category have 3 possibilities Birds Dogs Fish
House#3
Color category have 3 possibilities Blue Green Red
Nationality category is Brit
Pet category have 2 possibilities Dogs Fish
Unsatisfied constraints are Constraint#1 Constraint#2 Constraint#3 Constraint#4
=> Puzzle not solved yet
====== Propagation ======
Applying Constraint#1 [CALL]
PHASE 1: Single-value logic for Green and Birds under RIGHT constraint
Removing Blue from House#2 Color category because Blue is pinned in another house
Removing Blue from House#3 Color category because Blue is pinned in another house
Forcing Birds in House#2 Pet category because it can only appear here
PHASE 2: Handling relation Green RIGHT Birds
Green is immediately RIGHT of Birds
Since Birds is pinned to House#2 , removing Green from House#2 because Green must be right of House
    ↪ #2
Placing Green in House#3 because Birds is pinned to House#2
[SEP] House#2 Color category changed from 3 possibilities Blue Green Red to 1 possibilities Red
House#3 Color category changed from 3 possibilities Blue Green Red to 1 possibilities Green
House#2 Pet category changed from 3 possibilities Birds Dogs Fish to 1 possibilities Birds [RETURN]
Remove Constraint#1 because it is satisfied
Applying Constraint#2 [CALL]
PHASE 1: Single-value logic for Brit and German under RIGHT constraint
Removing Brit from House#2 Nationality category because Brit is pinned in another house
PHASE 2: Handling relation Brit RIGHT German
Brit is immediately RIGHT of German
German must be exactly one house to the LEFT , removing from House#1
```

```
Placing German in House#2 because Brit is pinned to House#3
[SEP] House#1 Nationality category changed from 2 possibilities German Swede to 1 possibilities
    ↪ Swede
House#2 Nationality category changed from 3 possibilities Brit German Swede to 1 possibilities
    ↪ German [RETURN]
Remove Constraint#2 because it is satisfied
Applying Constraint#3 [CALL]
PHASE 1: Single-value logic for Dogs and Red under SAME constraint
PHASE 2: Handling relation Dogs SAME Red
Dogs must be in the SAME house as Red
Since Red is pinned to House#2 , removing Dogs from House#1
Since Red is pinned to House#2 , removing Dogs from House#3
House#2 can't hold Dogs since it can't hold Red
[SEP] House#2 Color category changed from 1 possibilities Red to 0 possibilities empty
House#1 Pet category changed from 2 possibilities Dogs Fish to 1 possibilities Fish
House#3 Pet category changed from 2 possibilities Dogs Fish to 1 possibilities Fish [RETURN]
[SEP] No Solution [RETURN]
Trying possibility Red in House#1 Color category
[CALL] ====== Possible Assignments ======
House#1
Color category is Red
Nationality category have 2 possibilities German Swede
Pet category have 2 possibilities Dogs Fish
House#2
Color category have 3 possibilities Blue Green Red
Nationality category have 3 possibilities Brit German Swede
Pet category have 3 possibilities Birds Dogs Fish
House#3
Color category have 3 possibilities Blue Green Red
Nationality category is Brit
Pet category have 2 possibilities Dogs Fish
Unsatisfied constraints are Constraint#1 Constraint#2 Constraint#3 Constraint#4
=> Puzzle not solved yet
====== Propagation ======
Applying Constraint#1 [CALL]
PHASE 1: Single-value logic for Green and Birds under RIGHT constraint
Removing Red from House#2 Color category because Red is pinned in another house
Removing Red from House#3 Color category because Red is pinned in another house
Forcing Birds in House#2 Pet category because it can only appear here
PHASE 2: Handling relation Green RIGHT Birds
Green is immediately RIGHT of Birds
Since Birds is pinned to House#2 , removing Green from House#2 because Green must be right of House
    ↪ #2
Placing Green in House#3 because Birds is pinned to House#2
[SEP] House#2 Color category changed from 3 possibilities Blue Green Red to 1 possibilities Blue
House#3 Color category changed from 3 possibilities Blue Green Red to 1 possibilities Green
House#2 Pet category changed from 3 possibilities Birds Dogs Fish to 1 possibilities Birds [RETURN]
Remove Constraint#1 because it is satisfied
Applying Constraint#2 [CALL]
PHASE 1: Single-value logic for Brit and German under RIGHT constraint
Removing Brit from House#2 Nationality category because Brit is pinned in another house
PHASE 2: Handling relation Brit RIGHT German
Brit is immediately RIGHT of German
German must be exactly one house to the LEFT , removing from House#1
Placing German in House#2 because Brit is pinned to House#3
[SEP] House#1 Nationality category changed from 2 possibilities German Swede to 1 possibilities
    ↪ Swede
House#2 Nationality category changed from 3 possibilities Brit German Swede to 1 possibilities
    ↪ German [RETURN]
Remove Constraint#2 because it is satisfied
Applying Constraint#3 [CALL]
PHASE 1: Single-value logic for Dogs and Red under SAME constraint
PHASE 2: Handling relation Dogs SAME Red
Dogs must be in the SAME house as Red
Placing Dogs in House#1 since Red is in this house
Since Red is pinned to House#1 , removing Dogs from House#3
[SEP] House#1 Pet category changed from 2 possibilities Dogs Fish to 1 possibilities Dogs
```

```
House#3 Pet category changed from 2 possibilities Dogs Fish to 1 possibilities Fish [RETURN]
Remove Constraint#3 because it is satisfied
Applying Constraint#4 [CALL]
PHASE 1: Single-value logic for Birds and Swede under RIGHT constraint
PHASE 2: Handling relation Birds RIGHT Swede
Birds is immediately RIGHT of Swede
[SEP] No changes from this constraint [RETURN]
Remove Constraint#4 because it is satisfied
[SEP] [CALL] ====== Possible Assignments After Propagation ======
House#1
Color category is Red
Nationality category is Swede
Pet category is Dogs
House#2
Color category is Blue
Nationality category is German
Pet category is Birds
House#3
Color category is Green
Nationality category is Brit
Pet category is Fish
Unsatisfied constraints are [RETURN]
=> Puzzle is solved
[SEP] Solution House#1
Color category is Red
Nationality category is Swede
Pet category is Dogs
House#2
Color category is Blue
Nationality category is German
Pet category is Birds
House#3
Color category is Green
Nationality category is Brit
Pet category is Fish [RETURN]
[SEP] Solution House#1
Color category is Red
Nationality category is Swede
Pet category is Dogs
House#2
Color category is Blue
Nationality category is German
Pet category is Birds
House#3
Color category is Green
Nationality category is Brit
Pet category is Fish [RETURN]
=> House#3 owns the Fish
=> the Brit owns the Fish <|endoftext|>
```

## M.2. PENCIL

● **Prompt:**

```
<|startoftext|> Constraint#1 : the Green house is immediately to the right of the one who keeps Birds
Constraint#2 : the Brit is immediately to the right of the German
Constraint#3 : the one who keeps Dogs is the same house as the Red house
Constraint#4 : the one who keeps Birds is immediately to the right of the Swede <|endofprompt|>
```

● **Response:**

```
Solution House#1
Color category is Red
Nationality category is Swede
Pet category is Dogs
House#2
```

```
Color category is Blue
Nationality category is German
Pet category is Birds
House#3
Color category is Green
Nationality category is Brit
Pet category is Fish
=> House#3 owns the Fish
=> the Brit owns the Fish <|endoftext|>
```

## M.3. Internal Thinking Process of PENCIL

### Model Generation (1)

```
[CALL] ====== Possible Assignments ======
House#1
Color category have 3 possibilities Blue Green Red
Nationality category have 3 possibilities Brit German Swede
Pet category have 3 possibilities Birds Dogs Fish
House#2
Color category have 3 possibilities Blue Green Red
Nationality category have 3 possibilities Brit German Swede
Pet category have 3 possibilities Birds Dogs Fish
House#3
Color category have 3 possibilities Blue Green Red
Nationality category have 3 possibilities Brit German Swede
Pet category have 3 possibilities Birds Dogs Fish
Unsatisfied constraints are Constraint#1 Constraint#2 Constraint#3 Constraint#4
=> Puzzle not solved yet
====== Propagation ======
Applying Constraint#1 [CALL]
PHASE 1: Single-value logic for Fish and Red under RIGHT constraint
PHASE 2: Handling relation Fish RIGHT Red
Fish is immediately RIGHT of Red
Removing Fish from House#1 because Fish can't be in the leftmost house if it's to the RIGHT of Red
Removing Red from House#3 can't be in the rightmost house if it's to the LEFT of Fish
[SEP] House#3 Color category changed from 3 possibilities Blue Green Red to 2 possibilities Blue
     ↪  Green
House#1 Pet category changed from 3 possibilities Birds Dogs Fish to 2 possibilities Birds Dogs [
     ↪  RETURN]
```

### Reduction Rule (1)

```
[CALL] ====== Possible Assignments ======
House#1
Color category have 3 possibilities Blue Green Red
Nationality category have 3 possibilities Brit German Swede
Pet category have 3 possibilities Birds Dogs Fish
House#2
Color category have 3 possibilities Blue Green Red
Nationality category have 3 possibilities Brit German Swede
Pet category have 3 possibilities Birds Dogs Fish
House#3
Color category have 3 possibilities Blue Green Red
Nationality category have 3 possibilities Brit German Swede
Pet category have 3 possibilities Birds Dogs Fish
Unsatisfied constraints are Constraint#1 Constraint#2 Constraint#3 Constraint#4
=> Puzzle not solved yet
====== Propagation ======
Applying Constraint#1 House#3 Color category changed from 3 possibilities Blue Green Red to 2
     ↪  possibilities Blue Green
House#1 Pet category changed from 3 possibilities Birds Dogs Fish to 2 possibilities Birds Dogs
```

### Model Generation (2)

```
[CALL] ====== Possible Assignments ======
House#1
Color category have 3 possibilities Blue Green Red
Nationality category have 3 possibilities Brit German Swede
Pet category have 3 possibilities Birds Dogs Fish
House#2
Color category have 3 possibilities Blue Green Red
Nationality category have 3 possibilities Brit German Swede
Pet category have 3 possibilities Birds Dogs Fish
House#3
Color category have 3 possibilities Blue Green Red
Nationality category have 3 possibilities Brit German Swede
Pet category have 3 possibilities Birds Dogs Fish
Unsatisfied constraints are Constraint#1 Constraint#2 Constraint#3 Constraint#4
=> Puzzle not solved yet
====== Propagation ======
Applying Constraint#1 House#3 Color category changed from 3 possibilities Blue Green Red to 2
    ↪ possibilities Blue Green
House#1 Pet category changed from 3 possibilities Birds Dogs Fish to 2 possibilities Birds Dogs
Applying Constraint#2 [CALL]
PHASE 1: Single-value logic for Green and Red under LEFT constraint
PHASE 2: Handling relation Green LEFT Red
Green is immediately LEFT of Red
Removing Green from House#3 because Green can't be in the rightmost house if it's to the LEFT of Red
Removing Red from House#1 because Red can't be in the leftmost house if it's to the RIGHT of Green
[SEP] House#1 Color category changed from 3 possibilities Blue Green Red to 2 possibilities Blue
    ↪ Green
House#3 Color category changed from 2 possibilities Blue Green to 1 possibilities Blue [RETURN]
```

### Reduction Rule (2)

```
[CALL] ====== Possible Assignments ======
House#1
Color category have 3 possibilities Blue Green Red
Nationality category have 3 possibilities Brit German Swede
Pet category have 3 possibilities Birds Dogs Fish
House#2
Color category have 3 possibilities Blue Green Red
Nationality category have 3 possibilities Brit German Swede
Pet category have 3 possibilities Birds Dogs Fish
House#3
Color category have 3 possibilities Blue Green Red
Nationality category have 3 possibilities Brit German Swede
Pet category have 3 possibilities Birds Dogs Fish
Unsatisfied constraints are Constraint#1 Constraint#2 Constraint#3 Constraint#4
=> Puzzle not solved yet
====== Propagation ======
Applying Constraint#1 House#3 Color category changed from 3 possibilities Blue Green Red to 2
    ↪ possibilities Blue Green
House#1 Pet category changed from 3 possibilities Birds Dogs Fish to 2 possibilities Birds Dogs
Applying Constraint#2 House#1 Color category changed from 3 possibilities Blue Green Red to 2
    ↪ possibilities Blue Green
House#3 Color category changed from 2 possibilities Blue Green to 1 possibilities Blue
```

### Model Generation (3)

```
[CALL] ====== Possible Assignments ======
House#1
Color category have 3 possibilities Blue Green Red
Nationality category have 3 possibilities Brit German Swede
Pet category have 3 possibilities Birds Dogs Fish
House#2
Color category have 3 possibilities Blue Green Red
Nationality category have 3 possibilities Brit German Swede
Pet category have 3 possibilities Birds Dogs Fish
House#3
Color category have 3 possibilities Blue Green Red
```

```
Nationality category have 3 possibilities Brit German Swede
Pet category have 3 possibilities Birds Dogs Fish
Unsatisfied constraints are Constraint#1 Constraint#2 Constraint#3 Constraint#4
=> Puzzle not solved yet
====== Propagation ======
Applying Constraint#1 House#3 Color category changed from 3 possibilities Blue Green Red to 2
     ↪  possibilities Blue Green
House#1 Pet category changed from 3 possibilities Birds Dogs Fish to 2 possibilities Birds Dogs
Applying Constraint#2 House#1 Color category changed from 3 possibilities Blue Green Red to 2
     ↪  possibilities Blue Green
House#3 Color category changed from 2 possibilities Blue Green to 1 possibilities Blue
Applying Constraint#3 [CALL]
PHASE 1: Single-value logic for Fish and Swede under RIGHT constraint
PHASE 2: Handling relation Fish RIGHT Swede
Fish is immediately RIGHT of Swede
Removing Swede from House#3 can't be in the rightmost house if it's to the LEFT of Fish
[SEP] House#3 Nationality category changed from 3 possibilities Brit German Swede to 2 possibilities
     ↪   Brit German [RETURN]
```

## Reduction Rule (3)

```
[CALL] ====== Possible Assignments ======
House#1
Color category have 3 possibilities Blue Green Red
Nationality category have 3 possibilities Brit German Swede
Pet category have 3 possibilities Birds Dogs Fish
House#2
Color category have 3 possibilities Blue Green Red
Nationality category have 3 possibilities Brit German Swede
Pet category have 3 possibilities Birds Dogs Fish
House#3
Color category have 3 possibilities Blue Green Red
Nationality category have 3 possibilities Brit German Swede
Pet category have 3 possibilities Birds Dogs Fish
Unsatisfied constraints are Constraint#1 Constraint#2 Constraint#3 Constraint#4
=> Puzzle not solved yet
====== Propagation ======
Applying Constraint#1 House#3 Color category changed from 3 possibilities Blue Green Red to 2
     ↪  possibilities Blue Green
House#1 Pet category changed from 3 possibilities Birds Dogs Fish to 2 possibilities Birds Dogs
Applying Constraint#2 House#1 Color category changed from 3 possibilities Blue Green Red to 2
     ↪  possibilities Blue Green
House#3 Color category changed from 2 possibilities Blue Green to 1 possibilities Blue
Applying Constraint#3 House#3 Nationality category changed from 3 possibilities Brit German Swede to
     ↪   2 possibilities Brit German
```

## Model Generation (4)

```
[CALL] ====== Possible Assignments ======
House#1
Color category have 3 possibilities Blue Green Red
Nationality category have 3 possibilities Brit German Swede
Pet category have 3 possibilities Birds Dogs Fish
House#2
Color category have 3 possibilities Blue Green Red
Nationality category have 3 possibilities Brit German Swede
Pet category have 3 possibilities Birds Dogs Fish
House#3
Color category have 3 possibilities Blue Green Red
Nationality category have 3 possibilities Brit German Swede
Pet category have 3 possibilities Birds Dogs Fish
Unsatisfied constraints are Constraint#1 Constraint#2 Constraint#3 Constraint#4
=> Puzzle not solved yet
====== Propagation ======
Applying Constraint#1 House#3 Color category changed from 3 possibilities Blue Green Red to 2
     ↪  possibilities Blue Green
House#1 Pet category changed from 3 possibilities Birds Dogs Fish to 2 possibilities Birds Dogs
```

```
Applying Constraint#2 House#1 Color category changed from 3 possibilities Blue Green Red to 2
    ↪ possibilities Blue Green
House#3 Color category changed from 2 possibilities Blue Green to 1 possibilities Blue
Applying Constraint#3 House#3 Nationality category changed from 3 possibilities Brit German Swede to
    ↪ 2 possibilities Brit German
Applying Constraint#4 [CALL]
PHASE 1: Single-value logic for Brit and Birds under LEFT constraint
PHASE 2: Handling relation Brit LEFT Birds
Brit is immediately LEFT of Birds
Removing Brit from House#3 because Brit can't be in the rightmost house if it's to the LEFT of Birds
Removing Birds from House#1 because Birds can't be in the leftmost house if it's to the RIGHT of
    ↪ Brit
[SEP] House#3 Nationality category changed from 2 possibilities Brit German to 1 possibilities
    ↪ German
House#1 Pet category changed from 2 possibilities Birds Dogs to 1 possibilities Dogs [RETURN]
```

## Reduction Rule (4)

```
[CALL] ====== Possible Assignments ======
House#1
Color category have 3 possibilities Blue Green Red
Nationality category have 3 possibilities Brit German Swede
Pet category have 3 possibilities Birds Dogs Fish
House#2
Color category have 3 possibilities Blue Green Red
Nationality category have 3 possibilities Brit German Swede
Pet category have 3 possibilities Birds Dogs Fish
House#3
Color category have 3 possibilities Blue Green Red
Nationality category have 3 possibilities Brit German Swede
Pet category have 3 possibilities Birds Dogs Fish
Unsatisfied constraints are Constraint#1 Constraint#2 Constraint#3 Constraint#4
=> Puzzle not solved yet
====== Propagation ======
Applying Constraint#1 House#3 Color category changed from 3 possibilities Blue Green Red to 2
    ↪ possibilities Blue Green
House#1 Pet category changed from 3 possibilities Birds Dogs Fish to 2 possibilities Birds Dogs
Applying Constraint#2 House#1 Color category changed from 3 possibilities Blue Green Red to 2
    ↪ possibilities Blue Green
House#3 Color category changed from 2 possibilities Blue Green to 1 possibilities Blue
Applying Constraint#3 House#3 Nationality category changed from 3 possibilities Brit German Swede to
    ↪ 2 possibilities Brit German
Applying Constraint#4 House#3 Nationality category changed from 2 possibilities Brit German to 1
    ↪ possibilities German
House#1 Pet category changed from 2 possibilities Birds Dogs to 1 possibilities Dogs
```

## Model Generation (5)

```
[CALL] ====== Possible Assignments ======
House#1
Color category have 3 possibilities Blue Green Red
Nationality category have 3 possibilities Brit German Swede
Pet category have 3 possibilities Birds Dogs Fish
House#2
Color category have 3 possibilities Blue Green Red
Nationality category have 3 possibilities Brit German Swede
Pet category have 3 possibilities Birds Dogs Fish
House#3
Color category have 3 possibilities Blue Green Red
Nationality category have 3 possibilities Brit German Swede
Pet category have 3 possibilities Birds Dogs Fish
Unsatisfied constraints are Constraint#1 Constraint#2 Constraint#3 Constraint#4
=> Puzzle not solved yet
====== Propagation ======
Applying Constraint#1 House#3 Color category changed from 3 possibilities Blue Green Red to 2
    ↪ possibilities Blue Green
House#1 Pet category changed from 3 possibilities Birds Dogs Fish to 2 possibilities Birds Dogs
```

```
Applying Constraint#2 House#1 Color category changed from 3 possibilities Blue Green Red to 2
    ↪ possibilities Blue Green
House#3 Color category changed from 2 possibilities Blue Green to 1 possibilities Blue
Applying Constraint#3 House#3 Nationality category changed from 3 possibilities Brit German Swede to
    ↪  2 possibilities Brit German
Applying Constraint#4 House#3 Nationality category changed from 2 possibilities Brit German to 1
    ↪ possibilities German
House#1 Pet category changed from 2 possibilities Birds Dogs to 1 possibilities Dogs
[SEP] [CALL] ====== Possible Assignments After Propagation ======
House#1
Color category have 2 possibilities Blue Green
Nationality category have 3 possibilities Brit German Swede
Pet category is Dogs
House#2
Color category have 3 possibilities Blue Green Red
Nationality category have 3 possibilities Brit German Swede
Pet category have 3 possibilities Birds Dogs Fish
House#3
Color category is Blue
Nationality category is German
Pet category have 3 possibilities Birds Dogs Fish
Unsatisfied constraints are Constraint#1 Constraint#2 Constraint#3 Constraint#4 [RETURN]
```

## Reduction Rule (5)

```
[CALL] ====== Possible Assignments After Propagation ======
House#1
Color category have 2 possibilities Blue Green
Nationality category have 3 possibilities Brit German Swede
Pet category is Dogs
House#2
Color category have 3 possibilities Blue Green Red
Nationality category have 3 possibilities Brit German Swede
Pet category have 3 possibilities Birds Dogs Fish
House#3
Color category is Blue
Nationality category is German
Pet category have 3 possibilities Birds Dogs Fish
Unsatisfied constraints are Constraint#1 Constraint#2 Constraint#3 Constraint#4
```

## Model Generation (6)

```
[CALL] ====== Possible Assignments After Propagation ======
House#1
Color category have 2 possibilities Blue Green
Nationality category have 3 possibilities Brit German Swede
Pet category is Dogs
House#2
Color category have 3 possibilities Blue Green Red
Nationality category have 3 possibilities Brit German Swede
Pet category have 3 possibilities Birds Dogs Fish
House#3
Color category is Blue
Nationality category is German
Pet category have 3 possibilities Birds Dogs Fish
Unsatisfied constraints are Constraint#1 Constraint#2 Constraint#3 Constraint#4
=> Puzzle not solved yet
====== Branch ======
Branching on House#1 Color category with 2 possibilities Blue Green
Trying possibility Green in House#1 Color category
[CALL] ====== Possible Assignments ======
House#1
Color category is Green
Nationality category have 3 possibilities Brit German Swede
Pet category is Dogs
House#2
Color category have 3 possibilities Blue Green Red
```

```
Nationality category have 3 possibilities Brit German Swede
Pet category have 3 possibilities Birds Dogs Fish
House#3
Color category is Blue
Nationality category is German
Pet category have 3 possibilities Birds Dogs Fish
Unsatisfied constraints are Constraint#1 Constraint#2 Constraint#3 Constraint#4
=> Puzzle not solved yet
====== Propagation ======
Applying Constraint#1 [CALL]
PHASE 1: Single-value logic for Fish and Red under RIGHT constraint
Removing Dogs from House#2 Pet category because Dogs is pinned in another house
Removing Dogs from House#3 Pet category because Dogs is pinned in another house
Removing Green from House#2 Color category because Green is pinned in another house
Removing Blue from House#2 Color category because Blue is pinned in another house
PHASE 2: Handling relation Fish RIGHT Red
Fish is immediately RIGHT of Red
Since Red is pinned to House#2 , removing Fish from House#2 because Fish must be right of House#2
Placing Fish in House#3 because Red is pinned to House#2
[SEP] House#2 Color category changed from 3 possibilities Blue Green Red to 1 possibilities Red
House#2 Pet category changed from 3 possibilities Birds Dogs Fish to 1 possibilities Birds
House#3 Pet category changed from 3 possibilities Birds Dogs Fish to 1 possibilities Fish [RETURN]
```

### Reduction Rule (6)

```
[CALL] ====== Possible Assignments After Propagation ======
House#1
Color category have 2 possibilities Blue Green
Nationality category have 3 possibilities Brit German Swede
Pet category is Dogs
House#2
Color category have 3 possibilities Blue Green Red
Nationality category have 3 possibilities Brit German Swede
Pet category have 3 possibilities Birds Dogs Fish
House#3
Color category is Blue
Nationality category is German
Pet category have 3 possibilities Birds Dogs Fish
Unsatisfied constraints are Constraint#1 Constraint#2 Constraint#3 Constraint#4
=> Puzzle not solved yet
====== Branch ======
Branching on House#1 Color category with 2 possibilities Blue Green
Trying possibility Green in House#1 Color category
[CALL] ====== Possible Assignments ======
House#1
Color category is Green
Nationality category have 3 possibilities Brit German Swede
Pet category is Dogs
House#2
Color category have 3 possibilities Blue Green Red
Nationality category have 3 possibilities Brit German Swede
Pet category have 3 possibilities Birds Dogs Fish
House#3
Color category is Blue
Nationality category is German
Pet category have 3 possibilities Birds Dogs Fish
Unsatisfied constraints are Constraint#1 Constraint#2 Constraint#3 Constraint#4
=> Puzzle not solved yet
====== Propagation ======
Applying Constraint#1 House#2 Color category changed from 3 possibilities Blue Green Red to 1
     ↪ possibilities Red
House#2 Pet category changed from 3 possibilities Birds Dogs Fish to 1 possibilities Birds
House#3 Pet category changed from 3 possibilities Birds Dogs Fish to 1 possibilities Fish
```

### Model Generation (7)

```
[CALL] ====== Possible Assignments After Propagation ======
House#1
Color category have 2 possibilities Blue Green
Nationality category have 3 possibilities Brit German Swede
Pet category is Dogs
House#2
Color category have 3 possibilities Blue Green Red
Nationality category have 3 possibilities Brit German Swede
Pet category have 3 possibilities Birds Dogs Fish
House#3
Color category is Blue
Nationality category is German
Pet category have 3 possibilities Birds Dogs Fish
Unsatisfied constraints are Constraint#1 Constraint#2 Constraint#3 Constraint#4
=> Puzzle not solved yet
====== Branch ======
Branching on House#1 Color category with 2 possibilities Blue Green
Trying possibility Green in House#1 Color category
[CALL] ====== Possible Assignments ======
House#1
Color category is Green
Nationality category have 3 possibilities Brit German Swede
Pet category is Dogs
House#2
Color category have 3 possibilities Blue Green Red
Nationality category have 3 possibilities Brit German Swede
Pet category have 3 possibilities Birds Dogs Fish
House#3
Color category is Blue
Nationality category is German
Pet category have 3 possibilities Birds Dogs Fish
Unsatisfied constraints are Constraint#1 Constraint#2 Constraint#3 Constraint#4
=> Puzzle not solved yet
====== Propagation ======
Applying Constraint#1 House#2 Color category changed from 3 possibilities Blue Green Red to 1
    ↪ possibilities Red
House#2 Pet category changed from 3 possibilities Birds Dogs Fish to 1 possibilities Birds
House#3 Pet category changed from 3 possibilities Birds Dogs Fish to 1 possibilities Fish
Remove Constraint#1 because it is satisfied
Applying Constraint#2 [CALL]
PHASE 1: Single-value logic for Green and Red under LEFT constraint
PHASE 2: Handling relation Green LEFT Red
Green is immediately LEFT of Red
[SEP] No changes from this constraint [RETURN]
```

## Reduction Rule (7)

```
[CALL] ====== Possible Assignments After Propagation ======
House#1
Color category have 2 possibilities Blue Green
Nationality category have 3 possibilities Brit German Swede
Pet category is Dogs
House#2
Color category have 3 possibilities Blue Green Red
Nationality category have 3 possibilities Brit German Swede
Pet category have 3 possibilities Birds Dogs Fish
House#3
Color category is Blue
Nationality category is German
Pet category have 3 possibilities Birds Dogs Fish
Unsatisfied constraints are Constraint#1 Constraint#2 Constraint#3 Constraint#4
=> Puzzle not solved yet
====== Branch ======
Branching on House#1 Color category with 2 possibilities Blue Green
Trying possibility Green in House#1 Color category
[CALL] ====== Possible Assignments ======
House#1
```

```
Color category is Green
Nationality category have 3 possibilities Brit German Swede
Pet category is Dogs
House#2
Color category have 3 possibilities Blue Green Red
Nationality category have 3 possibilities Brit German Swede
Pet category have 3 possibilities Birds Dogs Fish
House#3
Color category is Blue
Nationality category is German
Pet category have 3 possibilities Birds Dogs Fish
Unsatisfied constraints are Constraint#1 Constraint#2 Constraint#3 Constraint#4
=> Puzzle not solved yet
====== Propagation ======
Applying Constraint#1 House#2 Color category changed from 3 possibilities Blue Green Red to 1
    ↪ possibilities Red
House#2 Pet category changed from 3 possibilities Birds Dogs Fish to 1 possibilities Birds
House#3 Pet category changed from 3 possibilities Birds Dogs Fish to 1 possibilities Fish
Remove Constraint#1 because it is satisfied
Applying Constraint#2 No changes from this constraint
```

## Model Generation (8)

```
[CALL] ====== Possible Assignments After Propagation ======
House#1
Color category have 2 possibilities Blue Green
Nationality category have 3 possibilities Brit German Swede
Pet category is Dogs
House#2
Color category have 3 possibilities Blue Green Red
Nationality category have 3 possibilities Brit German Swede
Pet category have 3 possibilities Birds Dogs Fish
House#3
Color category is Blue
Nationality category is German
Pet category have 3 possibilities Birds Dogs Fish
Unsatisfied constraints are Constraint#1 Constraint#2 Constraint#3 Constraint#4
=> Puzzle not solved yet
====== Branch ======
Branching on House#1 Color category with 2 possibilities Blue Green
Trying possibility Green in House#1 Color category
[CALL] ====== Possible Assignments ======
House#1
Color category is Green
Nationality category have 3 possibilities Brit German Swede
Pet category is Dogs
House#2
Color category have 3 possibilities Blue Green Red
Nationality category have 3 possibilities Brit German Swede
Pet category have 3 possibilities Birds Dogs Fish
House#3
Color category is Blue
Nationality category is German
Pet category have 3 possibilities Birds Dogs Fish
Unsatisfied constraints are Constraint#1 Constraint#2 Constraint#3 Constraint#4
=> Puzzle not solved yet
====== Propagation ======
Applying Constraint#1 House#2 Color category changed from 3 possibilities Blue Green Red to 1
    ↪ possibilities Red
House#2 Pet category changed from 3 possibilities Birds Dogs Fish to 1 possibilities Birds
House#3 Pet category changed from 3 possibilities Birds Dogs Fish to 1 possibilities Fish
Remove Constraint#1 because it is satisfied
Applying Constraint#2 No changes from this constraint
Remove Constraint#2 because it is satisfied
Applying Constraint#3 [CALL]
PHASE 1: Single-value logic for Fish and Swede under RIGHT constraint
Removing German from House#1 Nationality category because German is pinned in another house
```

```
Removing German from House#2 Nationality category because German is pinned in another house
PHASE 2: Handling relation Fish RIGHT Swede
Fish is immediately RIGHT of Swede
Swede must be exactly one house to the LEFT , removing from House#1
Placing Swede in House#2 because Fish is pinned to House#3
[SEP] House#1 Nationality category changed from 3 possibilities Brit German Swede to 1 possibilities
    ↪  Brit
House#2 Nationality category changed from 3 possibilities Brit German Swede to 1 possibilities Swede
    ↪  [RETURN]
```

### Reduction Rule (8)

```
[CALL] ====== Possible Assignments After Propagation ======
House#1
Color category have 2 possibilities Blue Green
Nationality category have 3 possibilities Brit German Swede
Pet category is Dogs
House#2
Color category have 3 possibilities Blue Green Red
Nationality category have 3 possibilities Brit German Swede
Pet category have 3 possibilities Birds Dogs Fish
House#3
Color category is Blue
Nationality category is German
Pet category have 3 possibilities Birds Dogs Fish
Unsatisfied constraints are Constraint#1 Constraint#2 Constraint#3 Constraint#4
=> Puzzle not solved yet
====== Branch ======
Branching on House#1 Color category with 2 possibilities Blue Green
Trying possibility Green in House#1 Color category
[CALL] ====== Possible Assignments ======
House#1
Color category is Green
Nationality category have 3 possibilities Brit German Swede
Pet category is Dogs
House#2
Color category have 3 possibilities Blue Green Red
Nationality category have 3 possibilities Brit German Swede
Pet category have 3 possibilities Birds Dogs Fish
House#3
Color category is Blue
Nationality category is German
Pet category have 3 possibilities Birds Dogs Fish
Unsatisfied constraints are Constraint#1 Constraint#2 Constraint#3 Constraint#4
=> Puzzle not solved yet
====== Propagation ======
Applying Constraint#1 House#2 Color category changed from 3 possibilities Blue Green Red to 1
    ↪  possibilities Red
House#2 Pet category changed from 3 possibilities Birds Dogs Fish to 1 possibilities Birds
House#3 Pet category changed from 3 possibilities Birds Dogs Fish to 1 possibilities Fish
Remove Constraint#1 because it is satisfied
Applying Constraint#2 No changes from this constraint
Remove Constraint#2 because it is satisfied
Applying Constraint#3 House#1 Nationality category changed from 3 possibilities Brit German Swede to
    ↪  1 possibilities Brit
House#2 Nationality category changed from 3 possibilities Brit German Swede to 1 possibilities Swede
```

### Model Generation (9)

```
[CALL] ====== Possible Assignments After Propagation ======
House#1
Color category have 2 possibilities Blue Green
Nationality category have 3 possibilities Brit German Swede
Pet category is Dogs
House#2
Color category have 3 possibilities Blue Green Red
Nationality category have 3 possibilities Brit German Swede
```

```
Pet category have 3 possibilities Birds Dogs Fish
House#3
Color category is Blue
Nationality category is German
Pet category have 3 possibilities Birds Dogs Fish
Unsatisfied constraints are Constraint#1 Constraint#2 Constraint#3 Constraint#4
=> Puzzle not solved yet
====== Branch ======
Branching on House#1 Color category with 2 possibilities Blue Green
Trying possibility Green in House#1 Color category
[CALL] ====== Possible Assignments ======
House#1
Color category is Green
Nationality category have 3 possibilities Brit German Swede
Pet category is Dogs
House#2
Color category have 3 possibilities Blue Green Red
Nationality category have 3 possibilities Brit German Swede
Pet category have 3 possibilities Birds Dogs Fish
House#3
Color category is Blue
Nationality category is German
Pet category have 3 possibilities Birds Dogs Fish
Unsatisfied constraints are Constraint#1 Constraint#2 Constraint#3 Constraint#4
=> Puzzle not solved yet
====== Propagation ======
Applying Constraint#1 House#2 Color category changed from 3 possibilities Blue Green Red to 1
    ↪  possibilities Red
House#2 Pet category changed from 3 possibilities Birds Dogs Fish to 1 possibilities Birds
House#3 Pet category changed from 3 possibilities Birds Dogs Fish to 1 possibilities Fish
Remove Constraint#1 because it is satisfied
Applying Constraint#2 No changes from this constraint
Remove Constraint#2 because it is satisfied
Applying Constraint#3 House#1 Nationality category changed from 3 possibilities Brit German Swede to
    ↪  1 possibilities Brit
House#2 Nationality category changed from 3 possibilities Brit German Swede to 1 possibilities Swede
Remove Constraint#3 because it is satisfied
Applying Constraint#4 [CALL]
PHASE 1: Single-value logic for Brit and Birds under LEFT constraint
PHASE 2: Handling relation Brit LEFT Birds
Brit is immediately LEFT of Birds
[SEP] No changes from this constraint [RETURN]
```

### Reduction Rule (9)

```
[CALL] ====== Possible Assignments After Propagation ======
House#1
Color category have 2 possibilities Blue Green
Nationality category have 3 possibilities Brit German Swede
Pet category is Dogs
House#2
Color category have 3 possibilities Blue Green Red
Nationality category have 3 possibilities Brit German Swede
Pet category have 3 possibilities Birds Dogs Fish
House#3
Color category is Blue
Nationality category is German
Pet category have 3 possibilities Birds Dogs Fish
Unsatisfied constraints are Constraint#1 Constraint#2 Constraint#3 Constraint#4
=> Puzzle not solved yet
====== Branch ======
Branching on House#1 Color category with 2 possibilities Blue Green
Trying possibility Green in House#1 Color category
[CALL] ====== Possible Assignments ======
House#1
Color category is Green
Nationality category have 3 possibilities Brit German Swede
```

```
Pet category is Dogs
House#2
Color category have 3 possibilities Blue Green Red
Nationality category have 3 possibilities Brit German Swede
Pet category have 3 possibilities Birds Dogs Fish
House#3
Color category is Blue
Nationality category is German
Pet category have 3 possibilities Birds Dogs Fish
Unsatisfied constraints are Constraint#1 Constraint#2 Constraint#3 Constraint#4
=> Puzzle not solved yet
====== Propagation ======
Applying Constraint#1 House#2 Color category changed from 3 possibilities Blue Green Red to 1
    ↪ possibilities Red
House#2 Pet category changed from 3 possibilities Birds Dogs Fish to 1 possibilities Birds
House#3 Pet category changed from 3 possibilities Birds Dogs Fish to 1 possibilities Fish
Remove Constraint#1 because it is satisfied
Applying Constraint#2 No changes from this constraint
Remove Constraint#2 because it is satisfied
Applying Constraint#3 House#1 Nationality category changed from 3 possibilities Brit German Swede to
    ↪ 1 possibilities Brit
House#2 Nationality category changed from 3 possibilities Brit German Swede to 1 possibilities Swede
Remove Constraint#3 because it is satisfied
Applying Constraint#4 No changes from this constraint
```

## Model Generation (10)

```
[CALL] ====== Possible Assignments After Propagation ======
House#1
Color category have 2 possibilities Blue Green
Nationality category have 3 possibilities Brit German Swede
Pet category is Dogs
House#2
Color category have 3 possibilities Blue Green Red
Nationality category have 3 possibilities Brit German Swede
Pet category have 3 possibilities Birds Dogs Fish
House#3
Color category is Blue
Nationality category is German
Pet category have 3 possibilities Birds Dogs Fish
Unsatisfied constraints are Constraint#1 Constraint#2 Constraint#3 Constraint#4
=> Puzzle not solved yet
====== Branch ======
Branching on House#1 Color category with 2 possibilities Blue Green
Trying possibility Green in House#1 Color category
[CALL] ====== Possible Assignments ======
House#1
Color category is Green
Nationality category have 3 possibilities Brit German Swede
Pet category is Dogs
House#2
Color category have 3 possibilities Blue Green Red
Nationality category have 3 possibilities Brit German Swede
Pet category have 3 possibilities Birds Dogs Fish
House#3
Color category is Blue
Nationality category is German
Pet category have 3 possibilities Birds Dogs Fish
Unsatisfied constraints are Constraint#1 Constraint#2 Constraint#3 Constraint#4
=> Puzzle not solved yet
====== Propagation ======
Applying Constraint#1 House#2 Color category changed from 3 possibilities Blue Green Red to 1
    ↪ possibilities Red
House#2 Pet category changed from 3 possibilities Birds Dogs Fish to 1 possibilities Birds
House#3 Pet category changed from 3 possibilities Birds Dogs Fish to 1 possibilities Fish
Remove Constraint#1 because it is satisfied
Applying Constraint#2 No changes from this constraint
```

```
Remove Constraint#2 because it is satisfied
Applying Constraint#3 House#1 Nationality category changed from 3 possibilities Brit German Swede to
    ↪   1 possibilities Brit
House#2 Nationality category changed from 3 possibilities Brit German Swede to 1 possibilities Swede
Remove Constraint#3 because it is satisfied
Applying Constraint#4 No changes from this constraint
Remove Constraint#4 because it is satisfied
[SEP] [CALL] ====== Possible Assignments After Propagation ======
House#1
Color category is Green
Nationality category is Brit
Pet category is Dogs
House#2
Color category is Red
Nationality category is Swede
Pet category is Birds
House#3
Color category is Blue
Nationality category is German
Pet category is Fish
Unsatisfied constraints are [RETURN]
```

### Reduction Rule (10)

```
[CALL] ====== Possible Assignments After Propagation ======
House#1
Color category have 2 possibilities Blue Green
Nationality category have 3 possibilities Brit German Swede
Pet category is Dogs
House#2
Color category have 3 possibilities Blue Green Red
Nationality category have 3 possibilities Brit German Swede
Pet category have 3 possibilities Birds Dogs Fish
House#3
Color category is Blue
Nationality category is German
Pet category have 3 possibilities Birds Dogs Fish
Unsatisfied constraints are Constraint#1 Constraint#2 Constraint#3 Constraint#4
=> Puzzle not solved yet
====== Branch ======
Branching on House#1 Color category with 2 possibilities Blue Green
Trying possibility Green in House#1 Color category
[CALL] ====== Possible Assignments After Propagation ======
House#1
Color category is Green
Nationality category is Brit
Pet category is Dogs
House#2
Color category is Red
Nationality category is Swede
Pet category is Birds
House#3
Color category is Blue
Nationality category is German
Pet category is Fish
Unsatisfied constraints are
```

### Model Generation (11)

```
[CALL] ====== Possible Assignments After Propagation ======
House#1
Color category have 2 possibilities Blue Green
Nationality category have 3 possibilities Brit German Swede
Pet category is Dogs
House#2
Color category have 3 possibilities Blue Green Red
Nationality category have 3 possibilities Brit German Swede
```

```
Pet category have 3 possibilities Birds Dogs Fish
House#3
Color category is Blue
Nationality category is German
Pet category have 3 possibilities Birds Dogs Fish
Unsatisfied constraints are Constraint#1 Constraint#2 Constraint#3 Constraint#4
=> Puzzle not solved yet
====== Branch ======
Branching on House#1 Color category with 2 possibilities Blue Green
Trying possibility Green in House#1 Color category
[CALL] ====== Possible Assignments After Propagation ======
House#1
Color category is Green
Nationality category is Brit
Pet category is Dogs
House#2
Color category is Red
Nationality category is Swede
Pet category is Birds
House#3
Color category is Blue
Nationality category is German
Pet category is Fish
Unsatisfied constraints are
=> Puzzle is solved
[SEP] Solution House#1
Color category is Green
Nationality category is Brit
Pet category is Dogs
House#2
Color category is Red
Nationality category is Swede
Pet category is Birds
House#3
Color category is Blue
Nationality category is German
Pet category is Fish [RETURN]
```

## Reduction Rule (11)

```
[CALL] ====== Possible Assignments After Propagation ======
House#1
Color category have 2 possibilities Blue Green
Nationality category have 3 possibilities Brit German Swede
Pet category is Dogs
House#2
Color category have 3 possibilities Blue Green Red
Nationality category have 3 possibilities Brit German Swede
Pet category have 3 possibilities Birds Dogs Fish
House#3
Color category is Blue
Nationality category is German
Pet category have 3 possibilities Birds Dogs Fish
Unsatisfied constraints are Constraint#1 Constraint#2 Constraint#3 Constraint#4
=> Puzzle not solved yet
====== Branch ======
Branching on House#1 Color category with 2 possibilities Blue Green
Trying possibility Green in House#1 Color category
Solution House#1
Color category is Green
Nationality category is Brit
Pet category is Dogs
House#2
Color category is Red
Nationality category is Swede
Pet category is Birds
House#3
```

```
Color category is Blue
Nationality category is German
Pet category is Fish
```

## Model Generation (12)

```
[CALL] ====== Possible Assignments After Propagation ======
House#1
Color category have 2 possibilities Blue Green
Nationality category have 3 possibilities Brit German Swede
Pet category is Dogs
House#2
Color category have 3 possibilities Blue Green Red
Nationality category have 3 possibilities Brit German Swede
Pet category have 3 possibilities Birds Dogs Fish
House#3
Color category is Blue
Nationality category is German
Pet category have 3 possibilities Birds Dogs Fish
Unsatisfied constraints are Constraint#1 Constraint#2 Constraint#3 Constraint#4
=> Puzzle not solved yet
====== Branch ======
Branching on House#1 Color category with 2 possibilities Blue Green
Trying possibility Green in House#1 Color category
Solution House#1
Color category is Green
Nationality category is Brit
Pet category is Dogs
House#2
Color category is Red
Nationality category is Swede
Pet category is Birds
House#3
Color category is Blue
Nationality category is German
Pet category is Fish
 [SEP] Solution House#1
Color category is Green
Nationality category is Brit
Pet category is Dogs
House#2
Color category is Red
Nationality category is Swede
Pet category is Birds
House#3
Color category is Blue
Nationality category is German
Pet category is Fish [RETURN]
```

## Reduction Rule (12)

```
Solution House#1
Color category is Green
Nationality category is Brit
Pet category is Dogs
House#2
Color category is Red
Nationality category is Swede
Pet category is Birds
House#3
Color category is Blue
Nationality category is German
Pet category is Fish
```

## Model Generation (Final Response)

```
Solution House#1
Color category is Green
Nationality category is Brit
Pet category is Dogs
House#2
Color category is Red
Nationality category is Swede
Pet category is Birds
House#3
Color category is Blue
Nationality category is German
Pet category is Fish
=> House#3 owns the Fish
=> the German owns the Fish <|endoftext|>
```

