# OpenReview forum: "PENCIL: Long Thoughts with Short Memory"
_ICML.cc/2025/Conference — ICML 2025 poster_

### Official Review · Reviewer_yxYs · 2025-02-28

**Overall Recommendation:** 4

**Summary:**

This paper introduces PENCIL, a novel method designed to overcome a fundamental limitation of standard chain‐of‐thought (CoT) reasoning in language models. The main idea is to interleave token generation with a reduction mechanism that “cleans up” intermediate reasoning steps—using specially defined tokens (e.g., [CALL], [SEP], [RETURN])—so that the context remains compact. This approach reduces the maximal context length from being proportional to the time complexity (often exponential) to being proportional to the actual space required (polynomial). The paper provides both theoretical results (showing that PENCIL can simulate Turing machines with optimal time and space efficiency) and extensive empirical evidence on challenging tasks such as SAT, QBF, and Einstein’s puzzle, where PENCIL achieves significantly higher accuracy and efficiency than standard CoT methods.

## update after rebuttal

I thank the authors for their detailed response. They provided helpful clarifications regarding the novelty and motivation for the proposed reduction rule, positioning it effectively against other context management techniques. The explanation of the training process and how the model learns to utilize the special tokens addressed my concerns about feasibility. Furthermore, the discussion on the theoretical claims regarding Turing machine simulation and the potential generality beyond the tested tasks was insightful. My assessment of the paper's contribution has been positively updated.

**Claims And Evidence:**

The paper makes several claims:

- That traditional CoT suffers from an unbounded accumulation of intermediate steps, leading to inefficient use of memory.
- That a simple reduction rule can compress the reasoning trace, reducing memory requirements from exponential to polynomial in many cases.
- That PENCIL, by interleaving generation and reduction, can simulate universal computation (i.e., a Turing machine) efficiently.
- That empirical results on hard reasoning tasks (e.g., a 97% success rate on a 5×5 Einstein puzzle with a relatively small model) support these claims.

The authors support these claims with a combination of rigorous theoretical analysis (including formal definitions and proofs) and comprehensive experimental evaluations across multiple benchmarks. While some proofs are deferred to the appendix, the provided sketches and derivations appear convincing.

**Essential References Not Discussed:**

While the paper cites a broad range of relevant literature, one potential gap is a deeper discussion of recent work on dynamic token pruning and memory compression in transformers (e.g., related to recent studies on efficient attention mechanisms). Including a comparison with methods that also aim to limit context length (even if via different techniques) could further strengthen the discussion.

**Experimental Designs Or Analyses:**

The experiments are designed to test both the accuracy and efficiency of PENCIL relative to standard CoT approaches. The benchmarks chosen (SAT, QBF, and various sizes of Einstein’s puzzle) are well established in the literature. The analyses include both convergence speed and scalability (in terms of inference time and maximal context length). The experimental design appears sound, with clear comparisons across different problem sizes and model capacities. One suggestion might be to include additional ablations that isolate the effect of the reduction mechanism from other architectural choices.

**Methods And Evaluation Criteria:**

The proposed method is well motivated and builds on the core idea of managing memory via reduction—analogous to garbage collection or stack unwinding in classical computation. The evaluation is conducted on standard yet challenging benchmarks (SAT, QBF, Einstein’s puzzle) that are appropriate for testing reasoning and memory efficiency. Metrics such as accuracy, trace rate, maximal sequence length, and inference time are used to compare PENCIL with standard CoT, and the evaluation criteria are both rigorous and relevant to the claims made.

**Other Comments Or Suggestions:**

The paper is well written and the key ideas are communicated clearly. A few minor typographical errors and formatting inconsistencies were noted in the supplementary material. Future revisions might also consider a broader discussion on potential limitations or failure cases of the reduction mechanism, especially in tasks where intermediate reasoning steps are critical for later stages.

**Other Strengths And Weaknesses:**

### Strengths:

1. The paper introduces a clear and intuitive mechanism for reducing memory footprint during reasoning.
1. The theoretical analysis is rigorous and establishes strong claims regarding universal computation.
1. Empirical results are compelling, demonstrating significant improvements on challenging tasks with modest model sizes.

### Weaknesses:

1. Some proofs and technical details are deferred to the appendix; a more self-contained presentation could aid clarity.
2. Additional ablation studies might help to isolate the impact of the reduction mechanism versus other design choices.
3. A more extensive discussion comparing with other memory-efficient transformer techniques such as state space models could be beneficial.

**Questions For Authors:**

1. Could the authors provide further ablation studies to isolate the contribution of the reduction rule from other architectural modifications? How sensitive is the performance to the exact design of the special tokens and reduction triggers?

2. While the paper shows impressive results on SAT, QBF, and Einstein’s puzzle, how does PENCIL perform on real-world tasks or benchmarks that involve natural language understanding beyond synthetic reasoning problems?

3.  Can the authors elaborate on how PENCIL compares with other recent memory compression or dynamic token pruning methods in terms of both efficiency and accuracy?

**Relation To Broader Scientific Literature:**

PENCIL is positioned at the intersection of chain-of-thought reasoning and memory-efficient computation. It builds upon previous works on CoT prompting (e.g., Wei et al., 2022) and extends the idea by introducing memory reduction techniques that are reminiscent of classical programming language strategies such as tail recursion and garbage collection. The paper effectively relates its contributions to prior work on external memory augmentation in LLMs as well as theoretical studies on the expressivity of transformers. It clearly advances the discussion by addressing the scalability limitations inherent in unbounded CoT approaches.

**Theoretical Claims:**

The paper includes several theoretical claims, notably that PENCIL (and its variant SCROLL) can simulate a Turing machine with time and space complexities that are optimal (i.e., time proportional to T(M, x) and space proportional to S(M, s, x)). I reviewed the sketch proofs in the main text and appendix (e.g., the formulation of the iterative next-token generator and the corresponding state function). While some details are deferred, the arguments are logically consistent and align with known theoretical frameworks. No major issues were found with the presented theoretical claims, although a more detailed step-by-step verification of the proofs would be beneficial in future revisions.

---

> ### Author Rebuttal · Authors · 2025-03-31
>
> > **Weakness 1: "Some proofs and technical details are deferred to the appendix; a more self-contained presentation could aid clarity."**
>
> Thank you for the suggestion. We will incorporate key technical details from the Appendix into the main paper once we have an additional page in the final version.
>
> > **Weakness 2 & Q1: "Could the authors provide further ablation studies to isolate the contribution of the reduction rule from other architectural modifications? How sensitive is the performance to the exact design of the special tokens and reduction triggers?"**
>
> We kindly note that in all our experiments, we have controlled for architectural and implementation choices by fixing the transformer architecture (including number of parameters, context window size, positional encoding, etc.) and the training algorithm (optimizer, batch size, learning rate, etc.). The only difference between CoT and PENCIL is in the data preprocessing (see the first three paragraphs of Section 4) and whether the reduction rule is applied during inference. This ensures the performance difference stems from the reduction rule.
>
> As shown in Section 4, when all other factors are fixed, PENCIL consistently outperforms CoT, especially on larger problems. Moreover, the superior performance is insensitive to the specific choice of architectures. For example, in Figure 7, when we vary the model size (for both CoT and PENCIL) and context window size, our approach still achieves better performance. Empirically, the performance gap is also insensitive to the positional encoding (e.g. simple absolute PE works as well).
>
> While the precise design of the reduction rule might affect performance, we have not yet found an alternative rule that achieves better space and time efficiency than the one proposed (Equation 1). That being said, we conjecture if there does exist a better rule, PENCIL should achieve better performance and larger gap with CoT.
>
>
> > **Q2: "While the paper shows impressive results on SAT, QBF, and Einstein’s puzzle, how does PENCIL perform on real-world tasks or benchmarks that involve natural language understanding beyond synthetic reasoning problems?"**
>
> It is indeed very promising to adapt PENCIL to general real-world tasks that involve natural language understanding. One potential way is to generate datasets with special tokens and fine-tune existing LLMs. See  However, this requires considerate engineering efforts, and thus we leave this as future work. Also see a discussion of "how PENCIL can be applied to standard LLMs" in our response to Reviewer zuSb.
>
> > **Weakness 3 & Q3: "A more extensive discussion comparing with other memory efficient transformer techniques such as state space models could be beneﬁcial; Can the authors elaborate on how PENCIL compares with other recent memory compression or dynamic token pruning methods in terms of both efficiency and accuracy?"**
>
> Existing memory-efficient architectures, such as those employing linear complexity attention (e.g., [1]), still require the context length to grow with the running time for problem solving, and thus do not address the fundamental limitation of CoT that PENCIL overcomes. While state-space models (e.g. [2]) avoid this issue by storing only the state, but this often comes at the cost of reduced expressiveness; for instance, SSMs have been shown (empirically and theoretical) to struggle with even simple tasks like copying [3].
>
> Other memory compression and token pruning methods (e.g., [4–7]) typically combine base models with external heuristic algorithms (such as those relying on score functions and ranking). As a result, the models do not have the capability to reduce the space themselves whereas PENCIL explicit trains the model to do so. Moreover, these methods do not have theoretical benefits of solving arbitrary problems with optimal space complexity that PENCIL offers.
>
> PENCIL essentially differs from these lines of works by handling the limitation of the next-token generation (i.e. CoT) paradigm, and is orthogonal to the contributions of aforementioned papers. In other words, PENCIL is compatible with different base model choices and existing memory compression techniques. We will incorporate a more comprehensive discussion on related work in the next revision.
>
> [1] Rethinking attention with performers, ICLR 2021
>
> [2] Mamba: Linear-time sequence modeling with selective state spaces, 2023
>
> [3] Repeat After Me: Transformers are Better than State Space Models at Copying, ICML 2024
>
> [4] Efficient streaming language models with attention sinks, ICLR 2024
>
> [5] H2o: Heavy-hitter oracle for efficient generative inference of large language models, NeurIPS 2023
>
> [6] Model tells you what to discard: Adaptive kv cache compression for llms, ICLR 2024
>
> [7] Llmlingua: Compressing prompts for accelerated inference of large language models, EMNLP 2023

---

### Official Review · Reviewer_TGaA · 2025-03-12

**Overall Recommendation:** 3

**Summary:**

The paper focused on the CoT reasoning, and proposed a PENCIL framework with reduction mechanism to exclude the unnecessary parts in the CoT. The authors conducted experiments on SAT, QBF and Einstein’s Puzzle to demonstrate the effectiveness of the framework. The authors also proved that the framework could simulate a Turing machine.
## update after rebuttal
The responses have addressed most of my concerns. I raised my score to 3.

**Claims And Evidence:**

Overall the claims are supported by the experimental results and the theoretically analysis. But the claim, that the framework could reduce the exponential CoT growing to polynomial, seems to only work for the SAT-like reasoning task.

**Essential References Not Discussed:**

The related works are well cited.

**Experimental Designs Or Analyses:**

The authors conduct experiments on three SAT-like tasks under different difficulties, and the experimental results demonstrated the effectiveness of the proposed framework. But the authors only conducted experiments on tasks similar to SAT, and lacked results on more general reasoning tasks to show its generalizability. Besides, the authors only compare the performances of the fine-tuned small LLMs. It would be better to provide results on larger LLMs and other CoT reasoning methods to further support the necessity of the reduction mechanism.

**Methods And Evaluation Criteria:**

Some concerns are listed as follows.
1. How to ensure that the reduced parts are really no longer necessary especially for more general reasoning tasks.
2. The framework needs to repeatedly input the existing steps into the LLM to get the output. But initially the LLM could generate the CoT in one call, and the computation can be greatly reduced with KV cache.

**Other Comments Or Suggestions:**

None.

**Other Strengths And Weaknesses:**

In summary, the strengths of the paper are listed as follows.
1. The paper proposed a novel reduction mechanism to reduce the CoT lengths and improve reasoning efficiency and capability.
2. The authors conducted experiments to demonstrate the effectiveness of the framework, and theoretically proved that the proposed framework could simulate Turing machine.

The weaknesses are as follows.
1. The three datasets used in the paper are all SAT-like ones, which lacks discussion on generalizability to more general reasoning. Besides, the reduced CoT growing from exponential to polynomial only works for the SAT-like tasks.
2. It’s unclear how to determine whether the reduced parts are truly unnecessary. And the repeatedly LLM call may greatly increases computation complexity compared with generating CoT in one call.
3. The proposed framework is easy to follow, but the authors introduced the method in a way quite hard to understand. It would be better to simplify the introduction, such as adding some examples and explanations.
4. The context window of current LLMs could be extended to quite large, such as 1M. So I wonder whether it’s necessary to remove unnecessary parts from CoT.

**Questions For Authors:**

1. Can the proposed framework and findings work on more general reasoning tasks.
2. How to determine whether the reduced parts are truly unnecessary.
3. As the context window of LLMs could be quite large, is it necessary to remove unnecessary parts from the CoT.

**Relation To Broader Scientific Literature:**

The paper proposed a novel reduction mechanism to reduce the CoT lengths and improve reasoning efficiency and capability. The authors further theoretically proved that the proposed framework could simulate Turing machine.

**Theoretical Claims:**

There are one lemma and one theorem in the paper. I roughly go through the proof in the appendix, and have not found issues.

---

> ### Author Rebuttal · Authors · 2025-03-31
>
> > **Weakness 1 & Q1: "The three datasets used in the paper are all SAT-like ones, which lacks discussion on generalizability to more general reasoning. Besides, the reduced CoT growing from exponential to polynomial only works for the SAT-like tasks."**
>
> We choose SAT and QBF because they are representative NP-complete and PSPACE-complete problems that no existing algorithm can solve efficiently, and thus they are ideal for stress-testing a model's capability in handling hard reasoning tasks. Moreover, PENCIL can be extended to arbitrary tasks provided the underlying algorithms can be written in a pure functional style. See our response to Reviewer xx5n's Q2 where we have briefly described the high-level idea.
>
> To work on general reasoning tasks, one potential approach is to fine-tune existing LLMs on datasets with special tokens (generated either manually or automatically), enabling the model to learn to reason in a structured manner and think longer to solve more complicated tasks. See detailed discussion of how PENCIL can be applied to standard LLMs in our response to Reviewer zuSb. We will elaborate on this in the next version and leave further extensions as future work.
>
> > **Weakness 2 & Q2 & Concern 1 in Methods And Evaluation Criteria: "How to ensure that the reduced parts are really no longer necessary especially for more general reasoning tasks."**
>
> For our data generation process, the reduced parts are guaranteed to be unnecessary, because when generating the dataset, the special tokens are placed in the trace in strict accordance with how function calls are used in the (Python) codes that implement the algorithm for solving the task.
>
> For general tasks whose structured reasoning trace is not explicitly included in the training set, there is no absolute guarantee that the model will always remove the unneeded parts. Nevertheless, this is a common limitation of all structured reasoning approaches (CoT, for example, does not always output useful intermediates steps that contribute to the final answer for general reasoning tasks, even models are trained to do so). Moreover, as we mentioned earlier, one potential way to mitigate this limitation is to fine-tune LLMs on appropriately formatted datasets, allowing them to learn to extract useful information from previous thoughts and use such a skill in general reasoning tasks.
>
>
> > **Weakness 2 & Concern 2 in Methods And Evaluation Criteria: "The repeatedly LLM call may greatly increases computation complexity compared with generating CoT in one call."**
>
> It is important to note that ***PENCIL does not repeatedly call LLMs, nor does it increase computation complexity; instead, it significantly improves the efficiency compared with one-pass CoT***.
>
> Specifically, as has been discussed in the last paragraph of Section 2.2, for each application of reduction $$\textbf{C} \texttt{[CALL]} \textbf{T} \texttt{[SEP]} \textbf{A} \texttt{[RETURN]} \Rightarrow
> \textbf{C}\textbf{A}$$ the same model is used to generate new tokens following the same sequence, rather than calling a new LLM and feeding the reduced sequence into it. The benefit of sticking to the same model and the sequence is that the KV cache of the context $\textbf{C}$ can be preserved, and only the KV cache for \textbf{A} should be recomputed, which incurs marginal cost, as reflected in Equation 8 that shows the minimal FLOPs (where KV cache usage is optimized) needed for PENCIL. Intuitively, PENCIL significantly saves computation because for each generated token the prefix length is significantly smaller than its CoT counterpart, while the total number of generated tokens is the same. Empirically, Figure 6 demonstrates that PENCIL is significantly more computationally efficient than direct CoT for each problem instance.
>
> > **Weakness 3: "It would be better to simplify the introduction, such as adding some examples and explanations."**
>
> Thanks for the suggestion. We will improve the clarity of introduction in the next revision.
>
> > **Weakness 4 & Q3: "The context window of current LLMs could be extended to quite large, such as 1M. So I wonder whether it’s necessary to remove unnecessary parts from CoT."**
>
> Indeed, there are many existing efforts trying to extend the context window of LLMs, see our response to Reviewer yxYs's Q3 for a detailed discussion and how PENCIL fundamentally differs from them. Our contribution is orthogonal to their efforts contributions of aforementioned papers, i.e. one can always combine PENCIL with larger context window to potentially even achieve better performance.
>
> Moreover, it has been argued that enlarging the context window size can introduce issues such as diminished ability to retrieve relevant information from a very long context (see, e.g. [1]), whereas PENCIL is immune to such an issue by completely eliminating the unneeded thoughts.
>
> [1] Liu, Nelson F., et al. "Lost in the middle: How language models use long contexts." ACL 2024.

---

> > ### Comment · Reviewer_TGaA · 2025-04-04
> >
> > Thanks for your time on answering my questions. The responses have addressed most of my concerns. I have raised my score.

---

### Official Review · Reviewer_xx5n · 2025-03-12

**Overall Recommendation:** 4

**Summary:**

The paper introduces PENCIL, an extension of the Chain-of-Thought (CoT) approach for language models. PENCIL addresses the "write-only" limitation of CoT, where intermediate reasoning steps accumulate indefinitely in the context, by incorporating a reduction mechanism. This mechanism uses special tokens ([CALL], [SEP], [RETURN]) to structure reasoning and discards unnecessary intermediate thoughts once a computation completes, reducing the maximal context length from exponential to polynomial for certain tasks. The main algorithmic idea is an iterative process alternating between CoT-style generation and a reduction rule (C [CALL] T [SEP] A [RETURN] ⇒ CA).

The paper evaluates PENCIL on SAT (Satisfiability), QBF (Quantified Boolean Formulas), and Einstein's puzzle, demonstrating significant context length reductions, high accuracy, and scalability to complex problems. Theoretically, PENCIL can simulate a Turing machine with O(T) tokens and O(S) maximal sequence length, where T is time and S is space.

**Claims And Evidence:**

The paper makes three primary claims:

1. **PENCIL reduces maximal context length from exponential to polynomial for certain tasks.**
    - **Evidence**: Empirical results show dramatic reductions (e.g., SAT: 13,804 to 2,507 tokens; QBF: 151,661 to 649 tokens at n=10; Einstein’s puzzle: 151,192 to 3,335 tokens for 5x5). Figure 4 provide clear comparisons with standard CoT.
2. **PENCIL achieves high accuracy on challenging reasoning tasks.**
    - **Evidence**: Table 1 shows 100% accuracy and trace rate for SAT and QBF up to n=10, outperforming baseline CoT. Table 2 reports 97% accuracy on the 5x5 Einstein’s puzzle versus 25% for CoT.
3. **PENCIL simulates a Turing machine with optimal time and space complexity.**
    - **Evidence**: Section 5 shows theoretical justifications that PENCIL can simulate a Turing machine with O(T) tokens and O(S) maximal sequence length.

The results and evidence are strong, with empirical data and theoretical arguments aligning with the claims.

**Essential References Not Discussed:**

I can't notice any.

**Experimental Designs Or Analyses:**

Three tasks are used in the experiment:

- **SAT and QBF**: Using a 6-layer 10M-parameter transformer, PENCIL achieves 100% accuracy up to n=10, with context length reductions validated by Figure 4.
- **Einstein’s Puzzle**: An 8-layer 25M-parameter transformer achieves 97% accuracy on the 5x5 version.
- The author also analyzes the convergence speed and conducts an ablation study on model size.

The designs are valid. My only concern is the model used in this study is too small compared to the current LLMs. This could disadvantage standard CoT approaches and limit the generalizability of the paper's conclusions to larger models.

**Methods And Evaluation Criteria:**

The methods and evaluation criteria are well-suited to the problem:

- **Methods**: PENCIL combines a next-token predictor (transformer) with a reduction rule, applied iteratively. The rule’s design, inspired by function calls, is well-motivated and effectively addresses context overflow in CoT.
- **Evaluation Criteria**: Tasks (SAT, QBF, Einstein’s puzzle) are compositional problems and computationally intensive, making them suitable for testing context efficiency and scalability.

The choice of benchmarks and metrics are appropriate for the problem.

**Other Comments Or Suggestions:**

See Questions

**Other Strengths And Weaknesses:**

**Strengths:**

- The proposed method is novel and well-motivated;
- Empirical results are strong and comprehensive;
- The paper is well-structured, with intuitive explanations and illustrative figures.

**Weaknesses:**

- The data generation process requires knowing the reasoning structure. How reasoning problems without a clear structure (e.g., math problems) can benefit from this approach remains a question.
- Models used in this study are too small compared to the current LLMs. This could disadvantage standard CoT approaches and limit the generalizability of the paper's conclusions to larger models.

**Questions For Authors:**

- How trace rate is calculated?
- How was the training data for Einstein's Puzzle generated? Additionally, how did you transform the algorithm's solutions into text? Did you use templates to verbalize the solutions?
- What is the complexity of the problem in the training data? Are all test problems in-domain in terms of complexity, or are some of them OOD?
- Could the model size and data generation process explain CoT's poor performance? Shah et al. (2024) trained transformers on Einstein puzzles with carefully constructed CoT and demonstrated that the model can achieve a high solve rate.
- Do you think this method can applied directly for fine-tuning LLMs?

Shah, Kulin, et al. "Causal language modeling can elicit search and reasoning capabilities on logic puzzles." arXiv preprint arXiv:2409.10502 (2024).

**Relation To Broader Scientific Literature:**

This work builds on the Chain-of-Thought (CoT) framework, addressing its scalability problem in context length. It might improve LLM performance in length generalization. It also connects to prior research on compressing model contexts.

**Theoretical Claims:**

I reviewed the theoretical results in Section 5 and Appendix B, and they appear valid to me. However, I'm not an expert in computational theory, so my judgment might be incorrect.

---

> ### Author Rebuttal · Authors · 2025-03-31
>
> > **Weakness 1: "The data generation process requires knowing the reasoning structure. How reasoning problems without a clear structure (e.g., math problems) can beneﬁt from this approach remains a question."**
>
> Indeed, language models do not inherently reason in a way that allows for convenient space reduction. To extend PENCIL to general reasoning problems, one potential approach is to fine-tune LLMs on specialized datasets, either manually labeled or automatically generated, so that models learn to reason in a structured, memory-efficient manner (see our response to “How PENCIL can be applied to standard LLMs” for Reviewer zuSb).
>
> Moreover, we want to kindly point out that many math problems actually do exhibit PENCIL-like structures. For example, lemmas consist of statements which needs to be remembered, and proofs which do not. Similarly, many math problems involve intermediate computations with a complex derivation which can be forgotten, leaving only a concise final expression that needs to be retained. This natural separation in mathematical reasoning directly aligns with the PENCIL approach.
>
> > **Weakness 2: "Models used in this study are too small compared to the current LLMs. This could disadvantage standard CoT approaches and limit the generalizability of the paper's conclusions to larger models."**
>
> While we have not yet conducted experiments on real-world LLMs (which we leave as future work), we believe using larger models with larger context window would further amplify the advantages of PENCIL.
>
> This is because, in terms of theoretical expressiveness, CoT requires the maximal context length to grow with the running time of a problem, whereas PENCIL only requires it to grow with the needed memory; the gap between time and space is significant (e.g. exponential) for inherently hard reasoning problems. Although this gap is less pronounced for smaller models on tasks, it would become much more significant on larger-scale problems.
>
> > **Q1: "How trace rate is calculated?"**
>
> Let $x$ be the ground-truth reasoning trace, and $\hat x$ be the reasoning trace generated by the model (as defined in Equation 8). The trace rate is defined as $$\frac{1}{\max\{|x|, |\hat x|\}}\sum_{i=1}^{\min\{|x|, |\hat x|\}} \mathbf 1(x_i =\hat x_i)$$, which quantifies the percentage of correctly predicted tokens. We choose this metric because it is both a direct measure of sequence similarity and tractable even for very long traces.
>
> > **Q2: "How was the training data for Einstein's Puzzle generated? Additionally, how did you transform the algorithm's solutions into text? Did you use templates to verbalize the solutions?"**
>
> We implemented the Einstein’s Puzzle solver in Python, and as the code runs, it uses templates to verbalize the key steps. For example, when removing an entry from all possibilities, the code generates thoughts like:
>
> *Since green must be immediately to the right of Birds, we remove “green” from House \#1 (it can’t be in the leftmost position if it’s supposed to be on the right of something else)"*
>
> Special tokens are appended automatically. The general rule is as follows: when the code calls a new function, it appends the "[CALL]" token to the trace, and when the function finishes all computations and is ready to return, it appends "[SEP] A [RETURN]" where A is the returned value. (And if the returned value is from another function, we use the tail recursion in Equation 10 to further optimize the space.)
>
> In fact, this method can be generalized to any algorithm written in a pure functional style; we will detail this further in the next version and open-source our code once published.
>
>
> > **Q3: "What is the complexity of the problem in the training data? Are all test problems in-domain in terms of complexity, or are some of them OOD?"**
>
> The complexity of the problems in the training data matches that of the test problems (i.e., the same n). We plan to explore extensions to the OOD setting as future work.
>
> > **Q4: "Could the model size and data generation process explain CoT's poor performance?"**
>
> It is possible that both the model size and the data generation format affect CoT’s performance. As shown in Figure 7, increasing the model size and context length improves CoT performance. And if we keep increasing the size, presumably CoT will eventually also solve $5\times 5$ or more complicated puzzles, but PENCIL will be able to go even further. That is, regardless of the model size and context window, we have shown PENCIL can consistently solve larger-scale problems than the problems CoT can solve with that model size. We will add [1] in the next version.
>
> [1] Shah, Kulin, et al. “Causal language modeling can elicit search and reasoning capabilities on logic puzzles.” arXiv:2409.10502 (2024).
>
> > **Q5: "Do you think this method can applied directly for fine-tuning LLMs?"**
>
> Yes, we believe that fine-tuning LLMs on structured datasets generated as described is a promising future direction.

---

> > ### Comment · Reviewer_xx5n · 2025-04-02
> >
> > Thanks for the clarifications. I maintain my positive assessment of the work!

---

### Official Review · Reviewer_zuSb · 2025-03-13

**Overall Recommendation:** 4

**Summary:**

The paper proposes PENCIL, a next-token generation scheme that incorporates a reduction mechanism to control the length of the generated sequence. This mechanism removes redundant context, enabling more efficient generation while reducing memory usage. Experimentally, transformers trained on 3SAT, QBF, and Einstein’s puzzle demonstrate that PENCIL achieves higher accuracy compared to standard un-reduced Chain-of-Thought reasoning. Additionally, it converges faster during optimization when evaluated under the same FLOP constraints. Theoretically, PENCIL is capable of universal computation.

**Claims And Evidence:**

The claims are generally supported.

**Essential References Not Discussed:**

To the reviewer's knowledge, all essential references are discussed.

**Experimental Designs Or Analyses:**

The experimental designs are sound.

**Methods And Evaluation Criteria:**

The proposed method and the evaluation setup make sense.

The primary concern is the applicability of PENCIL to standard large language models. In the current setup, the model is trained to generate CoT reasoning that can be systematically reduced, stemming from the well-structured solutions in 3SAT, QBF, and Einstein’s puzzle. The training data explicitly follow this structured pattern, so that the model learn this pattern to be reduced. However, in practical settings where such structured CoT data are unavailable and the solution patterns are unknown, how can PENCIL be effectively applied?

**Other Comments Or Suggestions:**

Line 365 (left): There is a duplicate "since".

**Other Strengths And Weaknesses:**

### Strength:

1. The problem addressed in the paper is important and relevant.
2. The paper is well-written and clearly presented.

**Questions For Authors:**

Besides the concern regarding how PENCIL can be applied to standard LLMs, the reviewer has two additional questions:

1. Effectiveness of Thought Reduction

The reviewer suggests that the authors include a discussion on the conditions under which PENCIL can effectively reduce the reasoning process to save memory. The examined problems in the paper exhibit a clear structured reasoning process, where intermediate steps can be meaningfully reduced without affecting the final solution. However, there exist other problem types where individual steps may not involve much computation, yet the reasoning process must retain or summarize all intermediate computational results.

For example, in longest increasing subsequence and subset sum problems, dynamic programming approaches must maintain all possible combinations' results (lengths or sums) and track these combinations to construct the final solution. In such cases, there may be limited potential for reduction, as the computation needs to be preserved throughout the process. The feasibility of reduction also depends on how solution patterns are designed. A discussion on this limitation would strengthen the paper.

2. Expressiveness of PENCIL in Relation to Finite-Precision Transformers

Recent work (e.g., Merrill et al. 2023, cited in the paper) has investigated the expressiveness of Chain-of-Thought (CoT) reasoning under finite-precision transformer constraints. Could the theoretical analysis be extended to demonstrate whether PENCIL expands the complexity class of problems that CoT can solve while maintaining the same rate of CoT length?

The reviewer is willing to increase the score if some of the concerns and questions raised in this review are adequately addressed.

**Relation To Broader Scientific Literature:**

CoT is widely used for reasoning with LLMs, enabling them to solve problems by generating long reasoning chains. However, recent concerns have emerged regarding the efficiency of this approach, particularly given the minimum sequence length required to solve complex problems with finite-precision transformers.

This paper addresses this concern by proposing a reduction scheme that adaptively shortens the reasoning chain, reducing memory usage while preserving the model’s reasoning capabilities.

**Theoretical Claims:**

Before Theorem 5.4, the paper constructs an efficient solution by defining $t^i$ as the smallest integer larger than $t_{i-1}$ such that the length of ... is no more than half of the sequence length of .... The proof for the existence of $t^i$ is missing.

---

> ### Author Rebuttal · Authors · 2025-03-31
>
> > **How PENCIL can be applied to standard LLMs**
>
> The way we envision applying PENCIL to standard LLMs is to fine-tune LLMs on examples that include our special tokens, with the goal that model learns to reason in a structured manner that leverages memory efficiently and enables longer reasoning. Such datasets can be generated by domain experts (which is arduous but fairly common for LLM alignment), possibly using automatic anotation of common structures in structured mathematical or logical writting. One can also generate such datasets automatically by converting the running trace of *any algorithms* written in a pure functional programming language into a training set (see our response to Reviewer xx5n's Q2 for how this can be done for Einstein's puzzle); correspondingly, the maximal context length for PENCIL is proportional to the maximal stack memory (typically much smaller than the running time). We will discuss in detail how to generate such datasets for general algorithms in the next version of the paper.
>
> While we have not yet applied PENCIL to standard LLMs (this requires significant resources, both for data collection and training), we nonetheless have provided empirical evidence by showing even small language models can learn to generate structured CoT. We will more comprehensively discuss potential ways to apply PENCIL to real-world LLMs and leave this as future work.
>
> > **Q1: Effectiveness of Thought Reduction**
>
> In principle, PENCIL can effectively reduce the reasoning trace for any problem whose space complexity is smaller than its time complexity, as we formally prove in Section 5. While the time and space gap is typically significant, there does exist cases where all computations are useful and should be preserved, as the reviewer suggested; it is true in this case the reduction is not necessarily useful. We appreciate the reviewer’s point and will include a discussion in the next version.
>
>
> > **Q2: Expressiveness of PENCIL in Relation to Finite-Precision Transformers**
>
> Yes, Theorem 5.4 can be extended to cases where the base model is a transformer. Particularly, we can prove that PENCIL with a fixed finite-size decoder-only transformer can perform universal space-efficient computation, by simulating Turing machine running in $T$ steps and $S$ space with $\mathcal O(T)$ generated tokens and maximal sequence length $\mathcal O(S)$. This is a significant improvement over standard CoT, which requires context length to grow proportionally with $\mathcal O(T)$ (i.e. results in [1]). An immediate implication is that PENCIL can solve all $\mathrm{PSPACE}$ problems using polynomial context-length, whereas CoT can only solve problems in $\mathrm{P}$.
>
> The high-level idea of the proof is as follows: for any Turing machine $\mathcal M$, we can construct a transformer (under some architectural specifications such as average-hard attention used in [1]) that can executed Algorithm 1 during its forward pass. Specifically, the transformer should be able to: (1) simulate the step-by-step execution of the Turing machine, (2) detect when to generate the $\texttt{[SEP]}$ token indicating the start of summarization, and (3) summarize thoughts by computing a compressed state representation of the current token sequence. Although the construction is not straight-forward, we will include the proof in the paper once we can make updates.
>
> [1] W. Merrill and A. Sabharwal. "The expresssive power of transformers with chain of thought." ICLR 2024.
>
> > **Minors**
>
> $t_i$ ($i>0$) always exists for Turing machines whose running time is at least twice of the maximal memory, otherwise we have $\mathcal O(T(\mathcal M,x)) = \mathcal O(S(\mathcal M, s, x))$ and reduction is unnecessary as space optimality is automatically achieved. We will also correct other typos the reviewer pointed out.

---

### Decision · Program_Chairs · 2025-05-01

**Decision:**

Accept (poster)

**Comment:**

This paper introduces a novel approach that addresses CoT reasoning's context length limitations by incorporating a reduction mechanism that cleans up intermediate computations. The method uses special tokens to structure reasoning and discard unnecessary thoughts, reducing context length from exponential to polynomial for complex tasks.

Its key strengths include: its strong theoretical foundation (ie, proving PENCIL can simulate Turing machines with optimal complexity), impressive empirical results (97% accuracy on Einstein's puzzle with a small 25M model), and clear potential to tackle problems intractable with standard CoT.

During rebuttal, the authors effectively addressed concerns about applicability to general reasoning and standard LLMs, computational efficiency, and necessity given growing context windows. While there are limitations in the current implementation's focus on structured reasoning tasks, the significant contribution to advancing language model reasoning capabilities merits acceptance.